# Large contribution of fossil-derived components to aqueous secondary organic aerosols in China

Buqing Xu [1,2], Gan Zhang [1,2] ✉, Örjan Gustafsson [3] ✉, Kimitaka Kawamura[4], Jun Li [1,2], August Andersson [3], Srinivas Bikkina[4,5], Bhagawati Kunwar[4], Ambarish Pokhrel[4,6], Guangcai Zhong[1,2], Shizhen Zhao [1,2], Jing Li[1,2], Chen Huang[1,2], Zhineng Cheng [1,2], Sanyuan Zhu[1,2], Pingan Peng [1,2] & Guoying Sheng[1,2]

Incomplete understanding of the sources of secondary organic aerosol (SOA) leads to large uncertainty in both air quality management and in climate change assessment. Chemical reactions occurring in the atmospheric aqueous phase represent an important source of SOA mass, yet, the effects of anthropogenic emissions on the aqueous SOA (aqSOA) are not well constrained. Here we use compound-specific dual-carbon isotopic fingerprints ($\delta^{13}C$ and $\Delta^{14}C$) of dominant aqSOA molecules, such as oxalic acid, to track the precursor sources and formation mechanisms of aqSOA. Substantial stable carbon isotope fractionation of aqSOA molecules provides robust evidence for extensive aqueous-phase processing. Contrary to the paradigm that these aqSOA compounds are largely biogenic, radiocarbon-based source apportionments show that fossil precursors produced over one-half of the aqSOA molecules. Large fractions of fossil-derived aqSOA contribute substantially to the total water-soluble organic aerosol load and hence impact projections of both air quality and anthropogenic radiative forcing. Our findings reveal the importance of fossil emissions for aqSOA with effects on climate and air quality.

Organic aerosol (OA) is the dominant component of atmospheric fine particulate mass[1]. The majority of OA is secondary (SOA), formed in the atmosphere from the oxidation of precursor gases[2]. Most SOA are composed of oxygenated and hygroscopic compounds, thereby, have harmful impacts on respiratory health and large, and uncertain effects on atmospheric radiative forcing[3]. However, current global models typically underpredict SOA magnitude, distribution, and dynamics, suggesting a limited understanding of their sources and formation processes[4]. An increasing number of modeling and experimental studies point toward aqueous chemical reactions occurring in cloud droplets and wet aerosols as an important missing pathway for SOA formation[2–8]. Aqueous SOA (aqSOA) formation often involves mixed effects of anthropogenic and biogenic emissions[2, 3]. The extent to which aqSOA is controllable or "natural" is an important subject of ongoing research for both air quality and climate issues[2, 3]. However, the poorly understood aqueous chemical processes and the complexity of anthropogenic-biogenic interactions renders any firm conclusion hard to come by[2].

In model simulations, aqSOA are put to largely derive from the natural biogenic precursors, such as the oxidation products of

[1]State Key Laboratory of Organic Geochemistry, Guangzhou Institute of Geochemistry, Chinese Academy of Sciences, Guangzhou 510640, China. [2]CAS Center for Excellence in Deep Earth Science, Guangzhou 510640, China. [3]Department of Environment Science and the Bolin Centre for Climate Research, Stockholm University, Stockholm 10691, Sweden. [4]Chubu Institute for Advanced Studies, Chubu University, Kasugai 487-8501, Japan. [5]Present address: CSIR-National Institute of Oceanography, Dona Paula 403004 Goa, India. [6]Present address: Institute of Science and Technology, Tribhuvan University, Kathmandu 44600, Nepal. ✉e-mail: zhanggan@gig.ac.cn; orjan.gustafsson@aces.su.se

isoprene[3, 4, 8]. Fossil precursors are noticed to be less polar and less hydrophilic than biogenic precursors, thus diminishing the possibility of partitioning to the aqueous phase, and subsequent aqSOA formation[9]. However, a few recent studies in East Asia suggest that fossil precursors may substantially contribute to the formation of aqSOA. For example, fast aqueous-phase oxidation of polycyclic aromatic hydrocarbons emitted from fossil fuel combustion was observed in Beijing winter haze and may explain the observed SOA[10]. Radiocarbon-based estimations attributed ~50% of water-soluble OA to fossil sources in the East Asia outflow[11,12], compared to <30% in Europe, U.S., and South Asia[13–18]. Nevertheless, on aqSOA formation, a quantitative estimation of the fossil contribution remains challenging, largely due to the lack of specific and reliable technical tools in tracking the sources and formation mechanism of aqSOA in ambient aerosols.

Here we leverage the recent advent of compound-specific dual-carbon isotope fingerprint ($\delta^{13}C-\Delta^{14}C$) of aqSOA molecules[19] to quantify and characterize aqSOA sources and atmospheric chemical processes. The molecular-level $\Delta^{14}C$ contents of aqSOA tracers will provide direct constrain on their origins[19], whereas $\delta^{13}C$ fingerprints of molecular tracers can differentiate various atmospheric processes/reactions[20]. The technique was applied to purified oxalic acid and other similarly abundant organic acids (e.g., glyoxylic acid). Oxalic acid, likely represents the highest oxidation state of OA, has frequently been used as a proxy for aqueous processing in both laboratory and field studies[21–23]. Organic acids that have high O/C ratios (around 1–2), are among the most abundant SOA components and are key end products in the aqueous-phase photochemical oxidation of various volatile organic compounds (VOCs) and intermediates such as glyoxal (Gly) and methylglyoxal (MeGly) in clouds or wet aerosols[4, 5, 24, 25]. Therefore, these organic acids can serve as signature compounds to trace the aqSOA formation pathways[21–23, 26].

We used a set of year-round aerosol samples from a regional receptor site (Heshan Atmospheric Environmental Monitoring Superstation, Supplementary Fig. 1) in South China. We measured the aqSOA compounds and their $\delta^{13}C-\Delta^{14}C$ isotope fingerprints during two contrasting phases of the East Asia monsoon system, i.e., continental outflow vis-à-vis South China Sea (SCS) air masses (Fig. 1). By combining the dual-carbon isotopic compositions and other chemical indicators together with meteorological parameters, we elucidate the relative contribution of fossil and biomass aqSOA precursors and their subsequent evolution processes, and provide ambient isotopic observational evidence for a large aqSOA formation from anthropogenic fossil emissions. Finally, the dual $\delta^{13}C-\Delta^{14}C$ characterization of aqSOA molecules are extended to wider spatial coverage of China, which further confirm a ubiquitous contribution of aqueous-phase transformation of fossil precursors to organic aerosols.

## Results and discussion

### Meteorological setting and aerosol molecular compositions

The Heshan receptor site features an East Asia Monsoon Climate. The air is controlled by continental monsoon in winter and oceanic monsoon in summer, respectively (Fig. 1). Backward trajectories revealed that clean SCS air mass dominated the airshed during June-August 2017, while continental outflow rendered by enhanced anthropogenic emissions from the Pearl River Delta (PRD) city clusters, dominated during September 2017 to March 2018 (Fig. 1). Therefore, we categorized the 2017-2018 samples into two groups based on their distinctly different air mass transport regimes (Supplementary Table 1), i.e., "coastal background ($n = 11$)" and "continental outflow ($n = 21$)" (see Supplementary Fig. 2 for air mass clusters and Supplementary Fig. 3 for relative cluster contribution).

The chemical composition of aerosols during the campaign varied greatly with the air mass transport regimes. Concentrations of gas-phase pollutants ($SO_2$, $NO_2$, $O_3$, CO) and major components of $PM_{2.5}$ during the continental outflow were 2-5 times higher than during the coastal background (Supplementary Table 2 and Supplementary Fig. 4). The observed organic matter (OM) and anthropogenic water-soluble inorganic constituents ($WSIC_{anth}$) (Supplementary Method 1)

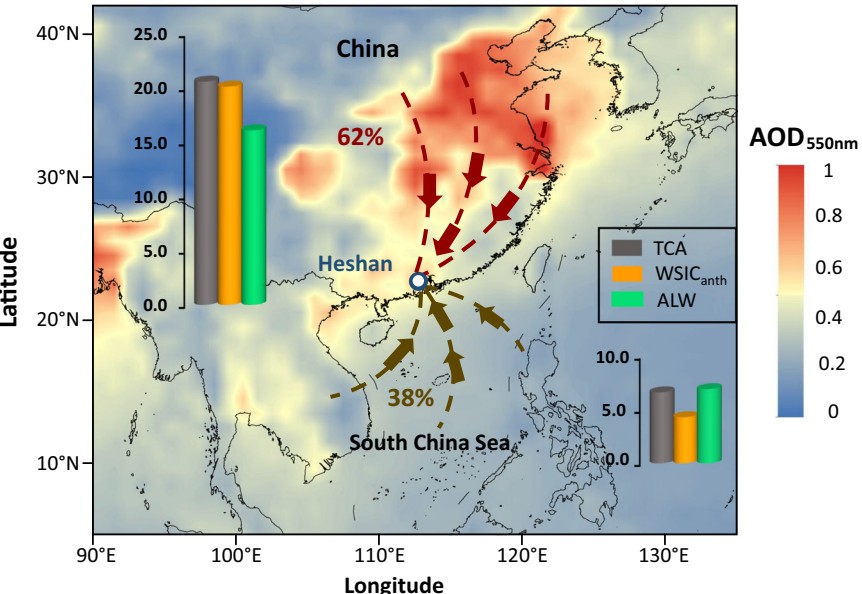

**Fig. 1 | Meteorology and general aerosol chemical characteristics of the study site (Heshan city).** The average aerosol optical depth (AOD) at 550 nm during June 2017 to May 2018 over the East Asia region are shown. The mean air mass back trajectories (BT) based on HYSPLIT cluster analysis show two dominant air mass transport pathways (dashed line and arrows): continental outflow (62% of the total clusters) and oceanic air masses (38% of total clusters). See Supplementary Fig. 2 for the 72-h BT with a 6-h interval and Supplementary Fig. 3 for relative cluster contribution. The bar charts depict the mass concentrations of total carbonaceous aerosol [TCA = organic matter (OM) + elemental carbon (EC)], anthropogenic WSIC [$WSIC_{anth}$ = non-sea-salt (nss) $K^+$ + $nss-SO_4^{2-}$ + $NH_4^+$ + $NO_3^-$ + $nss-Cl^-$], and aerosol liquid water (ALW). The AOD data is obtained from Moderate Resolution Imaging Spectroradiometer (MODIS) observations (https://giovanni.gsfc.nasa.gov/giovanni/). The coastline boundaries in the map are originated from Natural Earth free vector map data (https://www.naturalearthdata.com/). The administration boundaries in the map are originated from map products of National Geomatics Center of China (https://www.webmap.cn/).

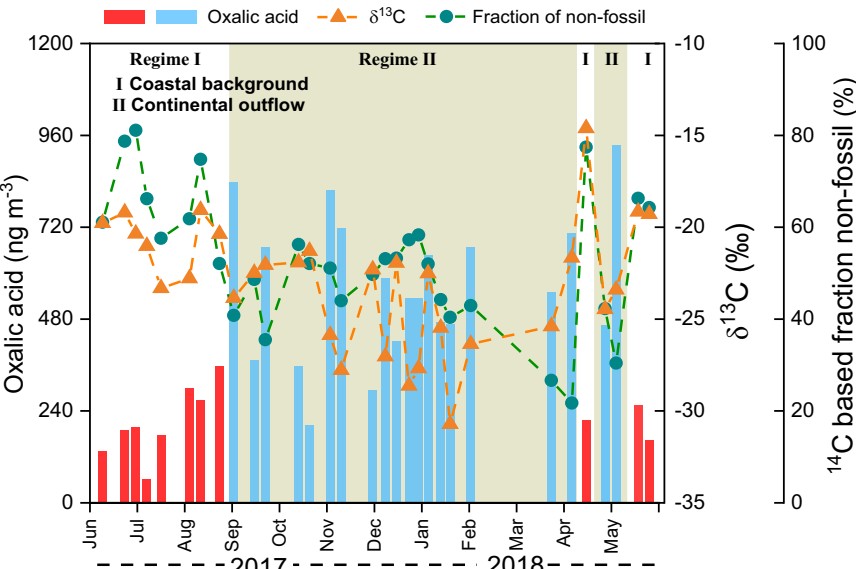

**Fig. 2 | Year-round variability of mass concentration and carbon isotopic composition of oxalic acid at the Heshan city.** The unshaded periods represent coastal background air mass regime, while the shadowed periods represent continental outflow air mass regime. Red and blue bars represent the concentration of oxalic acid during coastal background air mass regime and continental outflow air mass regime, respectively. The orange and green symbols show the $\delta^{13}C$ composition and $^{14}C$-based fraction of non-fossil for oxalic acid.

increased in continental outflow samples, along with the aerosol liquid water (ALW) content associated with hygroscopic aerosol components (Supplementary Method 2). The concentrations of OM, WSIC$_{anth}$, and ALW show a clear increase in continental outflow samples with 3, 5, and 3 times higher values than the coastal background, respectively (Fig. 1).

The molecular composition of diacids and related aqueous processing tracers are compiled in Supplementary Table 3. The malonic to succinic acid ratios ($C_3/C_4$) are significantly higher in the coastal background samples ($1.5 \pm 0.4$) than the continental outflow ($0.9 \pm 0.3$) (Supplementary Table 4). Since malonic acid is produced by photochemical oxidation of succinic acid in the atmosphere, this indicates that the coastal background aerosols have undergone more extensive photochemical aging[27, 28]. The ratio of water-soluble organic carbon (WSOC) to organic carbon (WSOC/OC) is another indicator of atmospheric photochemical aging[29]. For example, the WSOC/OC ratios increased from ~40% in South Asian source region to ~70% in the Indian Ocean receptor site due to the photochemical aging of OC during long-range atmospheric transport[17,29]. However, in this study, the WSOC/OC ratios of the continental outflow samples ($44 \pm 12\%$) cannot be distinguished statistically from the coastal background ($39 \pm 10\%$) (Supplementary Table 4), implying that additional mechanisms other than atmospheric aging may exist.

The uptake of water-soluble gaseous organic precursors to wet aerosol particles and fog/clouds, with subsequent aqueous-phase transformation to lower volatile compounds could be a significant pathway for WSOC aerosol formation[23, 26, 30, 31]. ALW mass was driven primarily by inorganic salts, especially by nitrate and sulfate[31]. As shown in Supplementary Fig. 5, the ALW mass well correlated with the concentrations of WSIC$_{anth}$ and fractions of WSIC$_{anth}$ in PM$_{2.5}$ during the campaign. The nitrate in the continental outflow samples display concentrations 20 times higher than the coastal background samples, and account for nearly half of the anthropogenic WSIC (Supplementary Fig. 4). Because nitrate is more hygroscopic than sulfate[30], such changes in aerosol compositions would be a dominant factor driving the ALW content. Given the high relative humidity (RH%) ($74 \pm 9\%$; Supplementary Table 2) and the abundant hygroscopic particles (particularly for particulate nitrate) in continental outflow samples, we expect that the changes in the gas-particle partitioning driven by the increased ALW could have enhanced the formation of aqSOA species,

such as oxalic acid[23, 26, 32]. This can be supported by the sixfold higher aqSOA precursors (Gly + MeGly) mass fractions in OC observed in the continental outflow samples than the coastal background (Supplementary Table 4). Observation in continental outflow of higher aqSOA precursor concentrations and higher WSOC/OC ratios is consistent with aqSOA formation processes. Yet more precise and robust evidence/information is needed to give insights into the aqSOA formation.

## Stable carbon isotope forensics of aqueous processing of aerosols

The $\delta^{13}C$ signatures reflect both sources and atmospheric processing. Emissions from different sources often have different end-member $\delta^{13}C$ values. Once formed, the $\delta^{13}C$ signature is then influenced by kinetic isotope effects (KIE) during atmospheric reactions/processing[15]. Secondary aerosols stemming from the oxidation of precursor gases due to the KIE typically lead to lower $\delta^{13}C$ values in particulate products compared to the gaseous reactants, whereas aerosol aging (i.e., oxidation of organic matter in the aerosol phase) can lead to a release of isotopically lighter carbon such as in $CO_2$ and CO, leaving residual (aged) aerosols enriched in $^{13}C$[12, 15, 17, 29]. Figure 2 shows the temporal variation of the mass concentrations and dual-carbon isotopic signatures of oxalic acid during the year-round observation. The $\delta^{13}C$ values for oxalic acid are on average $-24.6 \pm 2.7‰$ in the continental outflow samples and substantially more enriched in $^{13}C$ in the coastal background samples ($-19.9 \pm 2.3‰$). This clear contrast can likely be attributable to different emission sources and/or atmospheric processing.

Source mixing is not expected to have a major influence on the $\delta^{13}C$ composition of oxalic acid in this study. First, the $\delta^{13}C$ compositions of azelaic acid ($C_9$ diacids), an oxidation product of biogenic unsaturated fatty acids[33], in the coastal background samples ($-34.1 \pm 2.8‰$) and continental outflow ($-34.7 \pm 2.6‰$) are very close (Fig. 3a). These $\delta^{13}C$ values point to unsaturated fatty acids of C3 plant origin ($-38.5‰$ to $-32.5‰$)[34], while excluding marine phytoplankton and C4 plant sources that both instead are more enriched in $\delta^{13}C$[35, 36]. This is consistent with the limited marine-biogenic contributions as indicated by the mass ratios of nss-SO$_4^{2-}$ to SO$_4^{2-}$ ($99 \pm 1\%$) and nss-K$^+$ to K$^+$ ($94 \pm 4\%$) in all samples[29]. Given contributions from marine and C4

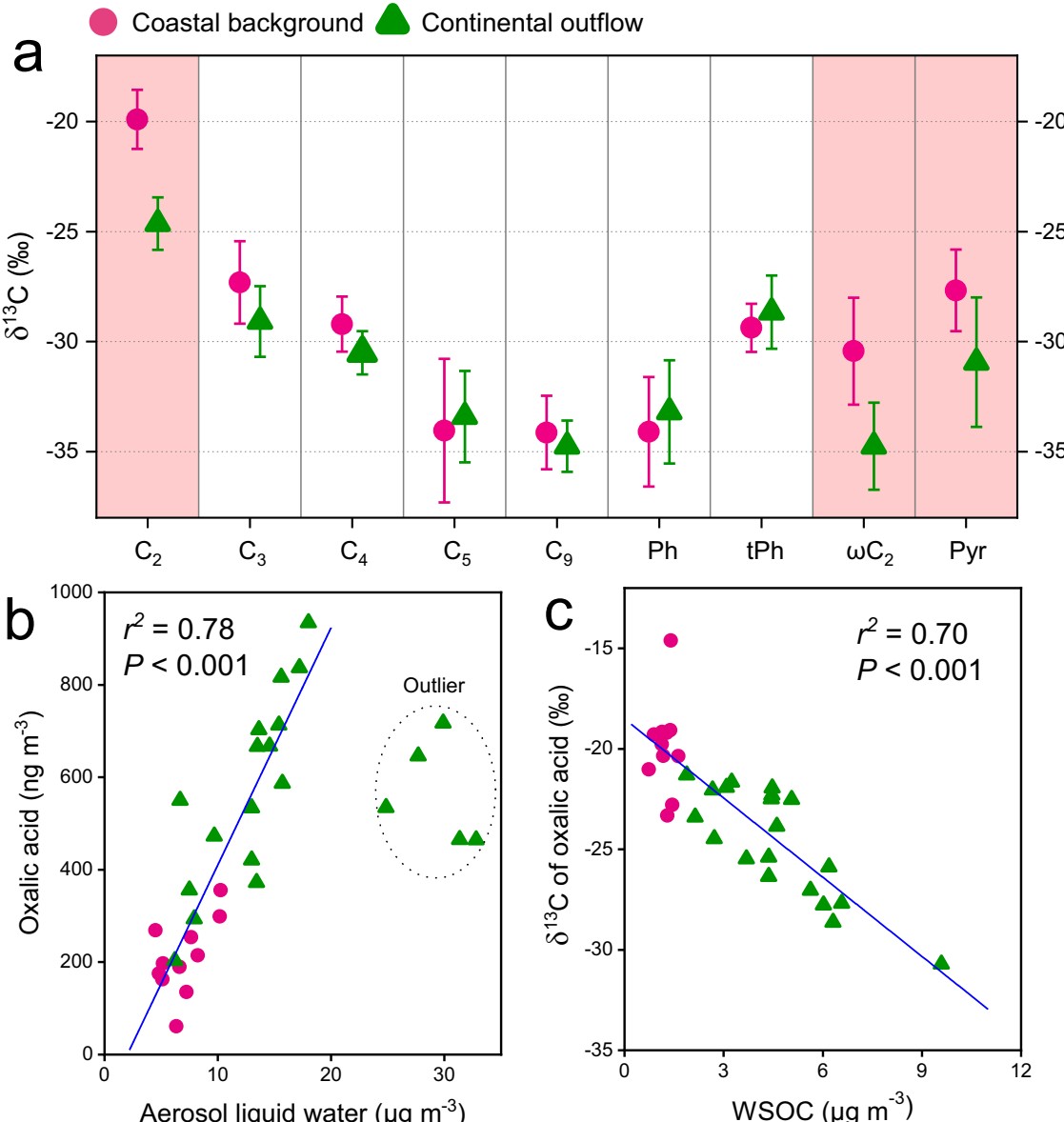

**Fig. 3 | Stable carbon isotopic evidence of aqueous secondary organic aerosol (aqSOA) formation. a** Average $\delta^{13}C$ values of saturated aliphatic dicarboxylic acids ($C_2$–$C_9$), aromatic dicarboxylic acids [phthalic acid (Ph) and terephthalic acid (tPh)], and oxocarboxylic acids [glyoxylic acid ($\omega C_2$) and pyruvic acid (Pyr)] for the two air mass regimes. The error bars indicate the 95% confidence interval of $\delta^{13}C$ values in each air mass regime. Pink shadows represent the compounds for which the $\delta^{13}C$ values are statistically significantly different ($p < 0.05$) between the two air mass origins. **b** Relationships between oxalic acid mass concentration and aerosol liquid water (ALW). **c** Relationships between $\delta^{13}C$-oxalic acid and mass concentration of water-soluble organic carbon (WSOC) as a proxy for aqSOA abundance.

plants sources are mostly excluded at the Heshan site, the potential precursors of oxalic acid would then be anthropogenic VOCs (AVOCs; e.g., aromatic hydrocarbons) and C3 plant-derived biogenic VOCs (BVOCs; e.g., isoprene), of which the $\delta^{13}C$ values are similar (Supplementary Table 5). Therefore, the isotopic range of various sources of oxalic acid at the supersite is minimal, leaving kinetic isotopic fractionation the most probable processes controlling the $\delta^{13}C$ composition of oxalic acid.

Oxalic acid is an SOA end product and can be formed by two pathways. The first pathway is a stepwise photochemical breakdown of longer-chain aliphatic diacids[27, 28], which leads to relatively enriched $\delta^{13}C$ values in the residual shorter-chain diacids[37]. Recently, significant amounts of oxalic acid followed by succinic acid were also found to be produced via ozone oxidation of isoprene under dry condition[38]. The second pathway is the oxidation of BVOCs and AVOCs to semi-volatile organic compounds (SVOCs, e.g., Gly and MeGly), followed by

partitioning of SVOCs to wet aerosols or cloud/fog droplets, with subsequent aqueous-phase transformation to pyruvic acid (Pyr) and glyoxylic acid ($\omega C_2$) and finally oxidized to oxalic acid[24–26]. In this case, oxalic acid-C carries a depleted $\delta^{13}C$ signal as the oxalic acid is an oxidation product, as opposed to a residual C pool. Our dataset displays a significant correlation of oxalic acid with ALW ($r^2 = 0.78$, $P < 0.001$, Fig. 3b), suggesting that oxalic acid was dominantly formed via gas-to-liquid transfer of precursors and subsequent aqueous-phase reactions. This is consistent with a previous model study revealing that aqSOA formation is limited by ALW rather than by the availability of precursors[39]. As shown in Fig. 3b, the five points with excessive ALW are off the trend, signifying that ALW was no longer the limiting parameter in the specific samples. Oxalic acid and its aqueous-phase intermediate product, glyoxylic acid, also display a significant correlation ($r^2 = 0.65$, $P < 0.001$), which stands for the aqueous-phase pathway.

The compound-specific $\delta^{13}C$ values of oxalic acids and relevant SOA species can help to explore the formation pathway. First, the higher homologs aliphatic diacids from $C_3$ to $C_9$, in contrast to $C_2$, show no significant statistical difference (t-test, $P > 0.05$, Supplementary Table 6) between the two air mass source regimes (Fig. 3a). Instead, $\omega C_2$ and Pyr show significantly lower (t-test, $P < 0.05$, Supplementary Table 6) $\delta^{13}C$ values of (by 4.4‰ and 3.2‰, respectively, Fig. 3a) during continental outflow than coastal background. As aqSOA formed from VOCs or SVOCs typically leads to lower $\delta^{13}C$ values relative to the precursors since lighter-isotope-containing precursors are preferentially oxidized to form reaction products. The continental outflow regime was characterized by high RH% and substantial anthropogenic hygroscopic particles, which could facilitate aqSOA formation. Therefore, the vast difference in the $\delta^{13}C$ composition of oxalic acid between the two air mass source regimes could be mainly resulted from the aqueous-phase reaction pathway, i.e., AVOCs/BVOCs $\rightarrow$SVOCs/WSOC $\rightarrow$ Pyr $\rightarrow$ $\omega C_2 \rightarrow C_2$[40], rather than gas-phase photochemical oxidation (or breakdown) processes. Note that the source mixing contribution is minimal as mentioned above.

This aqueous reaction effect on isotope fractionation may also apply to water-soluble OA, which is a cocktail of primary OA (e.g., sugars) and SOA species. Figure 3c reveals a clear decrease in $\delta^{13}C$ signatures of oxalic acid against an increase in WSOC concentrations ($r^2 = 0.70$, $P < 0.001$). Atmospheric aqueous-phase processing would simultaneously cause a decrease in the $\delta^{13}C$ signature of reaction products and an increase in the WSOC mass concentrations. This is precisely what was observed during this campaign. Overall, the results indicate that aqSOA formation processes may be a significant route to form water-soluble OA in the atmosphere of South China and similar settings.

## Molecular-level radiocarbon isotope forensics of aqSOA sources

Besides $\delta^{13}C$ signature, which undergoes strong isotope fractionation during atmospheric processing, radiocarbon isotopic fingerprint ($\Delta^{14}C$) is conservative and highly precise in distinguishing fossil from biogenic/biomass sources[19]. Individual water-soluble diacids and related compounds ($C_2$, $C_3$, $C_4$, $\omega C_2$, and MeGly) were isolated by preparative capillary gas chromatography techniques (Methods). It is possible to harvest a sufficient mass of $C_2$ oxalic acid from aerosol samples, while the other isolated compounds had to each be pooled per month to obtain sufficient carbon mass for the $^{14}C$ measurement. The determined $\Delta^{14}C$ compositions reported as the "fraction of modern carbon" ($F_m$) are shown in Supplementary Table 7 and Supplementary Table 8. It is further possible to translate the $F_m$ signatures into fractional contribution from fossil ($f_{fossil}$) versus non-fossil (including biogenic emissions: bio and biomass burning: bb; hereafter referred as $f_{bio/bb}$) sources (Supplementary Method 3).

The excellent sensitivity and precision of molecular-level $^{14}C$ analysis reveal a clear diversity in $^{14}C$ signatures of oxalic acid for different samples and regimes (Fig. 2 and Supplementary Table 7). In the coastal background air mass regime, $67 \pm 9\%$ of oxalic acid is from biogenic emission and biomass burning. Although summer monsoon originating from SCS (Fig. 1) results in efficient precipitation, there is still a substantial fossil contribution ($f_{fossil} = 19$–48%; $33 \pm 9\%$) from anthropogenic activities. For the continental outflow samples, the $f_{fossil}$–$C_2$ ($f_{fossil} = 42$–78%; $55 \pm 10\%$) are even higher than biogenic/biomass source contributions. Back trajectory analysis reveals that during the latter period the receptor site was downwind from highly polluted continental source regions, which thus apparently is a source of oxalic acid in the regional aerosol (Fig. 1). The high ratios of $NO_3^-$ in atmospheric particles (~10%; Supplementary Fig. 4) indicate a significant contribution of traffic emissions[41] to atmospheric particles. Moreover, terephthalic acid (tPh) was one of the most abundant anthropogenic diacids accounting for 5% of total diacids (Supplementary Table 3). As tPh is a combustion tracer of plastic polymers[42], this highlights the involvement of precursors from plastic burning to the formation of diacids. Thus, both traffic and plastic burning emissions were readily among the contributors to the fossil source-derived oxalic acid.

The mass ratio of adipic acid to azelaic acid ($C_6/C_9$) is often used to discriminate the relative significance of biogenic versus anthropogenic source contribution to diacids. Here, $C_6$ adipic acid is the photochemical oxidation product of cyclic hexene emitted from the fossil fuel combustion sources[27, 43], whereas $C_9$ azelaic acid is a photochemical degradation product of biogenic unsaturated fatty acids[33]. Lower $C_6/C_9$ ratios ($0.37 \pm 0.13$) were observed in the coastal background samples than continental outflow ($0.76 \pm 0.21$). Furthermore, $C_6/C_9$ ratios correlates well with the parameter of $f_{fossil}$–$C_2$ ($r^2 = 0.63$, $P < 0.001$; Supplementary Fig. 6). Hence, $C_6/C_9$ ratios gives strong and consistent support to the result of diacids $^{14}C$-based source apportionment.

The fossil contents of oxalic acid in the continental outflow samples are compared with its presumed precursors of longer-chain diacids ($C_3$ and $C_4$ diacids) and its aqueous processing precursors ($\omega C_2$ and MeGly) (Fig. 4c and Supplementary Table 8). The result indicates that fossil sources were the overwhelming contributor to both oxalic acid ($55 \pm 10\%$) and to its aqueous processing precursors, i.e., glyoxylic acid ($\omega C_2$; $69 \pm 4\%$) and methylglyoxal (MeGly; $67 \pm 5\%$). Therefore, a dominance from fossil fuel sources to their aqSOA formation in the continental outflow is obvious.

Although malonic $C_3$ and succinic $C_4$ acids are believed to be partly produced from aqueous processing[22, 26], there is for these $C_3$ and $C_4$ a somewhat lower but still large contribution from fossil sources ($C_3$: $31 \pm 4\%$; $C_4$: $36 \pm 2\%$). This demonstrates that $C_3$ and $C_4$ diacids instead were mainly originated from the stepwise photochemical breakdown of biogenic unsaturated fatty acids even in highly polluted continental outflow air mass. The large contrast of fossil fraction among $C_2$ diacids and its higher homologs (i.e., $C_3$ and $C_4$ diacids) further indicate that substantial fossil precursors contribute to the formation of $C_2$ through aqueous-phase processing, which results in a much higher fossil content of $C_2$ than of its higher homologs.

The $^{14}C$-derived concentrations of oxalic acid from non-fossil sources (bio/bb–$C_2$) span from 40 to 417 ng m$^{-3}$ ($207 \pm 88$ ng m$^{-3}$), whereas from fossil source (fossil–$C_2$) are 21 to 650 ng m$^{-3}$ ($230 \pm 168$ ng m$^{-3}$) (Supplementary Fig. 7). We observed a close similarity in the temporal variability of bio/bb–$C_2$ with nss-$K^+$ and levoglucosan (Supplementary Fig. 8), especially during fall to winter, when open burning of post-harvest agricultural crop residues is widespread in rural China[44]. For all data from the whole campaign, bio/bb–$C_2$ concentration are positively correlated with levoglucosan ($r^2 = 0.67$, $P < 0.001$; Fig. 4a). The ratios of bio/bb–$C_2$ and nss-$K^+$ also show significant correlation ($r^2 = 0.58$, $P < 0.001$; Supplementary Fig. 9). In contrast, the coastal background samples record low concentrations of levoglucosan and nss-$K^+$ (Fig. 4a and Supplementary Fig. 9), signifying an insignificant contribution from biomass burning. Further, a strong linear correlation ($r^2 = 0.81$, $P < 0.001$, Fig. 4b) was found between bio/bb–$C_2$ concentrations and biogenic SOA tracers (i.e., isoprene tracers and α/β–pinene tracers) for the coastal background samples, suggesting that the bio/bb–$C_2$ in low biomass burning scenario was formed mainly from biogenic precursors. These independent lines of evidence are suggestive of and consistent with that background $C_2$ were biogenic, while bb–$C_2$ likely represent additional non-fossil $C_2$ concentrations. We therefore assume that, when there is little biomass burning contribution (levoglucosan ≈ 0), the concentration of bio/bb–$C_2$ (128 ng m$^{-3}$; Fig. 4a) would approximately represent the background $C_2$ level, that is, the bio–$C_2$. With this estimation, the total concentration of oxalic acid can be quantitatively distributed into three sources, i.e., fossil–$C_2$, bb–$C_2$ and bio–$C_2$.

The anthropogenic $C_2$ (including fossil–$C_2$ and bb–$C_2$) accounted for 78% of total $C_2$ in the continental outflow samples, whereas its contribution is 39% in the coastal background (Fig. 4d). Note that

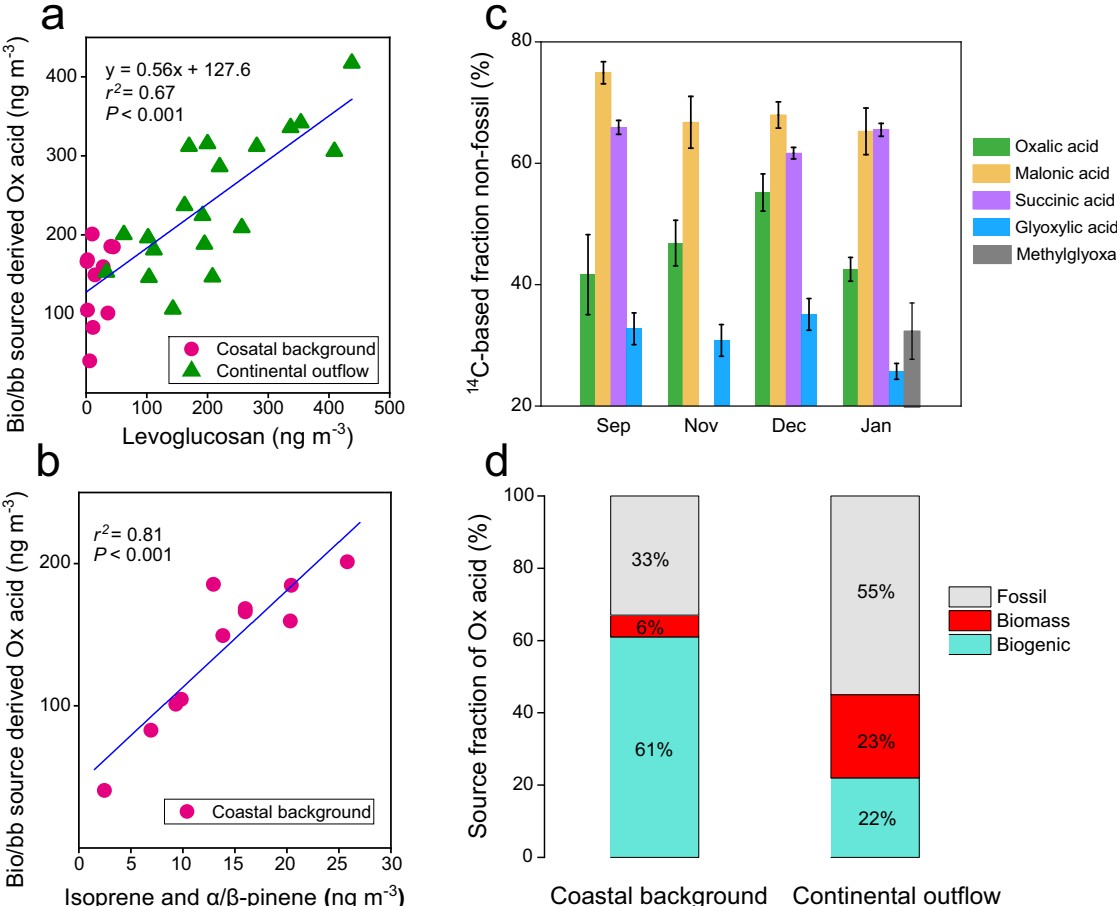

**Fig. 4 | Radiocarbon-based source apportionment of dicarboxylic acids and related compounds. a** Relationships between biogenic/biomass burning (bio/bb) source-derived oxalic acid vs. levoglucosan. **b** Relationships between bio/bb source-derived oxalic acid vs. biogenic secondary organic aerosol (SOA) tracer (include isoprene SOA tracers and α/β−pinene SOA tracers; Supplementary Method 1). **c** Monthly variability of non-fossil contribution to oxalic acid, malonic acid, succinic acid, glyoxylic acid, and methylglyoxal. The error bars represent ±1 standard deviation for total propagated uncertainty (Supplementary Method 3). **d** Relative contribution of biogenic emission, biomass burning, and fossil-fuel emission to oxalic acid in coastal background and continental outflow.

natural biomass burning (e.g., wildfires) was ignored since it contributed <4% of total biomass-burning-derived organic pollutants in China[45]. A tracer-based estimation suggests that anthropogenic gasSOA, which are formed by photochemical reaction in gas phase followed by thermodynamic partition to the condensed phase[4], account for 79% of the total gasSOA in the PRD during fall-winter[46]. This anthropogenic content of gasSOA is consistent with our [14]C-constrained source apportionment of aqSOA ($f_{anthropogenic}-C_2 = 78\%$). By extension, the contribution of anthropogenic emissions in aqSOA and gasSOA are likely to be consistent under ambient conditions with high ALW content. Our observation-based isotope constraints question the paradigm, largely based on model simulations, which instead have held that aqSOA, which may contribute as much mass as gasSOA to SOA budget, are largely formed from biogenic emissions[4]. Previous study considered that fossil precursors may be more hydrophobic than biogenic precursors, and thus are likely to form SOA preferentially through gas-phase processing[9]. However, the oxidation products of fossil precursors are largely water-soluble and thus their chemical reaction in the aqueous phase constitutes an unneglected pathway, which may be competitive to SOA formation via the gas-phase chemistry.

Figure 5 shows the dual effects of precursor sources and atmospheric processing of aqSOA by combining the [13]C and [14]C signatures of oxalic acid. Lower oxalic acid concentrations were found in the coastal background samples, with higher non-fossil contributions, and more enriched in [13]C. Higher oxalic acid concentrations in the continental outflow samples are characterized by higher fossil

contributions and more depleted in [13]C. The hygroscopic particles and precursors in continental outflow air masses facilitate the aqueous-phase chemical processes, which results in lighter δ[13]C values and higher fossil contribution to oxalic acid.

Conversely, such an aqSOA formation process became less important during the coastal background period, when there are less significant anthropogenic activities, and hence less precursors and hygroscopic particles in the atmosphere. In this case, the photochemical degradation (aging) of oxalic acid would surpass its aqueous-phase formation, which is evident in the increasing δ[13]C values of oxalic acid (Fig. 5). The interpretation may also apply to several outliers of the continental outflow samples (pink shadow in Fig. 5), of which the OC/EC ratios are much lower than the others, referring to exhaustive atmospheric aging.

### Effects of aqueous-phase processing on organic aerosols

Many studies have investigated the formation of aqSOA from biogenic/biomass precursors (e.g., isoprene)[3,7,8,22,47,48], while much less attention has been paid to fossil/anthropogenic precursors (e.g., aromatic hydrocarbons). Fossil precursors have been thought to consist mostly of hydrophobic components, and thus fossil-derived SOA have been expected to be less soluble in water[9]. Smog chamber experiments show that water-soluble organic aerosols exhibited more similarities to biogenic SOA than fossil SOA[49, 50]. In fact, low fossil contributions to WSOC (0−29%) have been reported in Europe, North America, and South Asia across different seasons[13−18], and [14]C analyses of aerosol OC

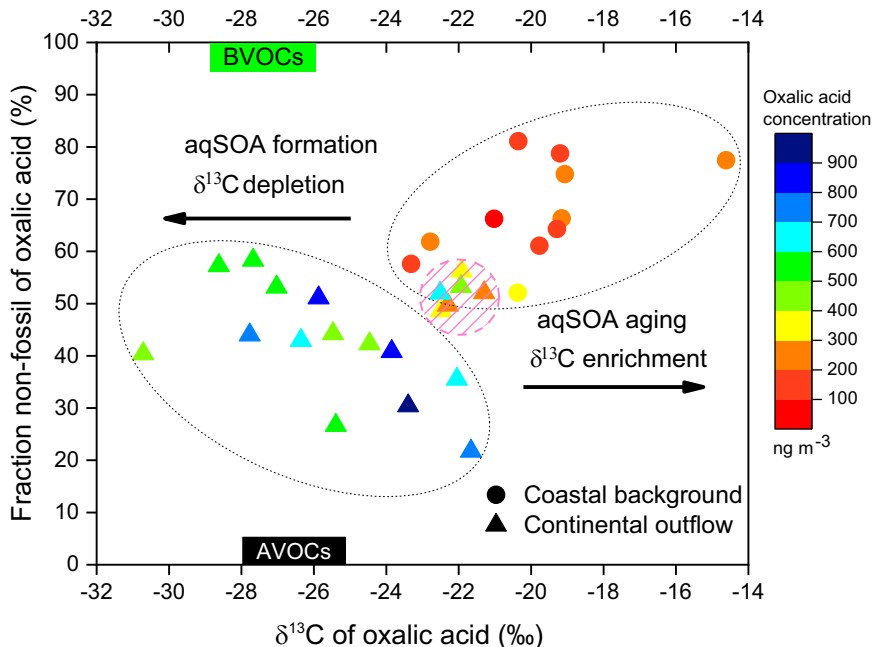

**Fig. 5 | Two-dimensional dual-carbon isotope characterization of individual oxalic acid for coastal background and continental outflow.** The color of the symbols represents the oxalic acid concentration. The $\delta^{13}C$ source signatures of oxalic acid precursor gases from biogenic volatile organic compounds (BVOCs; i.e., isoprene) and from anthropogenic volatile organic compounds (AVOCs; i.e., anthropogenic emitted non-methane hydrocarbons) are obtained from reported literature values (Supplementary Table 5). The two ovals include the coastal background samples and the majority of the continental outflow samples, respectively. The shadowed pink circle remarks the outliers of continental samples subjected to strong atmospheric aging.

and its sub-fractions have been interpreted such that the majority of fossil-derived OA was water-insoluble[51, 52]. Hence, these earlier field and laboratory studies show that water-soluble OA is only to very limited extent stemmed from fossil sources. In contrast, the results from the current study, using an extensive set of molecular $^{13}C/^{14}C$ data in China, strongly point to anthropogenic-fossil emissions as an important source of water-soluble OA. This is also supported by a large contribution of fossil sources to WSOC (30−60%) reported for both Chinese urban areas[52-55] and a receptor station in the southeast Yellow Sea intercepting a large regional footprint[11, 12]. Taken together, these two sets of findings imply a significant missing pathway responsible for the enhancement in fossil fuel-derived water-soluble OA in China.

The enhanced atmospheric aqueous-phase processing ($\delta^{13}C$ of oxalic acid is used as the proxy for aqSOA formation) simultaneously cause an increasing in the WSOC mass concentration, suggesting that aqSOA constitute a substantial water-soluble OA source (Fig. 3c). A recent study also showed that the oxidation of gas-phase components and subsequent uptake of water-soluble oxidation products triggered by the increase of ALW can explain the increase in SOA mass in Chinese megacities[32]. This is further corroborated by radiocarbon signatures of individual aqSOA tracers (i.e., $C_2$, $\omega C_2$, and MeGly), which reveals a dominant fossil source to aqSOA (Fig. 4c). The aqSOA even showed a fossil content (55−69%) comparable to the water-insoluble OC in other Chinese urban areas such as Beijing (62−68%) and Guangzhou (53−56%) in winter haze[56]. Our dual carbon isotope data of aqSOA tracers indicate that various fossil precursors may be oxidized to form aqSOA in the atmosphere and contribute substantially to the WSOC aerosols. We suggest that this atmospheric mechanism is responsible for the enhanced formation of fossil-derived WSOC in China.

To test the spatial extent of the findings, we retrieved aerosol samples from five emission hotspot megacities of China (Fig. 6a). The difference of $\delta^{13}C$ and $^{14}C$-derived $f_{bio/bb}$ values between oxalic acid and bulk WSOC were compared in winter and in summer, respectively (Fig. 6b). Although oxalic acid accounts for a large fraction of WSOC (~5.2%), probably the single most abundant compound, we observed

significant but opposite difference in the $\delta^{13}C$ and $\Delta^{14}C$ compositions between oxalic acid and WSOC during the two seasons. In winter, oxalic acid was more depleted in $^{13}C$ and $^{14}C$ than the WSOC pool in each of the cities, red-above-blue in Fig. 6b (refer to Supplementary Table 9). This agrees with the dominant role of aqueous-phase formation processes triggered by the increased ALW in winter[32]. As mentioned above, overwhelming fossil contributions to aqSOA compounds were observed, that were consistent with or even higher than the fossil content in water-insoluble OA reported in Chinese urban areas during winter haze[52, 56]. Substantial fossil-derived precursors are likely oxidized to WSOC aerosol through secondary aqueous processing in winter, resulting in products (such as oxalic acid) with more negative $\delta^{13}C$ values and more fossil contributions than the bulk WSOC pool.

On the contrary, in summer, oxalic acid was more enriched in $^{13}C$ and $^{14}C$ than the WSOC pool in Guangzhou, Wuhan and Shanghai. As shown in Fig. 6b, a general trend of blue-above-red is displayed, despite the very close values in Beijing and Chengdu which are within the analytical uncertainties. A probable interpretation is that biomass/biogenic component from the WSOC pool is preferentially subjected to oxidative aging to small molecules[12, 57], which results in oxalic acid enriched both in $\delta^{13}C$ and $\Delta^{14}C$. The differing isotopic signatures of different emission sources would also contribute to the enrichment of carbon isotopes of oxalic acid. Note here that biogenic contributions to oxalic acid can be expected to increase during summer due to it being the growing season.

**Atmospheric implications**

The large contribution of fossil-derived carbon to aqSOA highlighted in this study has important implications for aerosol climate forcing and regional air quality. Aqueous-phase processing of fossil precursors, competitive to gas-phase pathway, could amplify the impact of anthropogenic emissions on water-soluble OA, alter the chemical property and hygroscopicity of aerosols. The derived hygroscopic SOA affect not only direct radiative forcing, but also cloud condensation

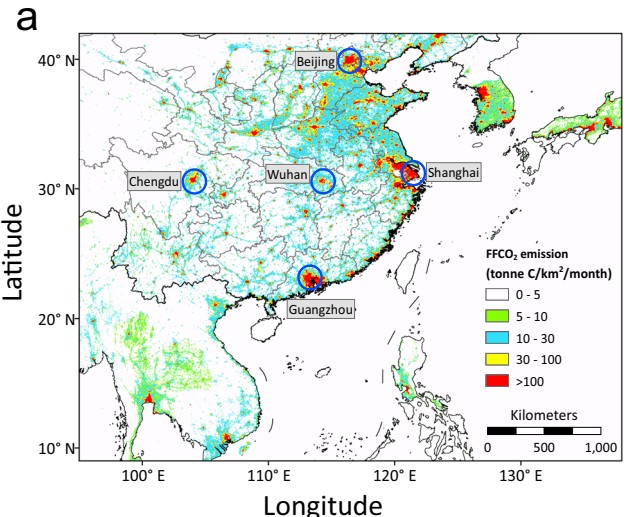
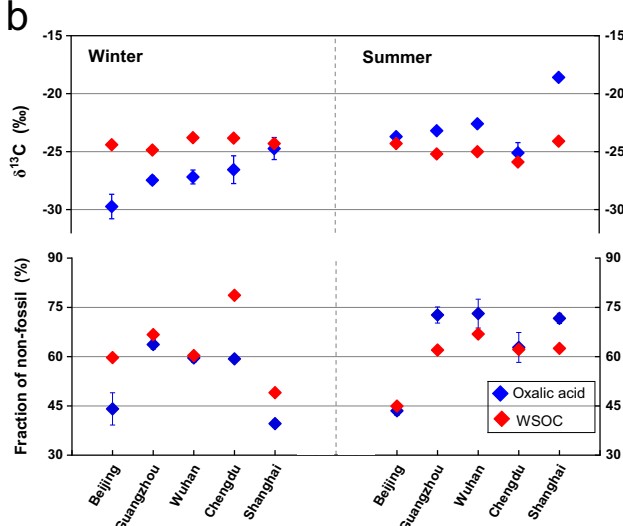

**Fig. 6 | The δ¹³C composition and ¹⁴C-based source apportionment of water-soluble organic carbon (WSOC) and oxalic acid in PM$_{2.5}$ collected from five emission hotspot megacities of China. a** Locations of the five megacities (Beijing, Guangzhou, Wuhan, Chengdu and Shanghai) and average fossil fuel CO$_2$ (FF CO$_2$) emissions during the year 2018. FF CO$_2$ emissions indicating high levels of anthropogenic activities over the five megacities. The FF CO$_2$ emission data is obtained from Open-source Data Inventory for Anthropogenic CO$_2$ (https://db.cger.nies.go.jp/dataset/ODIAC/DL_odiac2020b.html). The coastline boundaries in the map are originated from Natural Earth free vector map data (https://www.

naturalearthdata.com/). The administration boundaries in the map are originated from map products of National Geomatics Center of China (https://www.webmap.cn/). **b** The difference of δ¹³C and ¹⁴C-based fraction of non-fossil sources between oxalic acid and WSOC in winter (January 2018) and in summer (July 2018), respectively. The error bars for δ¹³C composition represent the relative standard deviation for repeat analyses (n = 3). The error bars for ¹⁴C-based fraction of non-fossil sources represent ±1 standard deviation for total propagated uncertainty (Supplementary Method 3).

nuclei (CCN) activity, and the subsequent cloud-mediated effect of anthropogenic aerosols on the climate system[58]. As for oxalic acid, its interactions with inorganic salts in the aqueous phase may modify the CCN activity of aerosol particle[59–61].

A mutual promotion effect may take place between ALW and inorganic particles (e.g., sulfate and nitrate)[31, 62]. Our year-round observation at the Heshan receptor site witnesses a positive feedback loop among ALW, inorganics, as well as aqSOA compounds. The dataset supports that the inorganic particles induce ALW growth (Supplementary Fig. 5), facilitating the partitioning of gas-phase oxidation products into aqueous-phase medium, and promoting the formation of aqSOA (Fig. 3b)[6, 30]. The formation of aqSOA in the organic aerosol fraction would decrease aerosol viscosity and increase ALW[63], as in turn promotes further gas-to-liquid transfer of water-soluble organic precursors[23, 32]. Oxalic acid is identified as the upper bound of the hygroscopic properties of OA, with a *k*-Kohler of 0.48[7]. Using the Zdanovskii−Stokes−Robinson (ZSR) mixing model[64], the effect of anthropogenic oxalic acid on aerosol water accounts for 3−38% (average: 10%) of the ALW contributed by organic compounds (Supplementary Method 2). The inorganic particles can be considered anthropogenic in most continental atmosphere[3, 30, 39]. We provide ¹⁴C-based evidence that, besides inorganic species, substantial aqSOA compounds are also derived from anthropogenic precursors. Therefore, a broad control of various anthropogenic emissions such as SO$_2$, NO$_x$, and VOC precursors is vital for reducing the organic particulate pollution.

Nitrate is the dominant atmospheric hygroscopic particles in China[30, 31, 41]. A model study predicted that 92% of increased ALW in eastern China by the year 2100 can be attributed to the increase in anthropogenic nitrate aerosol[30]. Here, the average nitrate concentration in the continental outflow samples was nearly 20 times higher than coastal background (Supplementary Table 2). Thus, this nitrate enhancement in the particulate phase could be the dominant factor that facilitated the increase of ALW in the continental outflow regime. Furthermore, there has been a rapid translation from coal combustion

to natural gas in China. However, natural gas combustion can produce more than three times as much water vapor as coal burning[65]. The combustion-derived water was interpreted to constitute 6.2% of the atmospheric moisture and added 5.1% of the anthropogenic PM$_{2.5}$ in northwestern China[65]. Therefore, the enhancing effects of particulate nitrate and combustion-derived water vapor on aqSOA formation should be addressed when changing energy structure in future climate and air quality scenarios.

The present findings highlight the relevance of aqueous-phase chemistry for processing of fossil-fuel emissions over China. These scenarios are not unique to East Asia and could be relevant in other high fossil-fuel consumption regions with humid weather. In a study in northern Georgia of the US, Weber et al.[13] observed strong correlations between WSOC and anthropogenic precursors, while in contrast the radiocarbon analysis revealed dominant biogenic contributions to WSOC. Our results suggest that aqSOA do not necessarily share the same source patterns with bulk WSOC. WSOC have a large primary contribution from biomass burning[66], which would mask the fossil content in aqSOA. Therefore, the fossil contributions to aqSOA would be much higher than expected in locations such as North America, Europe, and South Asia where the aqueous-phase chemistry pathway has been demonstrated significant[7, 39, 67]. Even in pristine environments where biogenic emissions are dominant, the importance of anthropogenic emissions to aqSOA formation would also be significant since the ALW are largely anthropogenic[2, 39].

In this study, the δ¹³C differences of oxalic acid was dominantly attributed to kinetic isotope effects in atmospheric processing. It should be noted that the differences in the isotopic signatures of emission sources may also contribute to the isotopic variations of oxalic acid. However, these factors are not quantitively constrained. The extent to which the atmospheric processing is representative for the differences in isotopic signatures remains to be investigated in more details. We also stress that apart from aqueous-phase processes, both biomass burning and gas-phase photochemical aging could also be sources for oxalic acid. To differentiating these pathways in field

measurements is still challenging. Therefore, cautious interpretations of oxalic acid isotopic signals are needed where there are intensive biomass burning, extensive atmospheric aging, and seasonality of C3/C4 vegetation changes. Future dual-carbon isotope studies on other aqSOA constituents (e.g., glyoxal) are strongly warranted to better constrain the chemical mechanism responsible for fossil-derived aqSOA formation.

Overall, compound-specific dual carbon isotopic evidences from field observations show that fossil anthropogenic precursors contribute to the formation of aqSOA to a much higher extent than expected. Atmospheric aqueous-phase processes are expected to increase due to the enhanced evapotranspiration in a warmer world[64]. An absence of accounting for such aqueous processing incurred by fossil precursors could lead to an underestimation of the anthropogenic contribution to organic aerosols. Understanding the role of anthropogenic emissions in aqSOA formation and the chemical mechanisms involved are of critical concern for future climate and air quality projections, and within the context of energy needs and choices.

## Methods

### Aerosol sampling
The regional receptor site was positioned in Heshan Atmospheric Environmental Monitoring Superstation (22.711° N, 112.927° E, 60 m asl), a rural site located at 50 km southwest to the megacity of Guangzhou in the central PRD. The surrounding area of the sampling site is dominated by farmlands and forests and is far from local anthropogenic emissions[68]. During the winter northeast-monsoon season, this specific site well intercepts high anthropogenically dominated outflow airmass from Chinese continental. The strength of anthropogenic impact decreases in the summer southeast/southwest monsoon, with the arrival of clean air mass originating from the Western Pacific/Indian Ocean. Field experiments started in June 2017 and ended in May 2018. $PM_{2.5}$ aerosol samples ($n = 32$) were collected onto pre-combusted quartz-fiber filters by using a high-volume sampler at an air-flow rate of $1\,m^3$ /min for 48 h.

$PM_{2.5}$ samples were additionally studied in five emission hot spot megacities of China (Beijing, Shanghai, Guangzhou, Chengdu, Wuhan), which were collected continuously every 24 h for 1 week in winter (January 2018) and in summer (July 2018). Continuous sets of 7 filters collected in one week were combined to represent the seasonal characteristic of each urban site. The sample filters were stored in darkness at −20 °C prior to analysis. Field blank filters were collected for each campaign.

### Backward air mass trajectories
Data of atmospheric concentration of meteorological parameters and gaseous pollutants were obtained from the hourly monitor by Heshan Atmospheric Environmental Monitoring Superstation. To characterize different geographical sources of air masses encountered at Heshan, 3-day isentropic backward air mass trajectories were computed at an elevation of 100 m for every 12 h using the HYSPLIT model (version-4) and meteorological datasets from the NOAA air resources laboratory (http://ready.arl.noaa.gov/HYSPLIT.php). The back trajectories are grouped into two major transport pathways with vastly different source regions during the sampling campaigns.

### Chemical analysis
The concentration of typical aqSOA species (diacids, oxoacids, and α-dicarbonyls) was measured following earlier protocols[19, 27]. An aliquot of filter samples was extracted with Milli-Q water. The extracts were concentrated to dryness and derivatized with 10% $BF_3$ in 1-butanol (Sigma-Aldrich, St Louis, MO) in a hot water bath at 100 °C for about 1 h. The derivatives were extracted with n-hexane and finally quantified using GC-MS (Agilent 7890 A gas chromatography and Agilent 5975C

mass spectrometry). The concentrations reported here are corrected for field blank, but are not corrected for recovery efficiencies.

Relevant details of the analysis method for aerosol carbonaceous components (organic carbon: OC, elemental carbon: EC; water-soluble organic carbon: WSOC), water-soluble inorganic constituents (WSIC, i.e., $Na^+$, $NH_4^+$, $K^+$, $Mg^{2+}$, $Ca^{2+}$, $Cl^-$, $NO_3^-$, and $SO_4^{2-}$), biogenic SOA tracers; and sugar compounds are provided in the Supplementary Method 1.

### Aerosol liquid water estimation
The total aerosol liquid water (ALW) is taken as the sum of water associated with individual aerosol chemical components, with the assumption that the particles are internally mixed[69]. ALW from inorganic species was estimated using a thermodynamic equilibrium model, ISORROPIA-II; http://nenes.eas.gatech.edu/ISORROPIA/ (Supplementary Method 2) using particle concentration of inorganic ions and meteorological parameters (temperature and RH%). The organic contribution for ALW was calculated by Zdanovskii–Stokes–Robinson (ZSR) mixing rule as discussed in Supplementary Method 2.

### Compound-specific $\delta^{13}C$ and $\Delta^{14}C$ analysis
The $\delta^{13}C$ values of diacids derivatives were determined using GC ISO-Link2/IRMS (Thermo Fisher Scientific). The $\delta^{13}C$ values of free diacids and related compounds were then calculated by an isotopic mass balance approach, based on the measured $\delta^{13}C$ values of derivatives and the derivatizing agent (1-butanol, $\delta^{13}C = −30.21‰$). The isotopic fractionation during the derivatization step has been reported to be less than 0.73‰ in our earlier publications[19]. Each sample was measured in duplicate and the mean $\delta^{13}C$ values were reported. The difference in $\delta^{13}C$ of diacids (i.e., $C_2$, $C_3$, $C_4$, $C_5$, $C_6$, Ph, tPh, $C_9$) in replicate analyses was generally <1‰, though the analytical accuracy was < 2‰ for other compounds (i.e., Pyr, $\omega C_2$).

The analytical method for the isolation of individual diacids to determine $\Delta^{14}C$ compositions builds on our previously publications[19]. In brief, the filter was extracted with the Milli-Q water and derivatized with $BF_3$/1-butanol. Microgram quantities of $C_2$, $C_3$, $C_4$, $\omega C_2$, and MeGly were isolated and collected by preparative capillary gas chromatography (pcGC) through around 50 consecutive runs (5 µl per injection), which was sufficient for off-line natural abundance radiocarbon analysis. The pcGC isolates were combusted at 920 °C and the resulting $CO_2$ was finally converted to graphite via hydrogen reduction method. Measurements of $^{14}C$ were performed using Accelerator Mass Spectrometry (1.5SDH-1, 0.5MV, NEC, USA)[19]. In all cases, $^{14}C$ results are reported as "fraction of modern carbon" ($F_m$) normalized to a common $\delta^{13}C$ value of −25‰. The $F_m$ values were further converted into the fraction of non-fossil source ($f_{bio/bb}$) with a conversion factor of 1.06 to compensate for the excess $^{14}C$ produced by nuclear bomb testing in the 1950s–1960s. The concentration of diacids in blank filters was <0.06% of the real samples, therefore, no filter blank subtraction was performed for the isotope analysis in this study. Sample results were corrected for procedural blanks using a pair of processing standards with modern or fossil radiocarbon composition. An isotopic mass-balance approach was adopted to correct the carbon contribution of butanol groups (1-butanol, $F_m = 0.0029 ± 0.001$) added in the derivatives of diacids, where appropriate. Calculations for blank correction, $F_m$ results and error propagation are shown in Supplementary Method 3.

### $\delta^{13}C$ and $\Delta^{14}C$ analysis of WSOC
The analytical method for the measurements of $\delta^{13}C$ and $\Delta^{14}C$ composition of WSOC builds on ultrasonication extraction protocols were reported in Supplementary Method 4. The validity of the ultrasonication method during extraction of water-soluble organic materials was demonstrated by comparing ultrasonication with another extraction protocol, i.e., soaking (see Supplementary Table 10 and

Supplementary Fig. 10). Measurements of the $\delta^{13}C$ composition of WSOC were performed on a Flash 2000 elemental analyzer connected to a Thermo Scientific Delta V isotope ratio mass spectrometer. Measurements of the $\Delta^{14}C$ composition were performed using accelerator mass spectrometry facility (1.5SDH-1, 0.5MV, NEC, USA) of the Guangzhou Institute of Geochemistry of the Chinese Academy of Sciences (GIGCAS).

## Data availability

The data supporting the findings of this study are available at the Figshare digital repository (https://doi.org/10.6084/m9.figshare.20469540). The aerosol optical depth data are available through the Moderate Resolution Imaging Spectroradiometer (MODIS) observations (https://giovanni.gsfc.nasa.gov/giovanni/). The isentropic backward air mass trajectories are available through the NOAA air resources laboratory (http://ready.arl.noaa.gov/HYSPLIT.php). The fossil fuel $CO_2$ emission data were accessed through the Open-source Data Inventory for Anthropogenic $CO_2$ (https://db.cger.nies.go.jp/dataset/ODIAC/DL_odiac2020b.html). Source data are provided with this paper.

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

## Acknowledgements

This study was funded by the Natural Science Foundation of China (42030715), the Alliance of International Science Organizations research

and cooperative projects (ANSO-CR-KP–2021-05), the Guangdong Foundation for Program of Science and Technology Research (2017BT01Z134), and the National Key R&D Program of China (2017YFC0212000). We would like to thank Dr Duohong Chen for providing the meteorological data, Dr Weiwei Hu for helpful discussions, and Dr Lili Ming for support with instruments.

## Author contributions

B.X. and G.Z. designed the experiment. JUN.L., Z.C., and S.-Z.Z. provided the samples. B.X., A.A., B.K., A.P., JING.L., C.H., Z.C. and S.-Y.Z. processed data and performed analyses. B.X. wrote the paper. G.Z., Ö.G., K.K., A.A., S.B., G.-C.Z., P.P., and G.S. commented on the manuscript.

## Competing interests

The authors declare no competing interests.
