## [Peer Review File · Nature Communications]

Large contribution of fossil-derived components to aqueous secondary organic aerosols in ChinaEditorial Note: Parts of this Peer Review File have been redacted as indicated to remove third-party material where no permission to publish could be obtained.

REVIEWER COMMENTS

Reviewer #1 (Remarks to the Author):

In this study, oxalic acid is treated as the dominant aqueous SOA (aqSOA) molecule, and its dual-carbon isotopic fingerprints (^{13}C and ^{14}C) were measured and used to track the precursor sources and formation mechanisms of aqSOA. My major concerns are the rationality and novelty, which hinder the publication of this work in Nature Communications. As for the rationality of oxalic acid as an aqueous-phase tracer: Oxalic acid has been reported to be emitted directly from motor exhaust, biogenic sources, and biomass burning, and/or formed in the atmosphere through secondary processes. The secondary formation routes for oxalic acid may involve photochemical formation followed by partitioning onto the condensed phase and heterogeneous formation which includes aqueous-phase processes (e.g., in-cloud processing) and condensed phase processes. Therefore, it is questionable to use oxalic acid to track the aqueous-phase secondary organic aerosol formation.

My second concern is about the novelty. $\delta^{13}\text{C}$ of oxalic acid and other dicarboxylic acids has been measured in many previous studies (including studies conducted at receptor background site like this study; Zhang et al., 2016, doi:10.1002/2015JD024081) to explore the formation pathway. It has been concluded that different formation pathway of oxalic acid leads to different ^{13}C signatures, and this fact can be used to track the formation pathway of oxalic acid.

The ^{14}C measurement of oxalic acid is an interesting part, but the details of data interpretation, e.g., how to discriminate from different potential fossil sources, are not fully discussed and sometimes are over-interpreted, leading to questionable conclusions. For example, ^{14}C measurements show that $55 \pm 10\%$ oxalic acid was contributed by fossil sources for continental outflow samples. Using source tracers (i.e., NO_3^- and tPh), the authors further qualitatively identify “both traffic and plastic burning emissions were important contributors to the fossil source-derived oxalic acid”. However, the data were overinterpreted:

Line 242–244: “The high ratios of NO_3^- in atmosphere particles ($\sim 10\%$)” probably suggests that the traffic emissions contribute significantly to atmospheric particles, but from this, we cannot draw any firm conclusion about the relation between traffic emissions and the formation of oxalic acid, thus “a significant role of traffic emission-derived precursors in the formation of oxalic acid” is an overinterpretation and should be revised.

Line 245–248: How did the authors conclude “the importance of precursors from plastic burning to the formation of fossil oxalic acid” from the high contribution of tPh to total diacids (5%), given that tPh is a combustion tracer of plastic polymers? Again, it is an overinterpretation. The high contribution of tPh to total diacids is likely pointing to the importance of plastic burning to total acids. However, one cannot conclude any relation between plastic burning and the formation of fossil oxalic acid.

Therefore, the conclusion that “both traffic and plastic burning emissions were important contributors to the fossil source-derived oxalic acid” is questionable and should be re-formulated.

Line 167–169: “potential precursors of oxalic acid would then be anthropogenic VOCs (AVOCs; e.g., aromatic hydrocarbons: $-27.7 \pm 1.7\text{‰}$ ³⁷) and C₃-plant-derived biogenic VOCs (BVOCs; e.g., isoprene: $-27 \pm 2\text{‰}$ ³⁸), of which the $\delta^{13}\text{C}$ values are also similar”. In the Ref. 38, the $\delta^{13}\text{C}$ values of $-27.7 \pm 2.0\text{‰}$ is for isoprene emitted from Velvet Bean (*Mucana pruriens* L. var. *utilis*). Does the specific “Velvet Bean” represent the biogenic source of isoprene in this study? In other words, is the ^{13}C signature of isoprene in this study different from $-27 \pm 2\text{‰}$ (i.e., $\delta^{13}\text{C}$ value of isoprene from Velvet Bean), so that different from $-27.7 \pm 1.7\text{‰}$ (i.e., $\delta^{13}\text{C}$ value for AVOCs)? I am concern about this point, because this is closely related to the argument that “the isotopic range of various sources of oxalic acid is minimal, leaving kinetic isotopic fractionation the most probable processes controlling the $\delta^{13}\text{C}$ of oxalic acid” (Line 170–172).

Line 173–180: Somewhere in the manuscript, to facilitate the reader’s understanding, it is good to explain how the $\delta^{13}\text{C}$ values change during the two formation pathways of oxalic acid.

Line 239–242: In Supplementary Fig.5, the data points are very scatter, and it looks like that the correlation between fossil-C₂ and C₂ concentrations for either continental outflow or coastal background is weak.

Line 380–381: Inconsistence between the text and figure. Text says “Oxalic acid is more enriched in ^{14}C than WSOC in summer”, but Fig.6b show the opposite.

Line 428 : year 2100?

Line 524: If using the NBS Oxalic Acid I standard ($\delta^{13}\text{C}$ value close to -19‰), the ^{14}C is measured and normalized to $\delta^{13}\text{C} = -19\text{‰}$ (not -25‰ as stated in the manuscript). Almost certainly the authors used Oxalic Acid II, because Oxalic Acid I is no longer commercially available. Using Oxalic Acid II, the ^{14}C is normalized to -25‰

Supplemental line 120–122: “ $F_{m,ref} = 1.10$ ” for wood burning. Please clarify assumptions on wood age and fell date.

Reviewer #2 (Remarks to the Author):

With “Large contribution of fossil-derived components to aqueous secondary organic aerosols in China,” Xu et al. have presented a large dataset of aerosol organic compositional measurements for samples with differing air mass histories. The data appear to show evidence for the aqueous phase production of secondary organic aerosols (SOA) from fossil-derived carbon. Many previous studies have demonstrated aqueous SOA formation and the influence of fossil-derived inorganic species, but radiocarbon measurements did not suggest significant contributions from fossil C. Rather, the previous work showed the C to be derived from natural biomass – so the findings here of very high fossil C contributions to both water soluble organic carbon and oxalic acid (an aqueous SOA tracer) are novel. The paper is well-reasoned and well-written. I do, however, have a potential concern about the use of ultrasonication to produce the water soluble aerosol extracts. This concern is elaborated upon below; the major concern is that ultrasonication may have produced compositional artifacts that render the results non-representative of ambient aerosol composition. In addition to this concern, I outline additional comments and suggestions in the comments that follow. Should the methodological concerns be allayed, this work would effectively demonstrate previously unrecognized inputs of fossil fuel-derived C to the secondary organic aerosol pool and have significant impact on atmospheric processes and climate.

Ultrasonication: In the Supplementary Text 4 section, ultrasonication is used to produce water extracts for isotopic analyses. Ultrasonication is known to produce reactive oxygen species (e.g., hydroxyl radical, superoxide radical, singlet oxygen) and organic radicals and is not recommended for extractions intended to characterize environmentally-relevant organic compositions. (See e.g., Miljevic et al. 2014, *Aerosol Sci. Tech.*, 48:1276-1284) These free radical species, once produced, are well known to react with organic molecules and alter organic molecular composition. Such organic transformation reactions can solubilize previously insoluble organics and produce small molecules such as oxalic acid. In fact, radical species are major players in the photochemical transformation of atmospheric species in the gaseous, aqueous, and particle phases. The use of ultrasonication, therefore, has the high potential for producing artifacts by reacting ambient aerosol organics (effectively driving the aqueous reactions the authors are suggesting occur naturally) and altering the fraction of WSOC as well as the aerosol molecular composition. Thus, a major question that I have is: were all water extractions conducted using ultrasonication? If so, do the authors have evidence that the technique does not alter the molecular composition of aerosol organics and specifically the quantities and isotopic composition of oxalic acid and WSOC that are the focus of this work? If ultrasonication was not used, please clarify the water extraction technique employed.

Additional Comments:

- the Results section includes considerable Discussion and is better headed as a Results and Discussion section. The Discussion section reads more like an “Implications for Atmospheric Cycling of Organics...” section.

- Line 145, change “typically” to “often”

-Line 273-5 – the authors note that “oxalic acid is mainly produced from fossil fuel-derived precursors...” But supplementary Figure 7 shows biological sources to be substantial during all sampling times – sometimes being higher than fossil fuel sources. Please reword this sentence to better characterize the data.

-Line 298 – the calculation $\text{fossil-C2} + \text{bb-C2} = \text{anthropogenic}$ assumes that all biomass burning is human-derived. It must be acknowledged that natural wildfires are a potential source of bb-C2.

- Lines 367-397 – this section represents a nice experiment demonstrating the spatial extent of the fossil C contributions; however, the data interpretation seems to be a bit lacking. The isotopic differences discussed in lines 372 to 382 are attributed to atmospheric aging which is one possibility due to the higher amounts of sunlight in summer. However, there are other potential reasons including different air mass back trajectories (the Heshan dataset showed these to vary drastically between summer and winter) and the fact that natural biomass inputs can be expected to increase during summer due to it being the growing season.

-Line 371 – change “minor” to “no”

- Lines 386-389 – the increase of ALW in winter is said to be important for oxalic acid depletions in winter, but no seasonal ALW data are presented. Supplementary table 2 suggests the data may be available. Referencing the ALW data to support the argument would be valuable.

- Lines 389-391 – fossil content in water insoluble OA is referenced but no data have been presented for water insoluble OA. If it exists, it should be shown to back up the point being made.

- Lines 412-439 – the authors present a positive feedback loop among hygroscopic particles, ALW, and aqSOA formation that does not connect exceptionally well to the Results presented already. A clearer reference to how the data gathered in this paper point to the existence of the feedback loop is needed. In lines 426-432, there is a discussion of how nitrate represents the hygroscopic particles of interest, but this is the first mention of nitrate data in the paper. If hygroscopicity is going to factor into the conclusions as it does here, nitrate and hygroscopicity should be presented much earlier.

- A discussion of how relevant these results are to the rest of the world is warranted. The study site receives air from some of the highest fossil fuel emission areas on the globe. Are there other locations of interest?

Tables and Figures:

- In Supplementary Figures 2 and 3, the air mass back trajectories are characterized differently than they are in the text. Please change for consistency.

- In Figures 2, 4, and 6, the fraction of non-fossil C is plotted as increasing on the y-axis, while in Figure 5 and Supplemental figures 5 and 6 the fraction of fossil C is plotted as increasing on the y-axis. This is confusing. I suggest using the same convention for all plots quantifying fossil or non-fossil C. My own

preference is to have the fraction of fossil C increase on the y-axis since the paper's focus is demonstrating fossil C contributions to aqSOA.

- In Supplementary Table 2, the WSON values are suspiciously high (higher than WSOC in the coastal background samples). The description in Supplementary Text 1 describes the measurement of total dissolved nitrogen (which would include inorganic species). Typically, WSON is calculated as follows: $WSON = TDN - (\text{nitrate} + \text{nitrite} + \text{ammonium})$. Is this what was done?

Reviewer #3 (Remarks to the Author):

In this manuscript, the authors describe chemical characterization and mass measurements of secondary organic aerosol formed via aqueous pathways (aqSOA) during two contrasting phases of the East Asia monsoon system. A novel and important attribute of this work is the simultaneous measurement of both $\delta^{13}\text{C}$ - $\Delta^{14}\text{C}$ isotopes, which provides insight to the parent organic gas and formation process. The authors find compelling evidence that fossil carbon contributes substantially to water-soluble SOA mass. This work contributes scientifically by providing insight to atmospheric transport and processing of trace species, in addition to the policy-relevant finding demonstrating the contribution of fossil carbon to aqSOA.

The contribution of fossil carbon to aqSOA is largest in continental air masses, and not in the marine derived air. This is clear in the presented graphs and is consistent

This work is noteworthy. The last section of the submitted manuscript is not constructed as carefully as other parts. I think this work should be published, but some revision is required. I list detailed comments below.

I largely buy the authors' argument that kinetic isotope effects (KIE) dominate the $\delta^{13}\text{C}$ composition of oxalic acid – but think a comment on what the possible impacts on interpretation are given limitations of this assumption is needed.

Line 141: "With the addition of aqSOA formation, the higher WSOC/OC ratios in the continental outflow samples may thus be explained." This sentence is overstated.

The evidence is consistent but not incontrovertible truth. I suggest rephrasing to something along the lines of:

Observation in continental outflow of higher GLY/MeGLY and higher [WSOC]/[OC] is consistent with aqSOA formation processes.

Line 203: The authors state "...rather than photochemical breakdown of longer-chain diacids." I think the point that aqueous-phase processing is much more like than gas-phase photochemistry is an important point. I think the authors should amend the sentence so that it "... reads rather than gas-phase photochemical ..."

Line 291: "These evidences confirm" Is overstated. Independent lines of evidence are suggestive of and consistent with background C2"

Line 312: The authors state:" However, the oxidation products of fossil precursors are largely water-soluble" I agree that that a large fraction of atmospheric organic gases are water-soluble. Perhaps to help drive home this point, the sequence on line 203 could be altered to list both SVOC and WSOC (water-soluble organic carbon).

The Figure 5 caption should include concise explanation for ovals and circle.

Starting at line 343: "...water-soluble OA is only to very limited extent stemming from fossil sources." The phrasing is awkward and should be re-worded.

Starting at Line 382: "This is probably due to that biomass/biogenic component from the WSOC pool is preferentially subjected to 383 oxidative aging to small molecules". This sentence is awkward and needs a little work.

Starting at Line 403: "Condensation of oxalic acid and its salts in the aqueous phase may significantly impact the CCN activity of aerosol particle." The word "condensation" in this sentence seems at odds with the authors' main premise – that oxalic acid often forms in the aqueous-phase. The sentence reads as if it describing the importance of the gas-phase formation pathway. Perhaps "condensation" could be replaced with "presence".

Editorial:

Line 42: “but” would read better as “and”

Line 119: “are” should be “is”

Line 120: “indicate” should be “indicates”

Line 187: remove “The” in front of oxalic acid

Line 400: “amplifies” should be “amplify”

Line 421: “provides” should be “provide”

Line 447: “accounting such” there should another word in between, perhaps “for”

The authors provide a link for the ISORROPIA model, but not the HYSPLIT model. Why?

Author responses to reviews and edits of *Nature Communications* manuscript

Title: “Large contribution of fossil-derived components to aqueous secondary organic aerosols in China (Manuscript ID: NCOMMS-21-48555A)”

We gratefully thank the reviewers for constructive comments that have clearly contributed to improve the manuscript during revision. We are encouraged by the reviewers’ recognition of the overall importance of “unrecognized inputs of fossil fuel-derived carbon to the secondary organic aerosol pool”. Nevertheless, we also acknowledge issues raised by reviewers about insufficient data interpretations/discussions.

Guided by many well-informed and constructive comments, we have thoroughly revised the entire manuscript, including now (i) a detailed assessment on the integrity of the well-established extraction method; and (ii) a more in-depth discussion in the last section (**Atmospheric implications**). For your and reviewer’s easiness to review the manuscript, an annotated manuscript is attached, in which all changes are highlighted in yellow.

All reviewer comments and our responses are listed below, organized such that each reviewer comment is shown first in *italics black font*, followed by our detailed response in upright black font. Our response refers to line numbers in the revised manuscript version.

We are in debt with the anonymous reviewers for their pertinent comments and suggestions, which have been of great help to the improvement of this manuscript.

We thank you very much for kindly handling and reviewing our manuscript.

Best regards on behalf of all authors,

Gan Zhang and Örjan Gustafsson

Responses to reviewer #1:

(1). *In this study, oxalic acid is treated as the dominant aqueous SOA (aqSOA) molecule, and its dual-carbon isotopic fingerprints (^{13}C and ^{14}C) were measured and used to track the precursor sources and formation mechanisms of aqSOA. My major concerns are the rationality and novelty, which hinder the publication of this work in Nature Communications. As for the rationality of oxalic acid as an aqueous-phase tracer: Oxalic acid has been reported to be emitted directly from motor exhaust, biogenic sources, and biomass burning, and/or formed in the atmosphere through secondary processes. The secondary formation routes for oxalic acid may involve photochemical formation followed by partitioning onto the condensed phase and heterogeneous formation which includes aqueous-phase processes (e.g., in-cloud processing) and condensed phase processes. Therefore, it is questionable to use oxalic acid to track the aqueous-phase secondary organic aerosol formation.*

Responses: Many thanks for your comments. We agree that oxalic acid can be generated from primary sources, such as fossil-fuel combustion and biomass burning. In this study, there are a series of evidences supported our hypothesis that oxalic acid can well track the aqueous-phase chemistry processes.

Firstly, the concentration of oxalic acid from primary sources would be very minor compared to the atmospheric oxalic acid pool. Although *Kawamura and Kaplan (1987)* suggest the importance of the contribution from auto exhaust to diacids in Los Angeles aerosols, this work may have historical limitations because motor technologies have been rapidly developing in the past decades. *Huang and Yu (2007)* suggest that vehicular emissions are not a significant primary source for oxalic acid because no enhancement in oxalate levels in the tunnel aerosol was seen in comparison with the oxalate levels in the ambient environment. We further analyzed the concentration of oxalic acid from smoke particles of biomass burning and coal combustion (sample characteristic was available in *Tang et al., 2020*). The resulted oxalic acid concentration in source emission aerosols (biomass burning: 157 ng m^{-3} ; coal burning: 78 ng m^{-3}) were significantly lower than the ambient oxalic acid level reported in this study (average: 446 ng m^{-3}). Moreover, the surrounding area of the sampling site (Heshan station) is dominated by farmlands and forests and is far from any strong anthropogenic

emissions (Peng *et al.*, 2014; Yuan *et al.*, 2016). The air mass in Heshan station is dominantly originated from either continental China or South China Sea during air transport. Therefore, the primary source of oxalic acid is expected negligible compared to its secondary sources.

Secondary, a large number of publications has demonstrated that oxalic acid is predominantly from aqueous processing. The first evidence on the aqueous-phase SOA formation was published in the early 2000's by Warneck [2003], who observed abundant oxalic acid formation in marine clouds. After that, an increasing number of laboratory and field evidences (e.g., Crahan *et al.*, 2004; Ervens *et al.*, 2004; Warneck, 2005; Lim *et al.*, 2005; Carlton *et al.*, 2006, 2007; Sorooshian *et al.*, 2006, 2007; Zhao *et al.*, 2019) were presented for aqueous-phase oxalic acid production in cloud droplets and aqueous particles as the dominant mechanism for oxalic acid formation in ambient aerosols. In aerosol-phase, oxalic acid correlates well with aerosol liquid water content (ALWC) and hygroscopic particles (such as sulfate and nitrate), also suggest the production of oxalic acid through aqueous-phase reactions (e.g., Wang *et al.*, 2012, Bikkina *et al.*, 2017, Cheng *et al.*, 2017). Recently, real-time measurements of atmospheric oxalic acid and LWC in the US concluded that in contrast to monocarboxylic which was mainly formed through gas-phase photochemical reactions, oxalic acid was predominantly originated from aqueous processing (Chen *et al.*, 2021). Actually, because “no gas-phase reaction is known that results in oxalic acid (Ervens *et al.*, 2018)”, oxalic acid has been frequently used as tracer compound of both aerosol aqueous-phase processing and cloud-water processing (e.g., Lim *et al.*, 2010, Ervens *et al.*, 2011, Lim *et al.*, 2013, Charbouillot *et al.*, 2012, Wang *et al.*, 2020).

Last, we give evidences that oxalic acid is indeed aqueous-phase SOAs in our samples: (i) a significant correlation of oxalic acid with aerosol liquid water; (ii) depleted $\delta^{13}\text{C}$ of oxalic acid and its aqueous-phase precursor glyoxylic acid, which point to the $\delta^{13}\text{C}$ fractionation through aqueous-phase processing; (iii) the $\Delta^{14}\text{C}$ -derived fossil carbon contents of oxalic acid and glyoxylic acid were significantly higher than longer-chain diacids (C_3 and C_4 diacids), indicating an inconsistency of formation pathways between oxalic acid and its higher homologs, to which photochemical oxidation/breakdown in the gas-phase followed by partitioning onto the condensed phase may be more applied.

Taken together, while we cannot completely rule out primary emission and gas-phase reaction as its potential sources, oxalic acid would be the most suitable tracer of aqueous-phase chemical so

far, due to its outstanding role in aqueous-phase processing (as the end product of various aqueous-phase reaction pathways; e.g., Figure R1-1, Figure R1-2, and Figure R1-3) and its high abundance in organic aerosol (account for 5–9% of water-soluble organic carbon burden in global tropospheric; *Myriokefalitakis et al., 2011*). We also added a discussion on the limitation for the use of oxalic acid in the last section:

The revised text:

In this study, oxalic acid was assessed to be dominantly produced from aqueous-phase processes. However, it should be noted that both biomass burning and gas-phase photochemical aging could also be sources for oxalic acid. Therefore, cautious interpretations of oxalic acid isotopic signals are needed where there are intensive biomass burning, extensive atmospheric aging, and seasonality of C3/C4 vegetation changes. Future dual-carbon isotope studies on other aqSOA constituents (e.g., glyoxal) are strongly warranted to better constrain the chemical mechanism responsible for fossil-derived aqSOA formation. (Line 462-469)

Figure R1-1. In-cloud isoprene chemistry for formation of hygroscopic acids. (*Lim et al., 2005*)

Figure R1-2. Multiphase organic reactions leading to the formation of oxalic acid. Shaded area indicated aqueous phase. (Sorooshian et al., 2006)

[Redacted]

Reference:

- Bikkina S, Kawamura K, Sarin M 2017. Secondary organic aerosol formation over coastal ocean: inferences from atmospheric water-soluble low molecular weight organic compounds. *Environ. Sci. Technol.*, 51: 4347-4357.
- Carlton A G, Turpin B J, Altieri K E, et al. 2007. Atmospheric oxalic acid and SOA production from glyoxal: Results of aqueous photooxidation experiments. *Atmos. Environ.*, 41: 7588-7602.
- Carlton A G, Turpin B J, Lim H-J, et al. 2006. Link between isoprene and secondary organic aerosol (SOA): Pyruvic acid oxidation yields low volatility organic acids in clouds. *Geophys. Res. Lett.*, 33.
- Charbouillot T, Gorini S, Vyard G, et al. 2012. Mechanism of carboxylic acid photooxidation in atmospheric aqueous phase: Formation, fate and reactivity. *Atmospheric Environment*, 56: 1-8.
- Chen Y, Guo H, Nah T, et al. 2021. Low-Molecular-Weight Carboxylic Acids in the Southeastern U.S.: Formation, Partitioning, and Implications for Organic Aerosol Aging. *Environ. Sci. Technol.*, 55: 6688-6699.
- Cheng C, Li M, Chan C K, et al. 2017. Mixing state of oxalic acid containing particles in the rural area of Pearl River Delta, China: implications for the formation mechanism of oxalic acid. *Atmospheric Chemistry and Physics*, 17: 9519-9533.

- Crahan K K, Hegg D, Covert D S, et al. 2004. An exploration of aqueous oxalic acid production in the coastal marine atmosphere. *Atmospheric Environment*, 38: 3757-3764.
- Ervens B 2018. Progress and Problems in Modeling Chemical Processing in Cloud Droplets and Wet Aerosol Particles [M], *Multiphase Environmental Chemistry in the Atmosphere*. American Chemical Society: 327-345.
- Ervens B, Feingold G, Clegg S L, et al. 2004. A modeling study of aqueous production of dicarboxylic acids: 2. Implications for cloud microphysics. 109.
- Ervens B, Turpin B J, Weber R J 2011. Secondary organic aerosol formation in cloud droplets and aqueous particles (aqSOA): a review of laboratory, field and model studies. *Atmos. Chem. Phys.*, 11: 11069-11102.
- Huang X-F, Yu J Z 2007. Is vehicle exhaust a significant primary source of oxalic acid in ambient aerosols? *Geophys. Res. Lett.* 34.
- Kawamura K, Kaplan I R 1987. Motor exhaust emissions as a primary source for dicarboxylic acids in Los Angeles ambient air. *Environ. Sci. Technol.*, 21: 105-110.
- Kawamura K, Bikkina S 2016. A review of dicarboxylic acids and related compounds in atmospheric aerosols: Molecular distributions, sources and transformation. *Atmos. Res.*, 170: 140-160.
- Lim H-J, Carlton A G, Turpin B J 2005. Isoprene forms secondary organic aerosol through cloud processing: Model simulations. *Environ. Sci. Technol.*, 39: 4441-4446.
- Lim Y B, Tan Y, Perri M J, et al. 2010. Aqueous chemistry and its role in secondary organic aerosol (SOA) formation. *Atmos. Chem. Phys.*, 10: 10521-10539.
- Lim Y B, Tan Y, Turpin B J 2013. Chemical insights, explicit chemistry, and yields of secondary organic aerosol from OH radical oxidation of methylglyoxal and glyoxal in the aqueous phase. *Atmos. Chem. Phys.*, 13: 8651-8667.
- Myriokefalitakis S, Tsigaridis K, Mihalopoulos N, et al. 2011. In-cloud oxalate formation in the global troposphere: a 3-D modeling study. *Atmospheric Chemistry and Physics*, 11: 5761-5782.
- Peng J F, Hu M, Wang Z B, et al. 2014. Submicron aerosols at thirteen diversified sites in China: size distribution, new particle formation and corresponding contribution to cloud condensation nuclei production. *Atmospheric Chemistry and Physics*, 14: 10249-10265.
- Sorooshian A, Ng N L, Chan A W H, et al. 2007. Particulate organic acids and overall water-soluble aerosol composition measurements from the 2006 Gulf of Mexico Atmospheric Composition and Climate Study (GoMACCS). *Journal of Geophysical Research-Atmospheres*, 112.
- Sorooshian A, Varutbangkul V, Brechtel F J, et al. 2006. Oxalic acid in clear and cloudy atmospheres: Analysis of data from International Consortium for Atmospheric Research on Transport and Transformation 2004. 111.
- Tang J, Li J, Su T, et al. 2020. Molecular compositions and optical properties of dissolved brown carbon in biomass burning, coal combustion, and vehicle emission aerosols illuminated by excitation–emission matrix spectroscopy and Fourier transform ion cyclotron resonance mass spectrometry analysis. *Atmospheric Chemistry and Physics*, 20: 2513-2532.
- Wang G, Kawamura K, Cheng C, et al. 2012. Molecular distribution and stable carbon isotopic composition of dicarboxylic acids, ketocarboxylic acids, and α -dicarbonyls in size-resolved atmospheric particles from Xi'an City, China. *Environmental Science & Technology*, 46:

4783-4791.

- Wang J, Wang G, Wu C, et al. 2020. Enhanced aqueous-phase formation of secondary organic aerosols due to the regional biomass burning over North China Plain. *Environ. Pollut.*, 256: 113401.
- Warneck P 2003. In-cloud chemistry opens pathway to the formation of oxalic acid in the marine atmosphere. *Atmospheric Environment*, 37: 2423-2427.
- Warneck P J J o A C 2005. Multi-phase chemistry of C2 and C3 organic compounds in the marine atmosphere. 51: 119-159.
- Yuan J F, Huang X F, Cao L M, et al. 2016. Light absorption of brown carbon aerosol in the PRD region of China. *Atmospheric Chemistry and Physics*, 16: 1433-1443.
- Zhao W, Fu P, Yue S, et al. 2019. Excitation-emission matrix fluorescence, molecular characterization and compound-specific stable carbon isotopic composition of dissolved organic matter in cloud water over Mt. Tai. *Atmospheric Environment*.

(2). My second concern is about the novelty. $\delta^{13}\text{C}$ of oxalic acid and other dicarboxylic acids has been measured in many previous studies (including studies conducted at receptor background site like this study; Zhang et al., 2016, doi:10.1002/2015JD024081) to explore the formation pathway. It has been concluded that different formation pathway of oxalic acid leads to different ^{13}C signatures, and this fact can be used to track the formation pathway of oxalic acid.

Response: Many thanks for your comments. In this paper, we presented a full dataset of the compound-specific radiocarbon composition ($\Delta^{14}\text{C}$) of diacids in ambient aerosol samples, of course along with $\delta^{13}\text{C}$. To our knowledge, this is the first comprehensive application of compound-specific dual-carbon isotope technique to address SOA formation in the field.

First, we agree that the $\delta^{13}\text{C}$ of oxalic acid and other diacids has been measured in many studies to track the atmospheric processes of organic aerosols. To our knowledge, even though $\delta^{13}\text{C}$ -depletion is expected for fresh SOA formation, a more $\delta^{13}\text{C}$ -enriched oxalic acid has been reported and attributed to the photochemical aging of organic aerosols in almost all previous studies (e.g., Wang et al., 2006; Aggarwal et al., 2008; Pavuluri et al., 2011, 2012, 2016; Mkoma et al., 2014; Meng et al., 2020). This was interpreted by subsequent photochemical aging of recently formed particles ($\delta^{13}\text{C}$ -enrichment) overwhelming the $\delta^{13}\text{C}$ -depletion trend (Wang et al., 2012; Kirillova et al., 2014). However, thanks to the well-designed sampling campaign and unique geographic setting of the Heshan receptor site, for the first time, we observed substantial $\delta^{13}\text{C}$ -depletion signal for individual diacids, which is a strong field evidence standing for the proposed atmospheric aqueous-phase formation of oxalic acid.

Second, the primary advance of this paper is the $\Delta^{14}\text{C}$ -constrained precursor sources of individual aqSOA compounds, in addition to the $\delta^{13}\text{C}$ values. Up to now, there are only a few reports on the ^{14}C -based source apportionment of primary emitted organic compounds in aerosols, with applications much limited to n-alkanes, polycyclic aromatic hydrocarbons (PAHs) and n-fatty acids (e.g., *Kawamura et al., 2010; Xu et al., 2012; Ren et al., 2020*). These works had little concern on SOA. An important advance in compound-specific ^{14}C measurements of SOA compounds (oxalic acid) was done by *Fahrni et al., (2010)*, which is very enlightening for us. We thus decided to measure $\delta^{13}\text{C}$ - $\Delta^{14}\text{C}$ isotope fingerprints of individual SOA molecules in year-round ambient aerosols at the Heshan receptor site. The unique source information constrained by radiocarbon in combination with $\delta^{13}\text{C}$ would provide a new window to constrain both precursor sources and atmospheric chemical processes of SOA, which was difficult to be achieved by other approaches, such as a single $\delta^{13}\text{C}$ signature.

Last, we ascertained the large fossil carbon contribution to aqSOA, using compound-specific radiocarbon, which is contradict to previous paradigm. Many previous studies using radiocarbon of bulk WSOC did not suggest significant contribution from fossil carbon to aqSOA (e.g., *Weber et al., 2007*). However, bulk WSOC cannot afford for the source diversity of different SOA and primary soluble organics. Our unique $\delta^{13}\text{C}$ - $\Delta^{14}\text{C}$ dataset of paired oxalic acid and bulk WSOC confirmed that aqSOA do not necessarily share the same source patterns with WSOC. The large fossil-C signatures in aqueous SOA were masked in the bulk WSOC aerosols which reflect the average ^{14}C content of all compounds derived from multiple sources.

Our results represent the first step toward a compound-specific dual-carbon isotopic constraints on the sources of SOA compounds in ambient atmosphere. We think these efforts could greatly help to disentangle the currently poorly constrained source/formation pathways of aqSOA, and to provide highly valuable insights into air quality and climate issues. Future studies on the dual-carbon isotope compositions of other aqSOA constituents (e.g., glyoxal) could thus be anticipated.

Reference:

Aggarwal S G, Kawamura K 2008. Molecular distributions and stable carbon isotopic compositions of dicarboxylic acids and related compounds in aerosols from Sapporo, Japan: Implications for photochemical aging during long-range atmospheric transport. *J. Geophys. Res.: Atmos.*, 113.

- Fahrni S M, Ruff M, Wacker L, et al. 2010. A Preparative 2D-Chromatography Method for Compound-Specific Radiocarbon Analysis of Dicarboxylic Acids in Aerosols. *Radiocarbon*, 52: 752-760.
- Kawamura K, Matsumoto K, Uchida M, et al. 2010. Contributions of modern and dead organic carbon to individual fatty acid homologues in spring aerosols collected from northern Japan. *J. Geophys. Res.*, 115.
- Kirillova E N, Andersson A, Tiwari S, et al. 2014. Water-soluble organic carbon aerosols during a full New Delhi winter: Isotope-based source apportionment and optical properties. *J. Geophys. Res.: Atmos.*, 119: 3476-3485.
- Meng J, Liu X, Hou Z, et al. 2020. Molecular characteristics and stable carbon isotope compositions of dicarboxylic acids and related compounds in the urban atmosphere of the North China Plain: Implications for aqueous phase formation of SOA during the haze periods. *Science of The Total Environment*, 705: 135256.
- Mkoma S L, Kawamura K, Tachibana E 2014. Stable carbon isotopic compositions of low-molecular-weight dicarboxylic acids, glyoxylic acid and glyoxal in tropical aerosols: implications for photochemical processes of organic aerosols. *Tellus B: Chemical and Physical Meteorology*, 66.
- Pavuluri C M, Kawamura K 2012. Evidence for ^{13}C - carbon enrichment in oxalic acid via iron catalyzed photolysis in aqueous phase. *Geophysical Research Letters*, 39: 3802.
- Pavuluri C M, Kawamura K 2016. Enrichment of ^{13}C in diacids and related compounds during photochemical processing of aqueous aerosols: New proxy for organic aerosols aging. *Scientific Reports*, 6.
- Pavuluri C M, Kawamura K, Swaminathan T, et al. 2011. Stable carbon isotopic compositions of total carbon, dicarboxylic acids and glyoxylic acid in the tropical Indian aerosols: Implications for sources and photochemical processing of organic aerosols. *Journal of Geophysical Research-Atmospheres*, 116.
- Ren L, Wang Y, Kawamura K, et al. 2020. Source forensics of n-alkanes and n-fatty acids in urban aerosols using compound specific radiocarbon/stable carbon isotopic composition. *Environ. Res. Lett.*
- Wang G, Kawamura K, Cheng C, et al. 2012. Molecular distribution and stable carbon isotopic composition of dicarboxylic acids, ketocarboxylic acids, and α -dicarbonyls in size-resolved atmospheric particles from Xi'an City, China. *Environmental Science & Technology*, 46: 4783-4791.
- Wang H, Kawamura K 2006. Stable carbon isotopic composition of low-molecular-weight dicarboxylic acids and ketoacids in remote marine aerosols. *J. Geophys. Res.*, 111.
- Weber R J, Sullivan A P, Peltier R E, et al. 2007. A study of secondary organic aerosol formation in the anthropogenic-influenced southeastern United States. *J. Geophys. Res.*, 112.
- Xu L, Zheng M, Ding X, et al. 2012. Modern and Fossil Contributions to Polycyclic Aromatic Hydrocarbons in PM_{2.5} from North Birmingham, Alabama in the Southeastern U.S. *Environmental Science & Technology*, 46: 1422-1429.

(3). The ^{14}C measurement of oxalic acid is an interesting part, but the details of data interpretation, e.g., how to discriminate from different potential fossil sources, are not fully discussed and

sometimes are over-interpreted, leading to questionable conclusions. For example, ^{14}C measurements show that $55 \pm 10\%$ oxalic acid was contributed by fossil sources for continental outflow samples. Using source tracers (i.e., NO_3^- and tPh), the authors further qualitatively identify “both traffic and plastic burning emissions were important contributors to the fossil source-derived oxalic acid”. However, the data were overinterpreted: Line 242 – 244: “The high ratios of NO_3^- in atmosphere particles ($\sim 10\%$)” probably suggests that the traffic emissions contribute significantly to atmospheric particles, but from this, we cannot draw any firm conclusion about the relation between traffic emissions and the formation of oxalic acid, thus “a significant role of traffic emission-derived precursors in the formation of oxalic acid” is an overinterpretation and should be revised.

Line 245 – 248: How did the authors conclude “the importance of precursors from plastic burning to the formation of fossil oxalic acid” from the high contribution of tPh to total diacids (5%), given that tPh is a combustion tracer of plastic polymers? Again, it is an overinterpretation. The high contribution of tPh to total diacids is likely pointing to the importance of plastic burning to total acids. However, one cannot conclude any relation between plastic burning and the formation of fossil oxalic acid. Therefore, the conclusion that “both traffic and plastic burning emissions were important contributors to the fossil source-derived oxalic acid” is questionable and should be reformulated.

Response: Thanks for your kind reminding. We agree that the conclusion in the Line 242-249 overinterpret what can be inferred from the data. The high concentration of NO_3^- and terephthalic acid only suggest that traffic emissions and plastic burning significantly contribute to the atmospheric particles. The available data do not support us to further distinguish the fossil sources of oxalic acid quantitatively. Moreover, the relatively contribution of different fossil sources to oxalic acid (e.g., traffic, coal burning, and plastic burning) is not the main interest of this manuscript. We particularly concerned with the ^{14}C -constrained total fossil contributions to the formation of aqSOA compounds. For these reasons, we decided to downplay the discussion about the contribution of different fossil sources to oxalic acid. Please refer to the revised manuscript, in which changes are highlighted in yellow.

The revised text:

The high ratios of NO_3^- in atmospheric particles ($\sim 10\%$; Supplementary Fig. 4) indicate a

significant contribution of traffic emissions⁴³ to atmospheric particles. Moreover, terephthalic acid (tPh) was one of the most abundant anthropogenic diacids accounting for 5% of total diacids (Supplementary Table 3). As tPh is a combustion tracer of plastic polymers⁴⁴, this highlights the involvement of precursors from plastic burning to the formation of diacids. Thus, both traffic and plastic burning emissions were readily among the contributors to the fossil source-derived oxalic acid. (Line 250-257)

(4).Line 167 – 169: “potential precursors of oxalic acid would then be anthropogenic VOCs (AVOCs; e.g., aromatic hydrocarbons: $-27.7 \pm 1.7\%$ ³⁷) and C₃-plant-derived biogenic VOCs (BVOCs; e.g., isoprene: $-27 \pm 2\%$ ³⁸), of which the $\delta^{13}\text{C}$ values are also similar”. In the Ref. 38, the $\delta^{13}\text{C}$ values of $-27.7 \pm 2.0\%$ is for isoprene emitted from Velvet Bean (*Mucuna pruriens* L. var. utilis). Does the specific “Velvet Bean” represent the biogenic source of isoprene in this study? In other words, is the ^{13}C signature of isoprene in this study different from $-27 \pm 2\%$ (i.e., $\delta^{13}\text{C}$ value of isoprene from Velvet Bean), so that different from $-27.7 \pm 1.7\%$ (i.e., $\delta^{13}\text{C}$ value for AVOCs)? I am concern about this point, because this is closely related to the argument that “the isotopic range of various sources of oxalic acid is minimal, leaving kinetic isotopic fractionation the most probable processes controlling the $\delta^{13}\text{C}$ of oxalic acid” (Line 170 – 172).

Response: Thanks for your comments. We agree with the reviewer that more vegetational form should be examined to demonstrate that the $\delta^{13}\text{C}$ composition of C₃ plant-derived isoprene is indeed close to the $\delta^{13}\text{C}$ of AVOCs. We compiled publications of $\delta^{13}\text{C}$ -BVOCs studies through the databases of the Web of Science and Google Scholar. The available data is listed in **TableR1-1**. We mostly concern isoprene because it represents half of the flux of the BVOCs emitted into the atmosphere (*Lamkaddam et al., 2021*) and has been demonstrated as the representative precursor of biogenic oxalic acid (*Bikkina et al., 2014*). *Affek and Yakir [2003]* found that the average $\delta^{13}\text{C}$ -isoprene values was $-29.2\% \pm 0.6\%$ from three types of C₃ plants (myrtle, buckthorn, and velvet bean). *Sharkey et al. [1991]* found an average carbon isotope fractionation of -2.8% for isoprene emitted from oak leaves relative to recently fixed carbon. *Rudolph et al. [2003]* gives an average $\delta^{13}\text{C}$ -isoprene of $-27.7\% \pm 2\%$ for velvet bean, which was 2.6% lighter than bulk leaf carbon, very similar to -2.8% found for oak leaves. Rudolph and coauthors thus estimated that the plausible range for isoprene emitted from vegetation is from -27% to -32% , based on the average of

approximately -27‰ for carbon assimilated by the C3-metabolism. Rudolph and coauthors further analyzed the $\delta^{13}\text{C}$ -isoprene in a small set of ambient air collected at mature mixed deciduous forest areas containing various C3-tree species (Iannone *et al.*, 2007). The resulted $\delta^{13}\text{C}$ values for isoprene are typically in the range from -29‰ to -26‰, consistent with those reported in the literature. Gromov *et al.* [2017] proposed a novel approach referring to the physiological properties of plants, and reckon the $\delta^{13}\text{C}$ values of surface emissions of isoprene within -30.4‰ to -29‰. Li *et al.* [2019] analyzed the $\delta^{13}\text{C}$ values of isoprene ($-25.2\text{‰} \pm 1.4\text{‰}$) in four forest areas in Southwest China, and successfully distinguished the isoprene into two sources: ~73% from C3 plants (adopt $\delta^{13}\text{C}$ -isoprene endmember of -29‰) and ~27% from C4 plants ($\delta^{13}\text{C}$ -isoprene endmember of -15‰).

Furthermore, the C3 plants-derived terpene, whose carbon skeletons is composed of isoprene units (Ruzicka, 1957), also exhibited consistent $\delta^{13}\text{C}$ values with isoprene. For example, Diefendorf *et al.* (2012) found an average $\delta^{13}\text{C}$ value of $-27.1\text{‰} \pm 2.5\text{‰}$ for di- and triterpenoids emitted from 44 C3 trees. Together, the variability of isoprene $\delta^{13}\text{C}$ values appear in different plant species, seasons, and geography would be very minor. Per your good suggestion, we aware that the $\delta^{13}\text{C}$ composition of isoprene ($-27.7 \pm 2\text{‰}$) emitted from the specific velvet bean lacks representativeness. The average $\delta^{13}\text{C}$ composition of isoprene (-29‰ to -26‰) suggested by Iannone *et al.*, [2007] could better represent the biogenic source of isoprene in our study. Thus, the Ref 38 “**Rudolph J, et al. The stable carbon isotope ratio of biogenic emissions of isoprene and the potential use of stable isotope ratio measurements to study photochemical processing of isoprene in the atmosphere. J. Atmos. Chem. 44, 39-55 (2003)**” was revised to “**Iannone R, Koppmann R, Rudolph J. A technique for atmospheric measurements of stable carbon isotope ratios of isoprene, methacrolein, and methyl vinyl ketone. Journal of Atmospheric Chemistry, 58: 181-202 (2007)**”. However, this revision would not affect our argument that the $\delta^{13}\text{C}$ values of C3-plants derived biogenic VOCs and anthropogenic VOCs are very similar.

The revised text:

Given contributions from marine and C4 plants sources are excluded, the potential precursors of oxalic acid would then be anthropogenic VOCs (AVOCs; e.g., aromatic hydrocarbons: $-27.7 \pm 1.7 \text{‰}$ ³⁷) and C3 plant-derived biogenic VOCs (BVOCs; e.g., isoprene: -29‰ to -26‰³⁸), of which the $\delta^{13}\text{C}$ values are also similar. (Line 172-176)

Table R1-1. $\delta^{13}\text{C}$ composition of isoprene and terpenoids directly emitted from C3 plants

Plant Species	Compounds	$\delta^{13}\text{C}$	References
Myrtle, buckthorn, velvet bean	Isoprene	$-29.2\text{‰} \pm 0.6\text{‰}$	Affek and Yakir (2003)
velvet bean	Isoprene	$-27.7\text{‰} \pm 2\text{‰}$	Rudolph et al. (2003)
Mixed deciduous forest in Germany	Isoprene	-29‰ to -26‰	Iannone et al. (2007)
Surface emission (model)	Isoprene	-30.4‰ to -29‰	Gromov et al. (2017)
Forest areas in Southwest China	Isoprene	-29‰	Li et al. (2019)
44 C3 plants	Terpenoids	$-27.1\text{‰} \pm 2.5\text{‰}$	Diefendorf et al. (2012)

Reference:

- Affek H P, Yakir D 2003. Natural abundance carbon isotope composition of isoprene reflects incomplete coupling between isoprene synthesis and photosynthetic carbon flow. *Plant Physiol.*, 131: 1727-1736.
- Bikkina S, Kawamura K, Miyazaki Y, et al. 2014. High abundances of oxalic, azelaic, and glyoxylic acids and methylglyoxal in the open ocean with high biological activity: Implication for secondary OA formation from isoprene. *Geophys. Res. Lett.*, 41: 3649-3657.
- Diefendorf A F, Freeman K H, Wing S L 2012. Distribution and carbon isotope patterns of diterpenoids and triterpenoids in modern temperate C3 trees and their geochemical significance. *Geochimica et Cosmochimica Acta*, 85: 342-356.
- Gromov S, Brenninkmeijer C A M, Jöckel P 2017. Uncertainties of fluxes and $^{13}\text{C} / ^{14}\text{C}$ ratios of atmospheric reactive-gas emissions. *Atmospheric Chemistry and Physics*, 17: 8525-8552.
- Iannone R, Koppmann R, Rudolph J 2007. A technique for atmospheric measurements of stable carbon isotope ratios of isoprene, methacrolein, and methyl vinyl ketone. *Journal of Atmospheric Chemistry*, 58: 181-202.
- Lamkaddam H, Dommen J, Ranjithkumar A, et al. 2021. Large contribution to secondary organic aerosol from isoprene cloud chemistry. *Sci. Adv.*, 7: eabe2952.
- Li L, Zhou Y, Bi X, et al. 2019. Determination of the stable carbon isotopic compositions of 2-methyltetrols for four forest areas in Southwest China: The implications for the $\delta^{13}\text{C}$ values of atmospheric isoprene and C3/C4 vegetation distribution. *Science of The Total Environment*, 678: 780-792.
- Rudolph J, Anderson R S, Czapiewski K V, et al. 2003. The stable carbon isotope ratio of biogenic emissions of isoprene and the potential use of stable isotope ratio measurements to study photochemical processing of isoprene in the atmosphere. *J. Atmos. Chem.*, 44: 39-55.
- Ruzicka L 1953. The isoprene rule and the biogenesis of terpenic compounds. *Experientia*, 9: 357-367.
- Sharkey T D, Loreto F, Delwiche C F, et al. 1991. Fractionation of Carbon Isotopes during Biogenesis of Atmospheric Isoprene 1. *Plant Physiology*, 97: 463-466.

(5).Line 173 – 180: Somewhere in the manuscript, to facilitate the reader's understanding, it is good

to explain how the $\delta^{13}\text{C}$ values change during the two formation pathways of oxalic acid.

Response: Thanks a lot for the suggestion. We have revised the manuscript to explain how the $\delta^{13}\text{C}$ values change during the two formation pathways of oxalic acid:

The revised text:

The first pathway is a stepwise photochemical breakdown of longer-chain aliphatic diacids^{27,28}, which leads to relatively enriched $\delta^{13}\text{C}$ values in the residual shorter-chain diacids³⁹. (Line 179-181)

The second pathway is the oxidation of BVOCs and AVOCs to semi-volatile organic compounds (SVOCs, e.g., Gly and MeGly), followed by partitioning of SVOCs to wet aerosols or cloud/fog droplets, with subsequent aqueous-phase transformation to pyruvic acid (Pyr) and glyoxylic acid (ωC_2) and finally oxidized to oxalic acid²⁴⁻²⁶. In this case, oxalic acid-C carries a depleted $\delta^{13}\text{C}$ signal as the oxalic acid is an oxidation product, as opposed to a residual C pool. (Line 183-189)

(6).Line 239 – 242: In Supplementary Fig.5, the data points are very scatter, and it looks like that the correlation between $f_{\text{fossil-C}_2}$ and C_2 concentrations for either continental outflow or coastal background is weak.

Responses: Thanks for your comment. As shown in new **Supplementary Figure 6a**, a positive relationship apparently existed between the fossil contributions to oxalic acid and the oxalic acid concentrations, despite the scattered datapoints appeared in the plot ($r^2 = 0.52$, $P < 0.001$). In contrast, a negative relationship was observed between the non-fossil contributions to oxalic acid and the oxalic acid concentrations, suggesting that non-fossil sources are very unlikely to be responsible for the increasing of the ambient abundance of oxalic acid (**Supplementary Figure 6b**). Therefore, we think the argument “The concentrations of C_2 oxalic acid increased with the increment of $f_{\text{fossil-C}_2}$ ($r^2 = 0.52$, $P < 0.001$; Supplementary Fig. 5), suggesting that the precursors from fossil-fuel emission was the dominant factor in controlling the ambient abundance of oxalic acid” is a reasonable interpret of the data. Per your good suggestion, we revised this sentence to express this point more clearly:

The revised sentence:

“The concentrations of C_2 oxalic acid increased with the increase of $f_{\text{fossil-C}_2}$ ($r^2 = 0.52$, $P < 0.001$; Supplementary Fig. 6), suggesting that the precursors from fossil-fuel emission was the major

factor responsible for the enhanced ambient abundance of oxalic acid. (Line 247-250)

Supplementary Figure 6. Relationship of (a) oxalic acid and the fraction fossil in oxalic acid and (b) oxalic acid and the fraction of non-fossil in oxalic acid.

(7).Line 380 – 381: Inconsistence between the text and figure. Text says “Oxalic acid is more enriched in ¹⁴C than WSOC in summer”, but Fig.6b show the opposite.

Response: Thanks for your comment. To better visualized the dataset, we plotted the dual-carbon isotopic composition (¹³C and ¹⁴C) of WSOC and oxalic acid from the five megacities in one Figure (**Figure R1-4**). Although in some sites, the difference between the carbon isotopic composition of oxalic acid and WSOC are very minor within the analytical error (“overlapped”), the trend that oxalic acid carries more enriched ¹³C and ¹⁴C signal than WSOC in summer whereas carries more depleted ¹³C and ¹⁴C signal than WSOC in winter is obvious as shown in **Figure R1-4**. Generally, we think that the overlapped points in summer Beijing doesn’t conflict with our argument “oxalic acid was more enriched in ¹³C and ¹⁴C than WSOC in summer”.

Figure R1-4. The $\delta^{13}\text{C}$ composition and ^{14}C -based fraction of non-fossil sources for water-soluble organic carbon (WSOC; red dot) and oxalic acid (blue dot) in Beijing (BJ), Guangzhou (GZ), Wuhan (WH), Chengdu (CD) and Shanghai (SH) during (a) summer and (b) winter. The error bars represent total propagated uncertainty.

(8).Line 428: year 2100?

Response: Many thanks! The “year of 2100” have been corrected to “year 2100”

(9).Line 524: If using the NBS Oxalic Acid I standard ($\delta^{13}\text{C}$ value close to -19‰), the ^{14}C is measured and normalized to $\delta^{13}\text{C} = -19\text{‰}$ (not -25‰ as stated in the manuscript). Almost certainly the authors used Oxalic Acid II, because Oxalic Acid I is no longer commercially available. Using Oxalic Acid II, the ^{14}C is normalized to -25‰

Response: Thanks for your kind reminding. In our laboratory, the fraction modern (F_m) results were normalized to $\delta^{13}\text{C}$ value of -25‰ using oxalic acid II standard in the following equation (Stenström et al. 2011):

$$\left(\frac{^{14}\text{C}}{^{12}\text{C}}\right)_{1950[-25]} = 0.95 \left(\frac{^{14}\text{C}}{^{12}\text{C}}\right)_{OXI} \left(\frac{1 - \frac{19}{1000}}{1 + \frac{\delta^{13}\text{C}_{OXI}}{1000}}\right)^2 = 0.7459 \left(\frac{^{14}\text{C}}{^{12}\text{C}}\right)_{OXII} \left(\frac{1 - \frac{25}{1000}}{1 + \frac{\delta^{13}\text{C}_{OXII}}{1000}}\right)^2$$

Aware of the previous expression would lead readers mistakenly believed that oxalic acid I standard were used in our laboratory, we have revised this sentence.

The revised text:

In all cases, ^{14}C results are reported as “fraction of modern carbon” (F_m) normalized to a common $\delta^{13}\text{C}$ value of -25‰. (Line 551-552)

Reference:

Stenström K et al, A guide to radiocarbon units and calculations. Lund University, Department of Physics, Division of Nuclear Physics Internal Report LUNFD6(NFFR-3111)/1-17/(2011)

(10). Supplemental line 120 – 122: “ $F_{m,ref} = 1.10$ ” for wood burning. Please clarify assumptions on wood age and fell date.

Response: Per your good suggestion, we have added more details on the values of $F_{m,ref}$ in **Supplemental Text S3**.

The revised text in supporting material Text S3:

Where the $F_{m,ref}$ is a reference value of F_m for contemporary carbon sources including biogenic emissions and biomass burning ($F_{m,bio}$ and $F_{m,bb}$, respectively). The values of $F_{m,bio}$ and $F_{m,bb}$ is parameterized following previous protocol¹¹. $F_{m,bio}$ is obtained from global contemporary $^{14}\text{CO}_2$ in year 2017–2018 at two representative background $\Delta^{14}\text{CO}_2$ observation station located in northern Hemisphere (Jungfrauoch; $\Delta^{14}\text{CO}_2=5.3\%$; <https://data.icos-cp.eu>) and Southern Hemisphere (Wellington; $\Delta^{14}\text{CO}_2=14.8\%$; <https://www.gns.cri.nz>), respectively¹². The value of $F_{m,bb}$ is higher than $F_{m,bio}$ because it is reflecting the $\Delta^{14}\text{C}$ of biomass that has accumulated over the decades-to-century-long life span of trees. For East Asia, there are several important contemporary biofuel types, including wood fuel and crop residue (freshly produced biomass). The $\Delta^{14}\text{C}$ for wood fuel ($\Delta^{14}\text{C} = 155\%$) was estimated by a tree-growth model¹³ including 10-year, 20-year, 40-year, 70-year, and 85-year old trees with weights of 0.2, 0.2, 0.4, 0.1, and 0.1, respectively, logged in the 2010s. The $\Delta^{14}\text{C}$ for cope residue corresponds to contemporary $^{14}\text{CO}_2$ in year 2016 ($\Delta^{14}\text{C}=15\%$). To regionally parameterize the contemporary $\Delta^{14}\text{C}_{bb}$ end member, the relative contribution of fuel wood (57%) and crop residue (43%) provided by Tao et al.¹⁴ were employed. Hence, a China-tailored $\Delta^{14}\text{C}_{bb}$ of

95‰ ($\Delta^{14}\text{C}_{bb} = 155\text{‰} \times 0.57 + 15\text{‰} \times 0.43$) was used. The corresponding $F_{m, bio}$ and $F_{m, bb}$ values were 1.02 and 1.10, respectively, calculated by the following equation:

$$\Delta^{14}\text{C} = (Fm \times e^{\lambda(1950-x)} - 1) \times 1000\text{‰} \quad \text{Eq. S8}$$

where λ is 1/(true mean-life) of radiocarbon (corresponding to 1/8267) and x is the year of collection (in this case 2018). The $F_{m,ref}$ were estimated as 1.06 (i.e., $F_{m,ref} = 1.10 \times 0.5 + 1.02 \times 0.5$), based on the assumption that contemporary carbon originates equally from biogenic emission and biomass burning¹³.

Reference in supporting material:

11. Gustafsson Ö, *et al.* Brown Clouds over South Asia: Biomass or Fossil Fuel Combustion. *Science* **323**, 495-498 (2009).
12. Zhang G, *et al.* Radiocarbon isotope technique as a powerful tool in tracking anthropogenic emissions of carbonaceous air pollutants and greenhouse gases: A review. *Fundamental Research* **1**, 306-316 (2021).
13. Zhang YL, *et al.* Radiocarbon-based source apportionment of carbonaceous aerosols at a regional background site on Hainan Island, South China. *Environ Sci Technol* **48**, 2651-2659 (2014).
14. Tao S, *et al.* Quantifying the rural residential energy transition in China from 1992 to 2012 through a representative national survey. *Nature Energy* **3**, 567-573 (2018).

Responses to reviewer #2:

With “Large contribution of fossil-derived components to aqueous secondary organic aerosols in China,” Xu et al. have presented a large dataset of aerosol organic compositional measurements for samples with differing air mass histories. The data appear to show evidence for the aqueous phase production of secondary organic aerosols (SOA) from fossil-derived carbon. Many previous studies have demonstrated aqueous SOA formation and the influence of fossil-derived inorganic species, but radiocarbon measurements did not suggest significant contributions from fossil C. Rather, the previous work showed the C to be derived from natural biomass – so the findings here of very high fossil C contributions to both water soluble organic carbon and oxalic acid (an aqueous SOA tracer) are novel. The paper is well-reasoned and well-written. I do, however, have a potential concern about the use of ultrasonication to produce the water soluble aerosol extracts. This concern is elaborated upon below; the major concern is that ultrasonication may have produced compositional artifacts that render the results non-representative of ambient aerosol composition. In addition to this concern, I outline additional comments and suggestions in the comments that follow. Should the methodological concerns be allayed, this work would effectively demonstrate previously unrecognized inputs of fossil fuel-derived C to the secondary organic aerosol pool and have significant impact on atmospheric processes and climate.

Response: Thank for your nice summary of our paper and positive assessment of the importance of this work. We have carefully revised the manuscript following your comments and suggestions. Our responses to your comments are given below, and the revised parts are shown after the responses (in blue font). Please refer to the revised manuscript, in which changes are highlighted in yellow.

General comments:

(1). Ultrasonication: In the Supplementary Text 4 section, ultrasonication is used to produce water extracts for isotopic analyses. Ultrasonication is known to produce reactive oxygen species (e.g., hydroxyl radical, superoxide radical, singlet oxygen) and organic radicals and is not recommended for extractions intended to characterize environmentally-relevant organic compositions. (See e.g., Miljevic et al. 2014, Aerosol Sci. Tech., 48:1276-1284) These free radical species, once produced,

are well known to react with organic molecules and alter organic molecular composition. Such organic transformation reactions can solubilize previously insoluble organics and produce small molecules such as oxalic acid. In fact, radical species are major players in the photochemical transformation of atmospheric species in the gaseous, aqueous, and particle phases. The use of ultrasonication, therefore, has the high potential for producing artifacts by reacting ambient aerosol organics (effectively driving the aqueous reactions the authors are suggesting occur naturally) and altering the fraction of WSOC as well as the aerosol molecular composition. Thus, a major question that I have is: were all water extractions conducted using ultrasonication? If so, do the authors have evidence that the technique does not alter the molecular composition of aerosol organics and specifically the quantities and isotopic composition of oxalic acid and WSOC that are the focus of this work? If ultrasonication was not used, please clarify the water extraction technique employed.

Response: We appreciate the reviewer in raising this “potential concern” and stimulating us to look into this in detail. While ultrasonication is broadly used for extraction of water-soluble organic materials from PM quartz filters, this is also the first time we aware that ultrasonication may have a protentional influence on the molecular composition and isotopic composition of aerosol organic substances.

To explore/address this important question, we first consulted the literature, and reviewed established protocols for extraction of both WSOC and oxalic acid from PM quartz filters:

- (1) WSOC:** Studies aimed at quantification of aerosol WSOC have used both ultrasonication (e.g., Decesari *et al.*, 2000,2001, Yang *et al.*, 2003, Mader *et al.*, 2004) and soaking (e.g., Facchini *et al.*, 1999, Graham *et al.*, 2002, Kiss *et al.*, 2002). WSOC mass recoveries of both ultrasonication (e.g., Karthikeyan *et al.*, 2005) and soaking (e.g., Wozniak *et al.*, 2012) have demonstrated >90% yields. There is to our knowledge only one study that has directly compared these two extraction protocols. Mladenov *et al.*, [2009] reported that the concentrations of WSOC extracted from filters by sonication, agitation, and soaking techniques were statistically inseparable. For studies of the isotopic composition of WSOC, particularly for carbon isotopes ($\delta^{13}\text{C}$ and $\Delta^{14}\text{C}$), ultrasonication extraction method has been used (e.g., Weber *et al.*, 2007, Wozniak *et al.*, 2008, Kirillova *et al.*, 2013, Liu *et al.*, 2018, Dasari *et al.*, 2019, Mo *et al.*, 2021). The $\delta^{13}\text{C}$ - and $\Delta^{14}\text{C}$ -WSOC analysis method using ultrasonication extraction protocols was thoroughly evaluated and established by

Kirillova et al., [2010] and published in the ACS journal Analytical Chemistry. This methods study demonstrated well-preserved carbon isotopic signal of WSOC through the method by the addition of synthetic WSOC probes of known isotopic composition to blank filter.

- (2) **Oxalic acid:** To the best of our knowledge, there is an overwhelming acceptance and use of ultrasonication protocol to extract oxalic acid from PM quartz filters throughout the peer-reviewed literature, irrespective of whether IC (e.g., *Baboukas et al., 2000, Karthikeyan et al., 2005, Huang et al., 2006, Yang and Yu 2008, Zhao and Gao 2008, Kawamura et al., 2010, Yu et al., 2021*) or GC (e.g., *Kawamura et al., 1993, Wang et al., 2002, Kawamura et al., 2010, Bikkina et al., 2017, Li et al., 2021*) were eventually employed to measure the concentration of oxalic acid. To the best of our knowledge, there has not been any direct comparison between ultrasonication and any other extraction protocols for oxalic acid. Reported analytical methods for $\delta^{13}\text{C}$ (*Kawamura and Watanabe, 2004*) and $\Delta^{14}\text{C}$ (*Fahrni et al., 2010, Xu et al., 2021*) of oxalic acid have applied ultrasonication and have demonstrated high recoveries. The validity of these methods was demonstrated by adding authentic oxalic acid standard to blank filter or the extracts.

Taken together, while one cannot completely exclude some effects, there is an overwhelming support in the peer-reviewed literature, both in Methods papers and Application papers, of high recoveries and intact isotopic composition of WSOC and oxalic acids in methods using ultrasonication as extraction method for aerosol samples.

Nevertheless, we have inspired by this review comment now conducted additional experiment to compare ultrasonication with soaking regarding their consistency in measurements of both concentration and $\delta^{13}\text{C}$ composition of oxalic acid and WSOC, respectively. The ambient $\text{PM}_{2.5}$ sample was collected at Guangzhou Institute of Geochemistry (23.1°N, 113.3°E), Guangzhou, China, December 9–11, 2021. An aliquot of the filter (4.9 cm²) was soaked in 15 mL Milli-Q water at room temperature (15–20°C) in dark for 24 hours to allow for water-soluble material desorption. Meanwhile, the same aliquot of the filter was extracted with 15 mL of Milli-Q water in a sonication bath. Ice water was used in the sonication bath to minimize possible loss of volatile OM. Ultrasonication were performed at the highest ultrasonic energy (~100W) for different periods of time (30-150 min). The extracts were then filtered through a 0.22 μm PTFE filter and analyzed for

the concentration and $\delta^{13}\text{C}$ composition of oxalic acid and WSOC (reports shown in the new Supplementary Table S9 and Supplementary Figure S11).

As shown in Figure S11a and S11b, the average concentration of WSOC and oxalic acid extracted by ultrasonication for 30 min (the extraction protocol applied in the manuscript) cannot be distinguished statistically from soaking for 24 hours. The WSOC and oxalic acid extracted by the ultrasonication-30min method yield $\delta^{13}\text{C}$ composition of $-24.53 \pm 0.09\%$, which is identical to the $\delta^{13}\text{C}$ signatures of WSOC from soaking ($-24.50 \pm 0.10\%$) (Figure S11c). Consistent $\delta^{13}\text{C}$ -oxalic acid values were also observed among the two extraction methods (Figure S11d). The increased ultrasonication times of 0.5–1–1.5–2–2.5 hours resulted similar concentration and $\delta^{13}\text{C}$ composition for WSOC and oxalic acid (Figure S11), respectively, further implying that any reactive oxygen species potentially produced from ultrasonication would not measurably alter the molecular and isotopic composition of WSOC and/or oxalic acid. Some previous studies also demonstrated that the extended ultrasonication time would not affect the recovery of water-soluble organics from ambient aerosols (*Karthikeyan et al., 2005, Kirillova et al., 2010*). The negligible organic transformation reactions during ultrasonication may be explained by the limited reactive oxygen species produced, the dark ultrasonication condition, and the lack of reaction surfaces in water solution (*NcNeill 2015*). Taken together, our dedicated experiments add to the earlier broad literature reviewed above to demonstrate that the ultrasonication method applied in our study yields reliable information on molecular and isotopic composition of WSOC and oxalic acid. We also added a discussion on the validity of the ultrasonication method in the Materials and Methods:

The revised text in Materials and Methods:

The analytical method for the measurements of \$\delta^{13}\text{C}\$ and \$\Delta^{14}\text{C}\$ composition of WSOC builds on ultrasonication extraction protocols were described previously⁵⁵ and reported in Supplementary Text 4. The validity of the ultrasonication method during extraction of water-soluble organic materials was demonstrated by comparing ultrasonication with another extraction protocol, i.e., soaking (see Supplementary Table 9 and Supplementary Figure 11). (Line 566-569)

Supplementary Figure 11. Tests for the extraction protocols. (a) concentration of WSOC after soaking or ultrasonication extraction. (b) concentration of oxalic acid after soaking or ultrasonication extraction. (c) $\delta^{13}\text{C}$ composition of WSOC after soaking or ultrasonication extraction. (d) $\delta^{13}\text{C}$ composition of oxalic acid dibutyl esters (C_2BE) after soaking or ultrasonication extraction. The error bars represent standard deviation from three replicate experiments.

Supplementary Table 9. Comparison of soaking and ultrasonication method

	Soaking-24 h ^a				Ultrasonic-30 min ^a				Ultrasonic -60 min	Ultrasonic -90 min	Ultrasonic -120 min	Ultrasonic -150 min
	first	second	third	av	first	second	third	av				
WSOC ($\mu\text{g m}^{-3}$)	3.86	4.18	4.25	4.10 \pm 0.21	4.35	4.02	3.90	4.09 \pm 0.23	4.35	4.53	4.19	4.17
$\delta^{13}\text{C}$-WSOC (‰)	-24.44	-24.62	-24.43	-24.50 \pm 0.10	-24.43	-24.59	-24.59	-24.53 \pm 0.09	-24.49	-24.50	-24.61	-24.42
C_2 (ng m^{-3})	384.8	344.1	328.0	352.3 \pm 29.3	304.3	373.0	353.7	343.7 \pm 35.4	340.4	328.0	390.4	333.0
$\delta^{13}\text{C}$-C_2BE^b (‰)	-27.84	-27.70	-27.71	-27.75 \pm 0.08	-27.87	-27.67	-27.60	-27.70 \pm 0.14	-27.84	-27.98	-27.81	-27.80

a. three replicate experiments were performed. b. oxalic acid dibutyl esters

References:

- Baboukas E D, Kanakidou M, Mihalopoulos N 2000. Carboxylic acids in gas and particulate phase above the Atlantic Ocean. *105*: 14459-14471.
- Bikkina S, Kawamura K, Sarin M 2017. Secondary organic aerosol formation over coastal ocean: inferences from atmospheric water-soluble low molecular weight organic compounds. *Environ. Sci. Technol.*, *51*: 4347-4357.
- Dasari S, Andersson A, Bikkina S, et al. 2019. Photochemical degradation affects the light absorption of water-soluble brown carbon in the South Asian outflow. *Sci. Adv.*, *5*: eaau8066.
- Decesari S, Facchini M C, Fuzzi S, et al. 2000. Characterization of water - soluble organic compounds in atmospheric aerosol: A new approach. *105*: 1481-1489.
- Decesari S, Facchini M C, Matta E, et al. 2001. Chemical features and seasonal variation of fine aerosol water-soluble organic compounds in the Po Valley, Italy. *Atmospheric Environment*, *35*: 3691-3699.
- Elena N K, Rebecca J S, August A, et al. 2010. Natural abundance ¹³C and ¹⁴C analysis of water-soluble organic carbon in atmospheric aerosols. *Analytical Chemistry*, *82*: 7973.
- Facchini M C, Fuzzi S, Zappoli S, et al. 1999. Partitioning of the organic aerosol component between fog droplets and interstitial air. *104*: 26821-26832.
- Fahmi S M, Ruff M, Wacker L, et al. 2010. A Preparative 2D-Chromatography Method for Compound-Specific Radiocarbon Analysis of Dicarboxylic Acids in Aerosols. *Radiocarbon*, *52*: 752-760.
- Graham B, Mayol - Bracero O L, Guyon P, et al. 2002. Water - soluble organic compounds in biomass burning aerosols over Amazonia 1. Characterization by NMR and GC - MS. *107*: LBA 14-11-LBA 14-16.
- Huang X-F, Yu J Z, He L-Y, et al. 2006. Water-soluble organic carbon and oxalate in aerosols at a coastal urban site in China: Size distribution characteristics, sources, and formation mechanisms. *Journal of Geophysical Research-Atmospheres*, *111*.
- Karthikeyan S, Balasubramanian R 2005. Rapid extraction of water soluble organic compounds from airborne particulate matter. *Analytical Sciences*, *21*: 1505-1508.
- Kawamura K, Barrie L A, Toom-Saunty D 2010. Intercomparison of the measurements of oxalic acid in aerosols by gas chromatography and ion chromatography. *Atmospheric Environment*, *44*: 5316-5319.
- Kawamura K, Ikushima K 1993. Seasonal changes in the distribution of dicarboxylic acids in the urban atmosphere. *Environ. Sci. Technol.*, *27*: 2227-2235.
- Kawamura K, Watanabe T 2004. Determination of stable carbon isotopic compositions of low molecular weight dicarboxylic acids and ketocarboxylic acids in atmospheric aerosol and snow samples. *Anal. Chem.*, *76*: 5762-5768.
- Kirillova E N, Andersson A, Sheesley R J, et al. 2013. ¹³C- and ¹⁴C-based study of sources and atmospheric processing of water-soluble organic carbon (WSOC) in South Asian aerosols. *J. Geophys. Res.: Atmos.*, *118*: 614-626.
- Kiss G, Varga B, Galambos I, et al. 2002. Characterization of water-soluble organic matter isolated from atmospheric fine aerosol. *Journal of Geophysical Research-Atmospheres*, *107*.
- Li P, Pavuluri C M, Dong Z, et al. 2021. Characteristics, Seasonality, and Secondary Formation Processes of Diacids and Related Compounds in Fine Aerosols During Warm and Cold

- Periods: Year - Round Observations at Tianjin, North China. *Journal of Geophysical Research: Atmospheres*, 126.
- Liu J, Mo Y, Ding P, et al. 2018. Dual carbon isotopes (^{14}C and ^{13}C) and optical properties of WSOC and HULIS-C during winter in Guangzhou, China. *Sci Total Environ*, 633: 1571-1578.
- Mader B, Yu J, Xu J, et al. 2004. Molecular composition of the water - soluble fraction of atmospheric carbonaceous aerosols collected during ACE - Asia. 109.
- McNeill V F 2015. Aqueous organic chemistry in the atmosphere: sources and chemical processing of organic aerosols. *Environ. Sci. Technol.*, 49: 1237-1244.
- Mladenov N, Alados-Arboledas L, Olmo Reyes F J, et al. 2009. Effects of aerosol collection and extraction procedures on the optical properties of water-soluble organic compounds [M]: B43D-0404.
- Mo Y, Li J, Cheng Z, et al. 2021. Dual carbon isotope-based source apportionment and light absorption properties of water-soluble organic carbon in PM_{2.5} over China. *J. Geophys. Res.: Atmos.*: e2020JD033920.
- Wang G H, Niu S L, Liu C, et al. 2002. Identification of dicarboxylic acids and aldehyde of PM₁₀ and PM_{2.5} aerosols in Nanjing, China. *Atmospheric Environment*, 36: 1941-1950.
- Weber R J, Sullivan A P, Peltier R E, et al. 2007. A study of secondary organic aerosol formation in the anthropogenic-influenced southeastern United States. *J. Geophys. Res.*, 112.
- Wozniak A S, Bauer J E, Dickhut R M 2012. Characteristics of water-soluble organic carbon associated with aerosol particles in the eastern United States. *Atmos. Environ.*, 46: 181-188.
- Wozniak A S, Bauer J E, Sleighter R L, et al. 2008. Technical Note: Molecular characterization of aerosol-derived water soluble organic carbon using ultrahigh resolution electrospray ionization Fourier transform ion cyclotron resonance mass spectrometry. *Atmos. Chem. Phys.*, 8: 5099-5111.
- Xu B, Cheng Z, Gustafsson Ö, et al. 2021. Compound-specific radiocarbon analysis of low molecular weight dicarboxylic acids in ambient aerosols using preparative gas chromatography: method development. *Environ. Sci. Technol. Lett.*, 8: 135-141.
- Yang H, Li Q, Yu J 2003. Comparison of two methods for the determination of water-soluble organic carbon in atmospheric particles. *Atmospheric Environment - ATMOS ENVIRON*, 37: 865-870.
- Yang L, Yu L E 2008. Measurements of Oxalic Acid, Oxalates, Malonic Acid, and Malonates in Atmospheric Particulates. *Environmental Science & Technology*, 42: 9268-9275.
- Yu Q, Chen J, Cheng S, et al. 2021. Seasonal variation of dicarboxylic acids in PM_{2.5} in Beijing: Implications for the formation and aging processes of secondary organic aerosols. *Science of The Total Environment*, 763: 142964.
- Zhao Y, Gao Y 2008. Mass size distributions of water-soluble inorganic and organic ions in size-segregated aerosols over metropolitan Newark in the US east coast. *Atmospheric Environment*, 42: 4063-4078.

(2). *the Results section includes considerable Discussion and is better headed as a Results and Discussion section. The Discussion section reads more like an “Implications for Atmospheric Cycling of Organics…” section.*

Response: Thanks for your great suggestion. We strongly agree with the point. Some section title has been revised for better summarization:

The revised title:

Results → Results and discussion (Line 93)

Discussion → Atmospheric implications (Line 406)

(3). *Line 145, change “typically” to “often”*

Response: Thanks for your comment. The word “typically” has been revised:

Emissions from different sources often have different end-member $\delta^{13}\text{C}$ values. (Line 151)

(4). *Line 273-5 – the authors note that “oxalic acid is mainly produced from fossil fuel-derived precursors…” But supplementary Figure 7 shows biological sources to be substantial during all sampling times – sometimes being higher than fossil fuel sources. Please reword this sentence to better characterize the data.*

Response: Thanks for your comment. We have reworded this sentence to better characterize the data:

The large contrast of fossil fraction among C_2 diacids and its higher homologs (i.e., C_3 and C_4 diacids) further indicate that substantial fossil precursors contribute to the formation of C_2 through aqueous-phase processing, which results in a much higher fossil content of C_2 than of its higher homologs. (Line 280-283)

(5). *Line 298 – the calculation $\text{fossil-C}_2 + \text{bb-C}_2 = \text{anthropogenic}$ assumes that all biomass burning is human-derived. It must be acknowledged that natural wildfires are a potential source of bb-C₂.*

Response: Thank you for your comments. We strongly agree that natural wildfires are a potential source of biomass-burning-derived oxalic acid which should be considered. However, the natural wildfires are generally considered negligible compared to anthropogenic biomass-burning

activities in China. For example, Zhou *et al.* (2017) reported that forest and grassland fires contributed 1.5% and 1.7% of VOC and PM_{2.5} in China, respectively. Per your great suggestion, we added discussion about the natural wildfires:

The revised sentence:

The anthropogenic C₂ (including fossil-C₂ and bb-C₂) accounted for 78% of total C₂ in the continental outflow samples, whereas its contribution is 39% in coastal background (Fig. 4d). Note that natural biomass burning (e.g., wildfires) was ignored since it contributed <4% of total biomass-burning-derived organic pollutants in China⁴⁷. (Line 306-310)

Reference:

47. Zhou Y, *et al.* A comprehensive biomass burning emission inventory with high spatial and temporal resolution in China. *Atmospheric Chemistry and Physics* **17**, 2839-2864 (2017).

(6). Lines 367-397 – this section represents a nice experiment demonstrating the spatial extent of the fossil C contributions; however, the data interpretation seems to be a bit lacking. The isotopic differences discussed in lines 372 to 382 are attributed to atmospheric aging which is one possibility due to the higher amounts of sunlight in summer. However, there are other potential reasons including different air mass back trajectories (the Heshan dataset showed these to vary drastically between summer and winter) and the fact that natural biomass inputs can be expected to increase during summer due to it being the growing season.

Response: Thanks for your comments. We fully agree with reviewer. We have modified the text accordingly by mentioning the stronger sunlight and enhanced biogenic inputs in summertime; and we further examined the $\delta^{13}\text{C}$ compositions of other diacids compounds from the same samples. We did not present these data in the manuscript owing to the length limit. However, as shown in **Figure R2-1** (unpublished data), the seasonal difference in $\delta^{13}\text{C}$ of longer-chain aliphatic diacids (i.e., C₃, C₄ and C₉ diacids) and phthalic acid (Ph) was minor (< 1.5‰), while those of oxalic acid and its aqueous-phase precursor glyoxylic acid reached 4.4‰. This shows that the seasonal variation of different air mass back trajectories is less likely to be responsible for the isotopic difference of oxalic acid.

The revised text:

Both aqSOA formation and aging can lead to the production of more polar compounds such as

oxalic acid. Oxalic acid was more enriched in ^{13}C (by 2.2‰) and ^{14}C ($f_{\text{bio/bb}}$ increased by 5.0%) than WSOC in summer, when there was stronger sunlight. This is probably due to that biomass/biogenic component from the WSOC pool is preferentially subjected to oxidative aging to small molecules^{12,59}, which results in oxalic acid enriched both in $\delta^{13}\text{C}$ and $\Delta^{14}\text{C}$. Note that natural biomass inputs can be expected to increase during summer. (Line 387-393)

Figure R2-1. Box-and-whisker plots of $\delta^{13}\text{C}$ values of five dicarboxylic acid (C₂, C₂, C₄, C₉, Ph) and one oxocarboxylic acids (ωC_2) in PM_{2.5} from five Chinese megacities from January 2018 (winter) and July 2018 (summer). Each box shows the median (middle line), the average (solid dot), the interquartile range (box), and the 10th and 90th percentile (the whiskers).

(7). Line 371 – change “minor” to “no”

Response: Thanks for your comment. The word “minor” has been revised:

The average $\delta^{13}\text{C}$ and ^{14}C -derived $f_{\text{bio/bb}}$ values of WSOC show no difference in winter and summer ($\delta^{13}\text{C}$: $-24.2 \pm 0.4\text{‰}$ vs. $-24.9 \pm 0.7\text{‰}$; $f_{\text{bio/bb}}$: $63 \pm 11\%$ vs. $60 \pm 9\%$) (Line 379)

(8). Lines 386-389 – the increase of ALW in winter is said to be important for oxalic acid depletions in winter, but no seasonal ALW data are presented. Supplementary text 2 suggests the data may be available. Referencing the ALW data to support the argument would be valuable.

Response: Thanks for your comment. The relative humidity (RH%) and the temperature in the specific sampling sites is not available. We actually used the average RH% and temperature in each city from the China National Environmental Monitoring center (<http://www.cnemc.cn/>). The aerosol

liquid water (ALW) content in each city is compiled in **Table R2-1**. In all cities except for Beijing, the ALW content significantly increased from summer to winter. In Beijing, the ALW content in winter was a little lower than in summer. Nevertheless, the aqueous-phase formation of oxalic acid ($\delta^{13}\text{C}$ depletion) would be significant compared to its photochemical aging ($\delta^{13}\text{C}$ enrichment) due to the low temperature and low oxidants level in Beijing winter (Yu *et al.*, 2019). Therefore, we think the depleted $\delta^{13}\text{C}$ -C₂ was also expected in Beijing winter. In addition, several previous studies also pointed out that aqueous-phase processes play a dominant role in the SOA formation in winter Beijing (*e.g.*, Xu *et al.*, 2017, Yu *et al.*, 2019, 2021, Wang *et al.*, 2021, Gkatzelies *et al.*, 2021).

However, on one hand, the average RH% and temperature of the city may not well represent the real meteorological parameters at the sampling sites. On the other hand, as the PM_{2.5} samples in one week were pooled for the ¹⁴C analysis (refer to **Line 495**), we can only roughly calculate the weekly averaged ALW content, which is weak to support the aqueous-phase processes. Hence, we did not include the ALW data of the five Chinese cities in the manuscript.

Table R2-1. The mean values of aerosol liquid water (ALW) content in five Chinese megacities from January 2018 (winter) and July 2018 (summer).

	ALW content ($\mu\text{g m}^{-3}$)	
	Winter	Summer
Beijing	5.34	8.36
Guangzhou	34.25	3.88
Wuhan	229.8	7.8
Chengdu	38.04	19.43
Shanghai	33.9	9.86

References:

- Gkatzelies G I, Papanastasiou D K, Karydis V A, et al. 2021. Uptake of water-soluble gas-phase oxidation products drives organic particulate pollution in Beijing. *Geophys. Res. Lett.*, 48: e2020GL091351.
- Wang J, Ye J, Zhang Q, et al. 2021. Aqueous production of secondary organic aerosol from fossil-fuel emissions in winter Beijing haze. *Proc. Natl Acad. Sci. USA*, 118: e2022179118.
- Xu W, Han T, Du W, et al. 2017. Effects of Aqueous-Phase and Photochemical Processing on Secondary Organic Aerosol Formation and Evolution in Beijing, China. *Environ Sci Technol*, 51: 762-770.
- Yu Q, Chen J, Cheng S, et al. 2021. Seasonal variation of dicarboxylic acids in PM_{2.5} in Beijing: Implications for the formation and aging processes of secondary organic aerosols. *Science of The Total Environment*, 763: 142964.

Yu Q, Chen J, Qin W, et al. 2019. Characteristics and secondary formation of water-soluble organic acids in PM1, PM2.5 and PM10 in Beijing during haze episodes. *Science of The Total Environment*, 669: 175-184.

(9). *Lines 389-391 – fossil content in water insoluble OA is referenced but no data have been presented for water insoluble OA. If it exists, it should be shown to back up the point being made.*

Response: Thanks for your comment. The fossil content in water insoluble OA has not been analyzed in this study. We would like to express that the overwhelming fossil contributions to aqSOA compounds observed in Heshan were consistent or even higher than the fossil content in water-soluble OA reported in previous publications. Per your good suggestion, we aware the confusing expression of this sentence. We have revised the text to express this point more clearly:

The revised sentence:

As mentioned above, overwhelming fossil contributions to aqSOA compounds were observed, that were consistent with or even higher than the fossil content in water-insoluble OA reported in Chinese urban areas during winter haze^{54,58}. (Line 397-399)

References:

54. Zhang Y-L, *et al.* Large contribution of fossil fuel derived secondary organic carbon to water soluble organic aerosols in winter haze in China. *Atmos. Chem. Phys.* **18**, 4005-4017 (2018).

58. Zhang Y-l, *et al.* Micro-scale (μg) radiocarbon analysis of water-soluble organic carbon in aerosol samples. *Atmos. Environ.* **97**, 1-5 (2014).

(10). *Lines 412-439 – the authors present a positive feedback loop among hygroscopic particles, ALW, and aqSOA formation that does not connect exceptionally well to the Results presented already. A clearer reference to how the data gathered in this paper point to the existence of the feedback loop is needed.*

Response: Thanks for your comment. Accordingly, we have revised the discussion text, and added a supplementary figure (**Supplementary Fig.5**) and more references to support the feedback loop:

The revised text:

A mutual promotion effect may take place between ALW and inorganic particles (e.g., sulfate and nitrate)^{31,64}. Our year-round observation at the Heshan receptor site witnesses a positive

feedback loop among ALW, inorganics, as well as aqSOA compounds. The dataset supports that the inorganic particles induce ALW growth (Supplementary Fig. 5), facilitating the partitioning of gas-phase oxidation products into aqueous-phase medium, and promoting the formation of aqSOA (Fig. 3b)^{6,30}. The formation of aqSOA in the organic aerosol fraction would decrease aerosol viscosity and increase ALW⁶⁵, as in turn promotes further gas-to-liquid transfer of water-soluble organic precursors^{23,32}. Oxalic acid is identified as the upper bound of the hygroscopic properties of OA, with a *k*-Köhler of 0.48⁷. Using the Zdanovskii–Stokes–Robinson (ZSR) mixing model⁶⁶, the effect of anthropogenic oxalic acid on aerosol water accounts for 3–38% (average: 10%) of the ALW contributed by organic compounds (Supplementary Text 2). The inorganic particles can be considered anthropogenic in most continental atmosphere^{3,30,41}. We provide ¹⁴C-based evidence that, besides inorganic species, substantial aqSOA compounds are also derived from anthropogenic precursors. Therefore, a broad control of various anthropogenic emissions such as SO₂, NO_x, and VOC precursors is vital for reducing the organic particulate pollution. (Line 416-433)

Supplementary Figure 5. Relationships between (a) aerosol liquid water content and anthropogenic water-soluble inorganic constituents (WSIC_{anth}); (b) aerosol liquid water

content and WSIC_{anth} to PM_{2.5} mass ratio.

Figure 3b. Relationships between oxalic acid mass concentration and aerosol liquid water.

References:

- McNeill VF. Aqueous organic chemistry in the atmosphere: sources and chemical processing of organic aerosols. *Environ. Sci. Technol.* **49**, 1237-1244 (2015).
- Hennigan CJ, Bergin MH, Dibb JE, Weber RJ. Enhanced secondary organic aerosol formation due to water uptake by fine particles. *Geophys. Res. Lett.* **35**, (2008).
- Gilardoni S, *et al.* Direct observation of aqueous secondary organic aerosol from biomass-burning emissions. *Proc. Natl Acad. Sci. USA* **113**, 10013-10018 (2016).
- Wang J, *et al.* Enhanced aqueous-phase formation of secondary organic aerosols due to the regional biomass burning over North China Plain. *Environ. Pollut.* **256**, 113401 (2020).
- Hodas N, *et al.* Aerosol liquid water driven by anthropogenic nitrate: implications for lifetimes of water-soluble organic gases and potential for secondary organic aerosol formation. *Environ. Sci. Technol.* **48**, 11127-11136 (2014).
- Wu Z, *et al.* Aerosol liquid water driven by anthropogenic inorganic salts: implying its key role in haze formation over the North China Plain. *Environ. Sci. Technol. Lett.* **5**, 160-166 (2018).
- Gkatzelis GI, *et al.* Uptake of water-soluble gas-phase oxidation products drives organic particulate pollution in Beijing. *Geophys. Res. Lett.* **48**, e2020GL091351 (2021).
- Carlton AG, Turpin BJ. Particle partitioning potential of organic compounds is highest in the Eastern US and driven by anthropogenic water. *Atmos. Chem. Phys.* **13**, 10203-10214 (2013).
- Wang Y, *et al.* Mutual promotion between aerosol particle liquid water and particulate nitrate enhancement leads to severe nitrate-dominated particulate matter pollution and low visibility. *Atmos. Chem. Phys.* **20**, 2161-2175 (2020).
- Shiraiwa M, Ammann M, Koop T, Pöschl U. Gas uptake and chemical aging of semisolid organic aerosol particles. *Proc. Natl Acad. Sci. USA* **108**, 11003-11008 (2011).
- Nguyen TKV, Zhang Q, Jimenez JL, Pike M, Carlton AG. Liquid water: ubiquitous contributor to

aerosol mass. *Environ. Sci. Technol. Lett.* **3**, 257-263 (2016).

(11). In lines 426-432, there is a discussion of how nitrate represents the hygroscopic particles of interest, but this is the first mention of nitrate data in the paper. If hygroscopicity is going to factor into the conclusions as it does here, nitrate and hygroscopicity should be presented much earlier.

Response: Thanks for your comment. In the “**Meteorological setting and aerosol molecular compositions**” section, we have added more text on the hygroscopicity of nitrate before the argument discussion. A new figure (supplementary Fig.5) which shows the relationship between aerosol liquid water and anthropogenic water-soluble inorganic constituents (WSIC_{anth}) has been added to “Supporting material”.

The revised manuscript:

ALW mass was driven primarily by inorganic salts, especially by nitrate and sulfate³¹. As shown in Supplementary Fig.5, the ALW mass well correlated with the concentrations of WSIC_{anth} and fractions of WSIC_{anth} in PM_{2.5} during the campaign. The nitrate in the continental outflow samples display concentrations 20 times higher than the coastal background samples, and account for nearly half of the anthropogenic WSIC (Supplementary Fig.4). Because nitrate is more hygroscopic than sulfate³⁰, such changes in aerosol compositions would be a dominant factor driving the ALW content. Given the high relative humidity (RH%) ($74 \pm 9\%$; Supplementary Table 2) and the abundant hygroscopic particles (particularly for particulate nitrate) in continental outflow samples, we expect that the changes in the gas-particle partitioning driven by the increased ALW could have enhanced the formation of aqSOA species, such as oxalic acid^{23, 26, 32}. (Line132-143)

Supplementary Figure 5. Relationships between (a) aerosol liquid water content and anthropogenic water-soluble inorganic constituents (WSIC_{anth}); (b) aerosol liquid water content and WSIC_{anth} to PM_{2.5} mass ratio.

Reference:

- 23. Wang J, *et al.* Enhanced aqueous-phase formation of secondary organic aerosols due to the regional biomass burning over North China Plain. *Environ. Pollut.* **256**, 113401 (2020).
- 26. Bikkina S, Kawamura K, Sarin M. Secondary organic aerosol formation over coastal ocean: inferences from atmospheric water-soluble low molecular weight organic compounds. *Environ. Sci. Technol.* **51**, 4347-4357 (2017).
- 30. Hodas N, *et al.* Aerosol liquid water driven by anthropogenic nitrate: implications for lifetimes of water-soluble organic gases and potential for secondary organic aerosol formation. *Environ. Sci. Technol.* **48**, 11127-11136 (2014).
- 31. Wu Z, *et al.* Aerosol liquid water driven by anthropogenic inorganic salts: implying its key role in haze formation over the North China Plain. *Environ. Sci. Technol. Lett.* **5**, 160-166 (2018).
- 32. Gkatzelis GI, *et al.* Uptake of water-soluble gas-phase oxidation products drives organic particulate pollution in Beijing. *Geophys. Res. Lett.* **48**, e2020GL091351 (2021).

(12). A discussion of how relevant these results are to the rest of the world is warranted. The study site receives air from some of the highest fossil fuel emission areas on the globe. Are there other locations of interest?

Response: Thank you! We strongly agree the point. A global relevance analysis would be very interesting. We have added it to the last section:

The revised text:

The present findings highlight the relevance of aqueous-phase chemistry for processing of fossil-fuel emissions over China. These scenarios are not unique to East Asia and could be relevant in other high fossil-fuel consumption regions with humid weather. In a study in northern Georgia of the US, Weber et al.¹³ observed strong correlations between WSOC and anthropogenic precursors, while in contrast the radiocarbon analysis revealed dominant biogenic contributions to WSOC. Our results suggest that aqSOA do not necessarily share the same source patterns with bulk WSOC. WSOC have a large primary contribution from biomass burning⁶⁸, which would mask the fossil content in aqSOA. Therefore, the fossil contributions to aqSOA would be much higher than expected in locations such as North America, Europe, and South Asia where the aqueous-phase chemistry pathway has been demonstrated significant^{7,26,41}. Even in pristine environments where biogenic emissions are dominant, the importance of anthropogenic emissions to aqSOA formation would also be significant since the ALW are largely anthropogenic^{2,41}. (Line 448-461)

Reference:

2. Carlton AG, Christiansen AE, Flesch MM, Hennigan CJ, Sareen N. Multiphase Atmospheric Chemistry in Liquid Water: Impacts and Controllability of Organic Aerosol. *Acc. Chem. Res.* **53**, 1715-1723 (2020).
7. Gilardoni S, *et al.* Direct observation of aqueous secondary organic aerosol from biomass-burning emissions. *Proc. Natl Acad. Sci. USA* **113**, 10013-10018 (2016).
13. Weber RJ, *et al.* A study of secondary organic aerosol formation in the anthropogenic-influenced southeastern United States. *J. Geophys. Res. Atmos* **112**, (2007).
26. Bikkina S, Kawamura K, Sarin M. Secondary organic aerosol formation over coastal ocean: inferences from atmospheric water-soluble low molecular weight organic compounds. *Environ. Sci. Technol.* **51**, 4347-4357 (2017).
41. Carlton AG, Turpin BJ. Particle partitioning potential of organic compounds is highest in the Eastern US and driven by anthropogenic water. *Atmos. Chem. Phys.* **13**, 10203-10214 (2013).
68. Mayol-Bracero OL, *et al.* Water-soluble organic compounds in biomass burning aerosols over Amazonia 2. Apportionment of the chemical composition and importance of the polyacidic fraction. *J. Geophys. Res. Atmos* **107**, LBA 59-51-LBA 59-15 (2002).

Tables and Figures:

(13). In Supplementary Figures 2 and 3, the air mass back trajectories are characterized differently than they are in the text. Please change for consistency.

Response: Thanks for your suggestion. The characterizations of the air mass back trajectories in Supplementary Figures 2 and 3 have been revised to be consistent with the text:

The revised figure legends in supporting material:

Supplementary Figure 2. Three-day backward trajectories (BTs) arriving at an altitude of 100 m over Heshan for every 6h. Based on BTs, aerosol samples are categorized into two major transport pathways: (a) Coastal background, (b) Continental outflow

Supplementary Figure 3. Fractional contribution of air mass clusters in each aerosol sample during the sampling campaign (Jun 2017 to May 2018). Red column represents the contribution of air mass originate from the South China Sea (coastal background), whereas blue column represents the contribution of air mass originate from Chinese continental (continental outflow).

(14). In Figures 2, 4, and 6, the fraction of non-fossil C is plotted as increasing on the y-axis, while in Figure 5 and Supplemental figures 5 and 6 the fraction of fossil C is plotted as increasing on the y-axis. This is confusing. I suggest using the same convention for all plots quantifying fossil or non-fossil C. My own preference is to have the fraction of fossil C increase on the y-axis since the paper's focus is demonstrating fossil C contributions to aqSOA.

Response: Thank you for your careful reminding. We strongly agree your point. In Figures 2 and 6, plotting non-fossil C increase on the y-axis would give better visual effects, because the same variation trend of ^{13}C and ^{14}C compositions would be clearer than fossil C increase. We have thus drawn all the plots in such way. We have revised Figure 5 to ensure the same convention for quantifying ^{14}C data was used in the manuscript. Please refer the revised Figure 5 as follows:

Figure 5. Two-dimensional dual-carbon isotope characterization of individual oxalic acid for coastal background and continental outflow. The color of the symbols represents the oxalic acid concentration. The $\delta^{13}\text{C}$ source signatures of oxalic acid precursor gases from biogenic VOCs (BVOCs; i.e., isoprene) and from anthropogenic VOCs (AVOCs; i.e., anthropogenic emitted non-methane hydrocarbons) are obtained from reported literature values^{37,38}. The two ovals include the

coastal background samples and the majority of the continental outflow samples, respectively. The shadowed pink circle remarks the outliers of continental samples subjected to strong atmospheric aging. (Line 812)

(15). In Supplementary Table 2, the WSON values are suspiciously high (higher than WSOC in the coastal background samples). The description in Supplementary Text 1 describes the measurement of total dissolved nitrogen (which would include inorganic species). Typically, WSON is calculated as follows: $WSON = TDN - (nitrate + nitrite + ammonium)$. Is this what was done?

Response: Many thanks for your reminding. The WSON was calculated by $TDN - WSIN$ (nitrate+ ammonium). However, we did not clearly check the TDN data because the TDN data was not used in the manuscript. We carefully looked back to the original data, and found mistakes in the TDN calibration curve. We have recalculated the WSON data as showed in **Table R2-2**. The WSON values in **Supplementary Table 2** have been revised.

Table R2-2. The concentration of water-soluble total nitrogen (WSTN), water-soluble inorganic nitrogen (WSIN), and water-soluble organic nitrogen (WSON).

	Coastal background (N = 11)		Continental outflow (N = 21)	
	min-max	Mean/SD	min-max	Mean/SD
TDN	0.3-1.65	0.83±0.46	1.4-14.4	5.3±3.0
NO ₃ ⁻	0.1-1.7	0.5±0.5	0.4-34.1	8.7±8.5
NH ₄ ⁺	0.4-1	0.6±0.2	0.9-6.7	2.6±1.5
NO ₃ ⁻ -N	0.02-0.39	0.11±0.11	0.1-7.7	1.9±1.9
NH ₄ ⁺ -N	0.28-0.76	0.45±0.16	0.7-5.2	1.9±1.2
WSON	0.00-0.60	0.28±0.21	0.1-4.3	1.4±1.0
WSOC	0.7-1.6	1.2±0.3	1.9-9.6	4.5±1.8

Responses to reviewer #3:

In this manuscript, the authors describe chemical characterization and mass measurements of secondary organic aerosol formed via aqueous pathways (aqSOA) during two contrasting phases of the East Asia monsoon system. A novel and important attribute of this work is the simultaneous measurement of both $\delta^{13}\text{C}$ - $\Delta^{14}\text{C}$ isotopes, which provides insight to the parent organic gas and formation process. The authors find compelling evidence that fossil carbon contributes substantially to water-soluble SOA mass. This work contributes scientifically by providing insight to atmospheric transport and processing of trace species, in addition to the policy-relevant finding demonstrating the contribution of fossil carbon to aqSOA.

The contribution of fossil carbon to aqSOA is largest in continental air masses, and not in the marine derived air. This is clear in the presented graphs and is consistent

Response: Thank for your nice summary of our paper and positive assessment of the importance of this work. We have carefully revised the manuscript following your comments and suggestions. Our responses to your comments are given below, and the revised parts are shown after the responses (in blue font). Please refer to the revised manuscript, in which changes are highlighted in yellow.

General comments:

*(1). This work is noteworthy. **The last section of the submitted manuscript is not constructed as carefully as other parts.** I think this work should be published, but some revision is required. I list detailed comments below.*

Response: Thanks for your comments. We think the construction of the last section may have three expression problems. First, the section title “**Discussion**” did not well summarize the content. Second, some of the discussions were repetitive and lengthy. Third, some discussion looks insufficient. Therefore, we have thoroughly rewritten the last section. Please refer to the revised manuscript, in which changes are highlighted in yellow.

The revised/rewritten text:

Atmospheric implications

The large contribution of fossil-derived carbon to aqSOA highlighted in this study has

important implications for aerosol climate forcing and regional air quality. Aqueous-phase processing of fossil precursors, competitive to gas-phase pathway, could amplify the impact of anthropogenic emissions on water-soluble OA, alter the chemical property and hygroscopicity of aerosols. The derived hygroscopic SOA affect not only direct radiative forcing, but also cloud condensation nuclei (CCN) activity, and the subsequent cloud-mediated effect of anthropogenic aerosols on the climate system⁶⁰. As for oxalic acid, its interactions with inorganic salts in the aqueous phase may modify the CCN activity of aerosol particle⁶¹⁻⁶³.

A mutual promotion effect may take place between ALW and inorganic particles (e.g., sulfate and nitrate)^{31,64}. Our year-round observation at the Heshan receptor site witnesses a positive feedback loop among ALW, inorganics, as well as aqSOA compounds. The dataset supports that the inorganic particles induce ALW growth (Supplementary Fig. 5), facilitating the partitioning of gas-phase oxidation products into aqueous-phase medium, and promoting the formation of aqSOA (Fig. 3b)^{6,30}. The formation of aqSOA in the organic aerosol fraction would decrease aerosol viscosity and increase ALW⁶⁵, as in turn promotes further gas-to-liquid transfer of water-soluble organic precursors^{32,32}. Oxalic acid is identified as the upper bound of the hygroscopic properties of OA, with a *k*-Kohler of 0.48⁷. Using the Zdanovskii–Stokes–Robinson (ZSR) mixing model⁶⁶, the effect of anthropogenic oxalic acid on aerosol water accounts for 3-38% (average: 10%) of the ALW contributed by organic compounds (Supplementary Text 2). The inorganic particles can be considered anthropogenic in most continental atmosphere^{3,30,41}. We provide ¹⁴C-based evidence that, besides inorganic species, substantial aqSOA compounds are also derived from anthropogenic precursors. Therefore, a broad control of various anthropogenic emissions such as SO₂, NO_x, and VOC precursors is vital for reducing the organic particulate pollution.

Nitrate is the dominant atmospheric hygroscopic particles in China^{30,31,43}. A model study predicted that 92% of increased ALW in eastern China by the year 2100 can be attributed to the increase in anthropogenic nitrate aerosol³⁰. Here, the average nitrate concentration in the continental outflow samples was nearly 20 times higher than coastal background (Supplementary Table 2). Thus, this nitrate enhancement in the particulate phase could be the dominant factor that facilitated the increase of ALW in the continental outflow regime. Furthermore, there has been a rapid transition from coal combustion to natural gas in China. However, natural gas combustion can produce more than three times as much water vapor as coal burning⁶⁷. The combustion-derived water was

interpreted to constitute 6.2% of the atmospheric moisture and added 5.1% of the anthropogenic PM_{2.5} in northwestern China⁶⁷. Therefore, the enhancing effects of particulate nitrate and combustion-derived water vapor on aqSOA formation should be addressed when changing energy structure in future climate and air quality scenarios.

The present findings highlight the relevance of aqueous-phase chemistry for processing of fossil-fuel emissions over China. These scenarios are not unique to East Asia and could be relevant in other high fossil-fuel consumption regions with humid weather. In a study in northern Georgia of the US, Weber et al.¹³ observed strong correlations between WSOC and anthropogenic precursors, while in contrast the radiocarbon analysis revealed dominant biogenic contributions to WSOC. Our results suggest that aqSOA do not necessarily share the same source patterns with bulk WSOC. WSOC have a large primary contribution from biomass burning⁶⁸, which would mask the fossil content in aqSOA. Therefore, the fossil contributions to aqSOA would be much higher than expected in locations such as North America, Europe, and South Asia where the aqueous-phase chemistry pathway has been demonstrated significant^{7,26,41}. Even in pristine environments where biogenic emissions are dominant, the importance of anthropogenic emissions to aqSOA formation would also be significant since the ALW are largely anthropogenic^{2,41}.

In this study, oxalic acid was assessed to be dominantly produced from aqueous-phase processes. However, it should be noted that both biomass burning and gas-phase photochemical aging could also be sources for oxalic acid. Therefore, cautious interpretations of oxalic acid isotopic signals are needed where there are intensive biomass burning, extensive atmospheric aging, and seasonality of C3/C4 vegetation changes. Future dual-carbon isotope studies on other aqSOA constituents (e.g., glyoxal) are strongly warranted to better constrain the chemical mechanism responsible for fossil-derived aqSOA formation. Overall, novel compound-specific dual carbon isotopic evidence from field observations show that fossil anthropogenic precursors contribute to the formation of aqSOA to a much higher extent than expected. Atmospheric aqueous-phase processes are expected to increase due to the enhanced evapotranspiration in a warmer world⁶⁶. An absence of accounting for such aqueous processing incurred by fossil precursors could lead to an underestimation of the anthropogenic contribution to organic aerosols. Understanding the role of anthropogenic emissions in aqSOA formation and the chemical mechanisms involved are of critical concern for future climate and air quality projections, and within the context of energy needs and

choices. (Line 406-478)

(2). *I largely buy the authors' argument that kinetic isotope effects (KIE) dominate the $\delta^{13}C$ composition of oxalic acid – but think a comment on what the possible impacts on interpretation are given limitations of this assumption is needed.*

Response: Thanks for your comments. We have added paragraphs addressing the limitations to the last section of the manuscript:

The revised text:

In this study, oxalic acid was assessed to be dominantly produced from aqueous-phase processes. However, it should be noted that both biomass burning and gas-phase photochemical aging could also be sources for oxalic acid. Therefore, cautious interpretations of oxalic acid isotopic signals are needed where there are intensive biomass burning, extensive atmospheric aging, and seasonality of C3/C4 vegetation changes. Future dual-carbon isotope studies on other aqSOA constituents (e.g., glyoxal) are strongly warranted to better constrain the chemical mechanism responsible for fossil-derived aqSOA formation. (Line 462-469)

(3). *Line 141: “With the addition of aqSOA formation, the higher WSOC/OC ratios in the continental outflow samples may thus be explained.” This sentence is overstated. The evidence is consistent but not incontrovertible truth. I suggest rephrasing to something along the lines of:*

Observation in continental outflow of higher GLY/MeGLY and higher [WSOC]/[OC] is consistent with aqSOA formation processes.

Response: Thanks for your comments. Per your good suggestion, we have revised this sentence:

The revised sentence:

Observation in continental outflow of higher aqSOA precursor concentrations and higher WSOC/OC ratios is consistent with aqSOA formation processes. (Line 145-147)

(4). *Line 203: The authors state “...rather than photochemical breakdown of longer-chain diacids.” I think the point that aqueous-phase processing is much more like than gas-phase photochemistry*

is an important point. I think the authors should amend the sentence so that it “... reads rather than gas-phase photochemical ...”

Response: Thanks for your comment. We have revised this sentence to emphasize the gas-phase photochemical processes.

The revised sentence:

Therefore, the vast difference in the $\delta^{13}\text{C}$ composition of oxalic acid between the two air mass source regimes could be mainly resulted from the aqueous-phase reaction pathway, i.e., AVOCs/BVOCs \rightarrow SVOCs \rightarrow Pyr \rightarrow $\omega\text{C}_2 \rightarrow \text{C}_2^{42}$, rather than gas-phase photochemical oxidation (or breakdown) processes. (Line 211-212)

(5). Line 291: “These evidences confirm ...” Is **overstated**. Independent lines of evidence are suggestive of and consistent with background C_2 ...”

Response: Thanks for your comment. We have revised this sentence to better interpret the evidence:

The revised sentence:

These independent lines of evidence are suggestive of and consistent with that background C_2 were biogenic, while $bb\text{-C}_2$ likely represent additional non-fossil C_2 concentrations. (Line 299-301)

(6). Line 312: The authors state: “ However, the oxidation products of fossil precursors are largely water- soluble” I agree that a large fraction of atmospheric organic gases are water-soluble. Perhaps to help drive home this point, the sequence on line 203 could be altered to list both SVOC and WSOC (water-soluble organic carbon).

Response: Many thanks! We have modified the SVOCs to both SVOCs/WSOC in Line 209.

The revised sentence:

Therefore, the vast difference in the $\delta^{13}\text{C}$ composition of oxalic acid between the two air mass source regimes could be mainly resulted from the aqueous-phase reaction pathway, i.e., AVOCs/BVOCs \rightarrow SVOCs/WSOC \rightarrow Pyr \rightarrow $\omega\text{C}_2 \rightarrow \text{C}_2$ (Line 211-212)

(7). The Figure 5 caption should include concise explanation for ovals and circle.

Response: Thanks for your comment. This information has been added to the Figure 5 caption:

The revised manuscript:

The two ovals include the coastal background samples and the majority of the continental outflow samples, respectively. The shadowed pink circle remarks the outliers of continental samples subjected to strong atmospheric aging. (Line 818-821)

(8). Starting at line 343: “...water-soluble OA is only to very limited extent stemming from fossil sources.” The phrasing is awkward and should be re-worded.

Response: Thanks for your suggestion. This sentence has been rewritten:

The revised sentence:

Hence, these earlier field and laboratory studies show that water-soluble OA is only to very limited extent stemmed from fossil sources. (Line 351-353)

(9). Starting at Line 382: “This is probably due to that biomass/biogenic component from the WSOC pool is preferentially subjected to oxidative aging to small molecules”. This sentence is awkward and needs a little work.

Response: Thanks for your suggestion. This sentence has been rewritten:

The revised sentence:

Oxalic acid was more enriched in ^{13}C (by 2.2‰) and ^{14}C ($f_{\text{bio/bb}}$ increased by 5.0%) than WSOC in summer, when there was stronger sunlight. This is probably due to that biomass/biogenic component from the WSOC pool is preferentially subjected to oxidative aging to small molecules^{12,59}, which results in oxalic acid enriched both in $\delta^{13}\text{C}$ and $\Delta^{14}\text{C}$. Note that natural biomass inputs can be expected to increase during summer. (Line 388-393)

(10). Starting at Line 403: “Condensation of oxalic acid and its salts in the aqueous phase may significantly impact the CCN activity of aerosol particle. “ The word “condensation” in this sentence seems at odds with the authors’ main premise – that oxalic acid often forms in the aqueous-phase. The sentence reads as if it describing the importance of the gas-phase formation pathway. Perhaps “condensation” could be replaced with “presence” .

Response: Thanks for your comment. We have revised this sentence:

The revised text:

As for oxalic acid, its interactions with inorganic salts in the aqueous phase may modify the CCN activity of aerosol particle⁶¹⁻⁶³. (Line 414-415)

Reference:

61. Prenni AJ, *et al.* The Effects of Low Molecular Weight Dicarboxylic Acids on Cloud Formation. *J. Phys. Chem. A* **105**, 11240-11248 (2001).
62. Jing B, *et al.* Hygroscopic behavior of multicomponent organic aerosols and their internal mixtures with ammonium sulfate. *Atmos. Chem. Phys.* **16**, 4101-4118 (2016).
63. Boreddy SKR, Kawamura K. Investigation on the hygroscopicity of oxalic acid and atmospherically relevant oxalate salts under sub- and supersaturated conditions. *Environ. Sci.: Processes Impacts* **20**, 1069-1080 (2018).

REVIEWER COMMENTS

Reviewer #1 (Remarks to the Author):

The authors provided thorough responses to the reviewers' comments. However, the authors did not address a key issue of whether oxalic acid can be used as an aqueous-phase SOA (aqSOA) tracer, which is the basis of this study. First, one should not mix up the aqueous-phase reaction and condensed phase reaction, especially for ambient measurements these two processes are very difficult to differentiate. In laboratory studies or in specific atmospheric environments (e.g., cloud processes), oxalic acid was indeed used to investigate aqSOA formation process of e.g., glyoxal, like in many previous publications the authors mentioned. However, in ambient air, it is much more complex and this aqueous-phase process is only one of formation pathways for oxalic acid. To my knowledge, there is no defined aqueous-phase tracer in real ambient air so far, although many colleagues suggested organosulfates could be the most promising tracer of aqueous-phase SOA. Second, photochemical oxidation could be still an important formation pathway for oxalic acid in Heshan city, located in south China where photochemical is often prevailing, resulting a mix of aqueous-phase/condensed phase processes and photochemical oxidation. Third, the primary emission of oxalic acid. As specified in the response, the authors measured oxalic acid in source emission samples, i.e., 157 ng m⁻³ for biomass burning and 78 ng m⁻³ for coal combustion, which are about 30% of the average ambient concentration (446 ng m⁻³). It should be noted that oxalic acid concentration in source emission aerosols is dependent on the mass of fuel burned in the experiments (i.e., 1–2 kg of biomass per burn, 1 kg of coal per burn; Tang et al., 2020). Biomass burning is a very important emission source in south China, from both local emission and regional transport depending on the origin of air masses. Therefore, I would consider what the authors measured in this study is a combining result from primary emissions and secondary formation (aqueous-phase/condensed phase processes + photochemical oxidation). This cannot support what the authors claimed “the concentration of oxalic acid from primary sources would be very minor compared to the atmospheric oxalic acid pool”.

Taken together, it is not convincing and even danger to say oxalic acid the authors measured in Heshan city was from aqueous-phase process.

Some more specific comments:

1. The authors tried to demonstrate that “the $\delta^{13}\text{C}$ composition of C3 plant-derived isoprene is indeed close to the $\delta^{13}\text{C}$ of AVOCs,” so that “kinetic isotopic fractionation the most probable processes controlling the $\delta^{13}\text{C}$ of oxalic acid” (Line 172-178). From the response, it is still not clear why “the $\delta^{13}\text{C}$ of isoprene (-29‰ to -26‰) suggested by Iannone et al. [2007] could represent the biogenic source of isoprene in our study”. The $\delta^{13}\text{C}$ value of -26‰ for isoprene is at the higher end of the range summarized in Table R1-1. Similarly, the $\delta^{13}\text{C}$ of anthropogenic VOCs (AVOCs; e.g., aromatic hydrocarbons: -27.7‰ \pm 1.7‰), is taken from the reference 37, which is specific for transportation. I believe at this receptor site, the coal combustion (more enriched in $\delta^{13}\text{C}$ than vehicular emissions) and biomass burning (in Line 206, it says clearly that anthropogenic sources include biomass burning) are also non-negligible sources of anthropogenic VOCs, especially during the continental outflow. The authors should fully consider the

$\delta^{13}\text{C}$ of anthropogenic VOCs (from transportation, coal combustion, also biomass burning) and $\delta^{13}\text{C}$ of C3-plant derived biogenic VOCs, and discuss whether the two are similar, because this is the basis of conclusions made from the $\delta^{13}\text{C}$ of oxalic acid, including conclusions relating to possible formation pathway of aqSOA.

In Figure 5, during the continental outflow with important influence of aqSOA formation, the $\delta^{13}\text{C}$ of oxalic acid ranges from -31‰ to -21‰, with some datapoints fall in the range of $\delta^{13}\text{C}$ of anthropogenic VOCs and biogenic VOCs. Following the discussion in Line 204-206, “As aqSOA formed from VOCs or SVOCs typically leads to lower $\delta^{13}\text{C}$ values relative to the precursors since lighter-isotope containing precursors are preferentially oxidized to form reaction products”, more depleted (lower) $\delta^{13}\text{C}$ values than $\delta^{13}\text{C}$ of anthropogenic VOCs and biogenic VOCs (around -29‰ to -26‰ in Figure 5) are expected, however, Figure 5 shows the opposite, with the most of continental outflow samples falls in the range -29‰ to -21‰.

2. The correlation between fossil-C2 and C2 concentrations for continental outflow is weak. And there is very weak correlation for coastal background (Supplementary Figure 6). The concentrations of C2 oxalic acid and fossil-C2 for continental outflow are larger than those for coastal background; however, it is not reasonable to pool the data of continental outflow and coastal background together, and conclude that “The concentrations of C2 oxalic acid increased with the increase of fossil-C2”. In the Supplementary Figure 6, the increased concentrations and fossil-C2 are found in the continental outflow, but the datapoints for continental outflow are scattered with no obvious correlations. For coastal background, the oxalic acid concentrations were low, and the non-fossil sources contributed more strongly to oxalic acid (i.e., fossil-C2 <50%). Therefore, the argument “The concentrations of C2 oxalic acid increased with the increment of fossil-C2 ($r^2=0.52$, $P<0.001$; Supplementary Fig. 5), suggesting that the precursors from fossil-fuel emission was the dominant factor in controlling the ambient abundance of oxalic acid” (Line 247-250) is an overinterpret of the data.

3. The argument “oxalic acid was more enriched in ^{13}C and ^{14}C than WSOC in summer” (Line 380-382) is not true for ^{14}C in Beijing (BJ) and Chengdu (CD), challenging again the validation to extend to wider spatial coverage of China. The spatial extent of the findings at Heshan site is further challenged by Line 378-380 “The average $\delta^{13}\text{C}$ and ^{14}C -derived fbio/bb values of WSOC show no difference in winter and summer ($\delta^{13}\text{C}$: $-24.2 \pm 0.4\text{‰}$ vs. $-24.9 \pm 0.7\text{‰}$; fbio/bb: $63 \pm 11\%$ vs. $60 \pm 9\%$)”, is not true for each city, e.g., in Chengdu, $\delta^{13}\text{C}$: -23.8‰ vs. -25.9‰ , fbio/bb: 78 % vs. 62 %; in Beijing, fbio/bb: 60 % vs. 45 %; Shanghai: fbio/bb: 49 % vs. 62 %, and the seasonal variation of fbio/bb of WSOC is different in different cities, in Wuhan and Shanghai, fbio/bb of WSOC was higher in summer, and in Beijing and Chengdu, it shows the opposite.

4. Line 350-353: “ ^{14}C analyses of aerosol OC and its sub-fractions have been interpreted such that the majority of fossil-derived OA was water-insoluble 53,54”. However, in fact, ^{14}C analysis interprets the majority of primary fossil-derived OA was water-insoluble (see reference 54). Therefore, the conclusion “Hence, these earlier field and laboratory studies show that water-soluble OA is only to very limited extent stemmed from fossil sources” is not always true. And it is contrary to what the authors state in the Introduction: “Radiocarbon-based estimations attributed ~50% of water-soluble OA to fossil sources in the East Asia outflow” (Line 61-62). Previous studies of direct ^{14}C measurement on water-soluble OC found that large fossil contributions to water-soluble OC, e.g., ~50% in winter in Beijing (reference 54),

~30% in autumn in Xi'an (Pavuluri et al., 2013), ~50% in summer in Guangzhou (Liu et al., 2018), ~50% of Chinese outflow in spring 2011 in Jeju Island (Fang et al., 2017).

Pavuluri CM, Kawamura K, Uchida M, Kondo M, Fu P. Enhanced modern carbon and biogenic organic tracers in Northeast Asian aerosols during spring/summer. *Journal of Geophysical Research: Atmospheres* 2013, 118(5): 2362-2371.

Liu J, Ding P, Zong Z, Li J, Tian C, Chen W, et al. Evidence of Rural and Suburban Sources of Urban Haze Formation in China: A Case Study From the Pearl River Delta Region. *Journal of Geophysical Research: Atmospheres* 2018, 123(9): 4712-4726.

Fang W, Andersson A, Zheng M, Lee M, Holmstrand H, Kim S-W, et al. Divergent Evolution of Carbonaceous Aerosols during Dispersal of East Asian Haze. *Scientific Reports* 2017, 7(1): 10422

Reviewer #2 (Remarks to the Author):

With their revision of "Large contribution of fossil-derived components to aqueous secondary organic aerosols in China," Xu et al. have answered the majority of my concerns with the initial submission. The data appear to show evidence for the aqueous phase production of secondary organic aerosols (SOA) from fossil-derived carbon which, to my knowledge, is a novel finding within the aerosol literature. I suggest that the paper be accepted for publication. Below, I note a few outstanding comments for the authors to consider prior to publication.

- Ultrasonication - the experimental data presented to demonstrate that ultrasonication does not effect the concentrations and isotopic signatures of WSOC and oxalic acid was convincing. The prior argument that it has been used extensively in the literature as evidence for the validity of the extraction technique was not at all convincing, however, and I caution the authors against using this type of argument. Rather, the experimental data are important for validating the lack of effects in these previous studies.

- lines 375-393 - The authors argue that seasonal differences in different compound classes (longer chain diacids) are evidence for aqueous processing rather than for different trajectories, but I'm missing something in their argument. It seems to me that there remains the possibility that the differences in the isotopic signatures could result from differing isotopic signatures of the precursor compounds coming from the two different source regions. Throughout the paper, the authors argue convincingly that the WSOC isotopic signature is not necessarily diagnostic for the aqSOA isotopic signature because the WSOC isotopic signature is a mixture of many different components with different isotopic signatures. With this argument in lines 375-393 though, they seem to suggest that the components do all start with the same signature before becoming altered via aqSOA, and I'm having a hard time

accepting that. I think it's fine for the authors to suggest that the differences are due to aqSOA, but I would like to see a better argument against the oxalic acid and Pyr precursors coming from the different trajectories/source regions simply having different isotopic signatures (or an acknowledgment that this is unresolved). The data presented in Fig. 3 and lines 198-213 share this trajectory/source region isotopic signature vs. aq processing issue. I'm very comfortable with the authors arguing for aqueous processing, but there remains some ambiguity at least in my mind that ought to be acknowledged or addressed.

- I am confused why the authors present ALW data in Figure 3b and supplementary Figure 5, but in comment 8 in their rebuttal, they say that they cannot calculate ALW due to not having RH% or temperature data. What am I missing for those two things to be true?

Responses to Reviewer #1:

General comments:

The authors provided thorough responses to the reviewers' comments. However, the authors did not address a key issue of whether oxalic acid can be used as an aqueous-phase SOA (aqSOA) tracer, which is the basis of this study. First, one should not mix up the aqueous-phase reaction and condensed phase reaction, especially for ambient measurements these two processes are very difficult to differentiate. In laboratory studies or in specific atmospheric environments (e.g., cloud processes), oxalic acid was indeed used to investigate aqSOA formation process of e.g., glyoxal, like in many previous publications the authors mentioned. However, in ambient air, it is much more complex and this aqueous-phase process is only one of formation pathways for oxalic acid. To my knowledge, there is no defined aqueous-phase tracer in real ambient air so far, although many colleagues suggested organosulfates could be the most promising tracer of aqueous-phase SOA. Second, photochemical oxidation could be still an important formation pathway for oxalic acid in Heshan city, located in south China where photochemical is often prevailing, resulting a mix of aqueous-phase/condensed phase processes and photochemical oxidation. Third, the primary emission of oxalic acid. As specified in the think response, the authors measured oxalic acid in source emission samples, i.e., 157 ng m⁻³ for biomass burning and 78 ng m⁻³ for coal combustion, which are about 30% of the average ambient concentration (446 ng m⁻³). It should be noted that oxalic acid concentration in source emission aerosols is dependent on the mass of fuel burned in the experiments (i.e., 1–2 kg of biomass per burn, 1 kg of coal per burn; Tang et al., 2020). Biomass burning is a very important emission source in south China, from both local emission and regional transport depending on the origin of air masses. Therefore, I would consider what the authors measured in this study is a combining result from primary emissions and secondary formation (aqueous-phase/condensed phase processes + photochemical oxidation). This cannot support what the authors claimed “the concentration of oxalic acid from primary sources would be very minor compared to the atmospheric oxalic acid pool”.

Taken together, it is not convincing and even danger to say oxalic acid the authors measured in Heshan city was from aqueous-phase process.

Response: We appreciate the efforts by the reviewer and have carefully considered the different aspects raised. First, we understand that the reviewer holds the view that there may be substantial primary sources of oxalic acid. It would have been useful if the reviewer would have provided updated references to support this view. Contrary to the perspective, there is an extensive literature pointing to a very limited contribution from primary sources; these many available convincing references are contributed by different groups around the world over the past two decades (and key ones are cited in the manuscript). Although early studies observed oxalic acid in motor exhaust (*Kawamura and Kaplan, 1987*) and biomass burning plume samples (*Narukawa et al., 1999*), oxalic acids in the ambient atmosphere is less likely to be of primary emission, based on findings from a large number of available studies: (i) For traffic exhaust, the evidence provided by *Huang and Yu (2007, GRL)* in their ms entitled “*Is vehicle exhaust a significant primary source of oxalic acid in ambient aerosols?*”, shows that (a) no enhancement in oxalate levels in the tunnel aerosol was seen in comparison with the oxalate levels in the ambient environment, (b) oxalate in the tunnel samples bore little resemblance in its size distribution to that of elemental carbon and had no correlation with elemental carbon, and (c) the ratio of oxalate/elemental carbon in the tunnel samples was more than 15 times lower than that in the ambient samples. (ii) For biomass burning, we highlight the broadly recognized work done by *Pinxteren et al., (2014, ACP)*, in which a large data set of size-resolved dicarboxylic acid concentrations from several inland sites in Germany was compiled and statistically analyzed to demonstrate that primary sources (biomass burning and traffic) were non-existent for the oxalic acid formation. (iii) In fact, the extent of oxidation reactions from biomass fuel to oxalic acid requires more energy and is very unlikely to be achieved by the biomass combustion process. Instead, the radical chemistry involved in ambient air are more intense and, hence, it is likely that subsequent photochemical reactions convert gaseous precursors to particulate oxalic acid by gas/aqueous phase reactions. Both vehicle exhaust and biomass burning are considered to emit methylglyoxal, glyoxal and other oxidants (e.g., *Zhang et al., 2016; Zarzana et al., 2018; Kluge et al., 2020; Liu et al, 2020*), which as precursors would rapidly photo-oxidize in the atmosphere to form oxalic acid and their homologues (*Grosjean et al., 2001; Carlton et al., 2007; Rinaldi et al., 2011; Wu et al., 2021; Lee et al., 2011*).

Second, in the manuscript, we did not deny the diversity of sources/origins of oxalic acid in

the atmosphere. In fact, the manuscript points out that photochemical oxidation and gas-phase processes would be important during coastal background period. We would like to underscore here some text in the manuscript relevant to this point, such as: *(i) Oxalic acid is an SOA end product and can be formed by two pathways. The first pathway is a stepwise photochemical breakdown of longer-chain aliphatic diacids, which leads to relatively enriched $\delta^{13}\text{C}$ values in the residual shorter-chain diacids. Recently, significant amounts of oxalic acid followed by succinic acid were also found to be produced via ozone oxidation of isoprene under dry condition³⁸. (Line 177-181) (ii) In this case, the photochemical degradation (aging) of oxalic acid would surpass its aqueous-phase formation, which is evident in the increasing $\delta^{13}\text{C}$ values of oxalic acid (Line 328-330) (iii) We also stress that apart from aqueous-phase processes, both biomass burning and gas-phase photochemical aging could also be sources for oxalic acid. (Line 452-454).* When we said in the manuscript that the aqueous processing was dominant, it was confined to wintertime when the air mass arrived at Heshan from upstream cities. We did not exclude the contributions from gas-phase reactions and photochemical oxidation. On the contrary, we said that in summertime, gas-phase photochemical processes dominated the formation of oxalic acid.

The case made in the manuscript for aqueous processing was based on careful interpretation of the dataset (such as Fig.3; Fig 4c). In our first-round revision, we replied by summarizing the key evidences to support that oxalic acid is indeed an aqueous-phase SOAs in the continental outflow samples, including: (i) a significant correlation of oxalic acid with aerosol liquid water; (ii) depleted $\delta^{13}\text{C}$ of oxalic acid and its aqueous-phase precursor glyoxylic acid and pyruvic acid, which points to $\delta^{13}\text{C}$ fractionation through aqueous-phase processing; (iii) the $\Delta^{14}\text{C}$ -derived fossil carbon contents of oxalic acid and glyoxylic acid were significantly higher than longer-chain diacids (C_3 and C_4 diacids), indicating an inconsistency in formation pathways between oxalic acid and its higher homologs, to which photochemical oxidation/breakdown in the gas-phase followed by partitioning onto the condensed phase may be more applied.

To further support this point, we did a new round of literature search. We identified a paper in ACP (*Cheng et al., 2017*), conducted at the same Heshan atmospheric supersite but in 2014–2015, that reported valuable information about the mixing state of oxalic acid using a single-particle aerosol mass spectrometer (SPAMS). This paper concluded the formation of oxalic acid was closely associated with the oxidation of organic precursors in the aqueous phase in Heshan, by giving

evidences that (1) oxalic acid was predominantly mixing with sulfate and nitrate during the whole sampling period, likely due to aqueous-phase reactions, (2) the peak areas of nitrate, sulfate and oxalic acid had similar temporal change in the carbonaceous type oxalic acid particles, (3) the organosulfate-containing oxalic acid particles correlated well with total oxalic acid particles during the haze episode. Obviously, this is consistent with our compound-specific $\delta^{13}\text{C}/\Delta^{14}\text{C}$ results.

Reference:

- Carlton A G, Turpin B J, Altieri K E, Seitzinger S, Reff A, Lim H-J, Ervens B 2007. Atmospheric oxalic acid and SOA production from glyoxal: Results of aqueous photooxidation experiments. *Atmos. Environ.* [J], 41: 7588-7602.
- Cheng C, Li M, Chan C K, Tong H, Chen C, Chen D, Wu D, Li L, Wu C, Cheng P, Gao W, Huang Z, Li X, Zhang Z, Fu Z, Bi Y, Zhou Z 2017. Mixing state of oxalic acid containing particles in the rural area of Pearl River Delta, China: implications for the formation mechanism of oxalic acid. *Atmospheric Chemistry and Physics* [J], 17: 9519-9533.
- Grosjean D, Grosjean E, Gertler A W 2001. On-Road Emissions of Carbonyls from Light-Duty and Heavy-Duty Vehicles. *Environmental Science & Technology* [J], 35: 45-53.
- Huang X-F, Yu J Z 2007. Is vehicle exhaust a significant primary source of oxalic acid in ambient aerosols? *Geophysical Research Letters* [J], 34.
- Kawamura K, Kaplan I R 1987. Motor exhaust emissions as a primary source for dicarboxylic acids in Los Angeles ambient air. *Environ. Sci. Technol.* [J], 21: 105-110.
- Kluge F, Hüneke T, Knecht M, Lichtenstern M, Rotermund M, Schlager H, Schreiner B, Pfeilsticker K 2020. Profiling of formaldehyde, glyoxal, methylglyoxal, and CO over the Amazon: normalized excess mixing ratios and related emission factors in biomass burning plumes. *Atmospheric Chemistry and Physics* [J], 20: 12363-12389.
- Lee A K Y, Zhao R, Gao S S, Abbatt J P D 2011. Aqueous-Phase OH Oxidation of Glyoxal: Application of a Novel Analytical Approach Employing Aerosol Mass Spectrometry and Complementary Off-Line Techniques. *The Journal of Physical Chemistry A* [J], 115: 10517-10526.
- Liu J, Li X, Li D, Xu R, Gao Y, Chen S, Liu Y, Zhao G, Wang H, Wang H, Lou S, Chen M, Hu J, Lu K, Wu Z, Hu M, Zeng L, Zhang Y 2020. Observations of glyoxal and methylglyoxal in a suburban area of the Yangtze River Delta, China. *Atmospheric Environment* [J], 238.
- Narukawa M, Kawamura K, Takeuchi N, Nakajima T 1999. Distribution of dicarboxylic acids and carbon isotopic compositions in aerosols from 1997 Indonesian forest fires. *Geophysical Research Letters* [J], 26: 3101-3104.
- Rinaldi M, Decesari S, Carbone C, Finessi E, Fuzzi S, Ceburnis D, O'Dowd C D, Sciare J, Burrows J P, Vrekoussis M, Ervens B, Tsigaridis K, Facchini M C 2011. Evidence of a natural marine source of oxalic acid and a possible link to glyoxal. 116.
- van Pinxteren D, Neusüß C, Herrmann H 2014. On the abundance and source contributions of dicarboxylic acids in size-resolved aerosol particles at continental sites in central Europe. *Atmospheric Chemistry and Physics* [J], 14: 3913-3928.
- Wu Z, Zhang Y, Pei C, Huang Z, Wang Y, Chen Y, Yan J, Huang X, Xiao S, Luo S, Zeng J, Wang J, Fang H, Zhang R, Li S, Fu X, Song W, Wang X 2021. Real-world emissions of carbonyls from vehicles in an urban tunnel in south China. *Atmospheric Environment* [J], 258: 118491.

Zarzana K J, Selimovic V, Koss A R, Sekimoto K, Coggon M M, Yuan B, Dubé W P, Yokelson R J, Warneke C, de Gouw J A, Roberts J M, Brown S S 2018. Primary emissions of glyoxal and methylglyoxal from laboratory measurements of open biomass burning. *Atmospheric Chemistry and Physics* [J], 18: 15451-15470.

Zhang Y, Wang X, Wen S, Herrmann H, Yang W, Huang X, Zhang Z, Huang Z, He Q, George C 2016. On-road vehicle emissions of glyoxal and methylglyoxal from tunnel tests in urban Guangzhou, China. *Atmospheric Environment* [J], 127: 55-60.

Specific comments:

1. The authors tried to demonstrate that “the $\delta^{13}\text{C}$ composition of C3 plant-derived isoprene is indeed close to the $\delta^{13}\text{C}$ of AVOCs,” so that “kinetic isotopic fractionation the most probable processes controlling the $\delta^{13}\text{C}$ of oxalic acid” (Line 172-178). From the response, it is still not clear why “the $\delta^{13}\text{C}$ of isoprene (-29‰ to -26‰) suggested by Iannone et al. [2007] could represent the biogenic source of isoprene in our study”. The ^{13}C value of -26‰ for isoprene is at the higher end of the range summarized in Table R1-1.

Response: We adapt the $\delta^{13}\text{C}$ of isoprene (-29‰ to -26‰) suggested by Iannone et al. [2007] could represent the biogenic source of isoprene in our study, because it is the only literature which directly measured the $\delta^{13}\text{C}$ of isoprene in real forest environment where dominated by C3 plants. We do not agree that the ^{13}C value of -29‰ to -26‰ is at the higher end of the range summarized in Table R1-1. The reasons are as follows.

First, the selection of isoprene- $\delta^{13}\text{C}$ value. We exclude the values from modeling (-30.4‰ to -29‰; Gromov et al. 2017) and estimation (-29‰; Li et al. 2019). We then exclude isoprene- $\delta^{13}\text{C}$ value from specific C3 plants, such as Velvet Bean, per your suggestion in the previous review. Therefore, the values in Iannone et al. [2007] is the most appropriate one, and they were from direct measurements in real forest environment where dominated by C3 plants.

Second, we would like to note that the $\delta^{13}\text{C}$ value of -29‰ to -26‰ (Iannone et al. 2007) were actually in the similar range of the directly measured value of isoprene (-27.7‰ \pm 2‰; Rudolph et al. 2003), as well as those of terpenoids (-27.1‰ \pm 2.5‰; Diefendorf et al. 2012), when plus/minus (\pm) are noted for the latter two values.

Table R1-1. $\delta^{13}\text{C}$ composition of isoprene and terpenoids emitted from C3 plants

Plant Species	Compounds	$\delta^{13}\text{C}$	References
---------------	-----------	-----------------------	------------

Myrtle, buckthorn, velvet bean	Isoprene	$-29.2‰ \pm 0.6‰$	Affek and Yakir (2003)
velvet bean	Isoprene	$-27.7‰ \pm 2‰$	Rudolph et al. (2003)
Mixed deciduous forest in Germany	Isoprene	$-29‰$ to $-26‰$	Iannone et al. (2007)
Surface emission (from model)	Isoprene	$-30.4‰$ to $-29‰$	Gromov et al. (2017)
Forest areas in Southwest China (from estimation)	Isoprene	$-29‰$	Li et al. (2019)
44 C3 plants	Terpenoids	$-27.1‰ \pm 2.5‰$	Diefendorf et al. (2012)

Similarly, the $\delta^{13}\text{C}$ of anthropogenic VOCs (AVOCs; e.g., aromatic hydrocarbons: $-27.7‰ \pm 1.7‰$), is taken from the reference 37, which is specific for transportation. I believe at this receptor site, the coal combustion (more enriched in ^{13}C than vehicular emissions) and biomass burning (in Line 206, it says clearly that anthropogenic sources include biomass burning) are also non-negligible sources of anthropogenic VOCs, especially during the continental outflow. The authors should fully consider the $\delta^{13}\text{C}$ of anthropogenic VOCs (from transportation, coal combustion, also biomass burning) and $\delta^{13}\text{C}$ of C3-plant derived biogenic VOCs, and discuss whether the two are similar, because this is the basis of conclusions made from the $\delta^{13}\text{C}$ of oxalic acid, including conclusions relating to possible formation pathway of aqSOA.

Response: Thanks for your comments. We agree with the reviewer that the AVOCs from coal and biomass burning should also be considered. We compiled average $\delta^{13}\text{C}$ -AVOCs from different source types, including industrial stack, coal, biomass-burning, transportation and gasoline. The available data is listed in **Supplementary Table 5**. The $\delta^{13}\text{C}$ values of AVOCs from biomass burning (C3 plants), gasoline and transportation (including tunnel, gas station and underground garage) are similar to the average values of C3-plants-derived BVOCs. The average $\delta^{13}\text{C}$ of AVOCs derived from industrial stack/coal combustion were around $-25‰$. Although coal-derived AVOCs are enriched in $\delta^{13}\text{C}$ compared to vehicular emissions and biomass burning, the difference is minor ($\sim 2‰$). There are also previous literatures supporting our viewpoint that $\delta^{13}\text{C}$ values of isoprene emitted from C3 plants are similar to VOCs emitted from major anthropogenic sources (Iannone et al. 2007; Vitzthum et al. 2011).

Second, in continental outflow (refer to **Figure 5**), oxalic acid with higher fossil-carbon

contributions was more depleted in $\delta^{13}\text{C}$. Therefore, coal contribution would not be responsible for the depleted $\delta^{13}\text{C}$ of oxalic acid, since the coal emissions are more enriched in $\delta^{13}\text{C}$ than biogenic/biomass burning emissions. The remaining possibility is the source mixing of C4-plant and marine-biogenic emissions, because carbon pool from fossil fuel is significantly depleted in $\delta^{13}\text{C}$ than C4 plants (-17‰ to -9‰; *Smith and Epstein, 1971*) and marine phytoplankton (-22‰ to -18‰; *Miyazaki et al., 2011*). However, in our dataset, the combination of isotopes data and inorganic ratios diagnosed an absence of significant contributions from marine-biogenic sources and C4-plant sources (refer to **Line 162-170**).

Last, we have provided other observation-based evidences demonstrating that source mixing is not expected to have a major influence on the $\delta^{13}\text{C}$ composition of oxalic acid at the Heshan site. As shown in **Figure 3a**, higher homologs aliphatic diacids from C₃ to C₉, in contrast to oxalic acid, show no significant statistical difference between the two air mass source regions. If source mixing was a significant factor influencing the $\delta^{13}\text{C}$ compositions, the $\delta^{13}\text{C}$ of higher homologs aliphatic diacids from different source regions would have also been different. Taken together, this supports that kinetic isotopic fractionation, instead of sources mixing, was responsible for the discrepancies of $\delta^{13}\text{C}$ values of oxalic acid among the two source regions.

Per your kind suggestion, we have fully considered the $\delta^{13}\text{C}$ of VOCs from various anthropogenic sources. Please refer to the revised manuscript, in which changes are highlighted in yellow:

The revised text:

Given contributions from marine and C4 plants sources are mostly excluded at the Heshan site, the potential precursors of oxalic acid would then be anthropogenic VOCs (AVOCs; e.g., aromatic hydrocarbons) and C3 plant-derived biogenic VOCs (BVOCs; e.g., isoprene), of which the $\delta^{13}\text{C}$ values are similar (Supplementary Table 5). (Line 170-174)

Supplementary Table 5. Stable carbon isotope ratios ($\delta^{13}\text{C}$, ‰) of major sources of biogenic VOCS (BVOCs) and anthropogenic VOCs (AVOCs).

Source	Compounds	$\delta^{13}\text{C}$ (‰)
BVOCs	Myrtle, buckthorn, velvet bean ^a	-29.2‰ ± 0.6‰
	velvet bean ^b	-27.7‰ ± 2‰

	Mixed deciduous forest ^c	Isoprene	-29‰ to -26‰
	44 C3 plants ^d	Terpenoids	-27.1‰ ± 2.5‰
	Industrial stack ^e	Benzene and Toluene	-25.4‰ to -23.5‰
	Industrial stack ^f	n -alkanes, aromatics, ketones and n -alcohol	-25.5‰ ± 2.5‰
	Coal ^g	2-6 rings aromatics	-29.0‰ to -24.2‰
	Biomass burning (C3 plants) ^h	Non-methane Hydrocarbons	-26.5‰ to -25.7‰
AVOCs	Biomass burning (C3 plants) ⁱ	Benzene and Toluene	-27.6‰ to -27.1‰
	Biomass burning (C3 plants) ^j	C ₆ -C ₁₀ VOCs	-27.6‰ ± 1.6‰
	Transportation ^k	Non-methane Hydrocarbons	-27.7‰ ± 1.7‰
	Fossil fuel combustion ^l	Benzene and Toluene	-27.5‰ to -26.9‰
	Gas station ^l	C ₅ -C ₁₁ VOCs	-26.8‰ ± 1.9‰
	Gasoline ^m	Non-methane Hydrocarbons	-26.4‰ ± 1.1‰

^a Affek and Yakir (2003). ^b Rudolph et al. (2003). ^c Iannone et al. (2007). ^d Diefendorf et al. (2012). ^e Turner et al. (2006). ^f Vitzthum et al. (2011). ^g McRae et al. (1996). ^h Czapiewski et al. (2002). ⁱ Giebel et al. (2010). ^j Vitzthum et al. (2012). ^l Kawashima et al. (2014). ^m Averages and standard deviations was estimated by a figure reported by Smallwood et al. (2002).

References:

- Affek H P, Yakir D 2003. Natural abundance carbon isotope composition of isoprene reflects incomplete coupling between isoprene synthesis and photosynthetic carbon flow. *Plant Physiol.* [J], 131: 1727-1736.
- Czapiewski K v, Czuba E, Huang L, Ernst D, Norman A, Koppmann R, Rudolph J J J o a c 2002. Isotopic composition of non-methane hydrocarbons in emissions from biomass burning. 43: 45-60.
- Diefendorf A F, Freeman K H, Wing S L 2012. Distribution and carbon isotope patterns of diterpenoids and triterpenoids in modern temperate C3 trees and their geochemical significance. *Geochimica et Cosmochimica Acta* [J], 85: 342-356.
- Giebel B M, Swart P K, Riemer D D 2010. $\delta^{13}\text{C}$ Stable Isotope Analysis of Atmospheric Oxygenated Volatile Organic Compounds by Gas Chromatography-Isotope Ratio Mass Spectrometry. *Analytical Chemistry* [J], 82: 6797-6806.
- Iannone R, Koppmann R, Rudolph J 2007. A technique for atmospheric measurements of stable carbon isotope ratios of isoprene, methacrolein, and methyl vinyl ketone. *Journal of Atmospheric Chemistry* [J], 58: 181-202.
- Kawashima H, Murakami M 2014. Measurement of the stable carbon isotope ratio of atmospheric volatile organic

- compounds using chromatography, combustion, and isotope ratio mass spectrometry coupled with thermal desorption. *Atmospheric Environment* [J], 89: 140-147.
- McRae C, Love G D, Murray I P, Snape C E, Fallick A E J A C 1996. Potential of gas chromatography isotope ratio mass spectrometry to source polycyclic aromatic hydrocarbon emissions. 33: 331-333.
- Miyazaki Y, Kawamura K, Jung J, Furutani H, Uematsu M 2011. Latitudinal distributions of organic nitrogen and organic carbon in marine aerosols over the western North Pacific. *Atmos. Chem. Phys.* [J], 11: 3037-3049.
- Rudolph J, Anderson R S, Czapiewski K V, Czuba E, Ernst D, Gillespie T, Huang L, Rigby C, Thompson A E 2003. The stable carbon isotope ratio of biogenic emissions of isoprene and the potential use of stable isotope ratio measurements to study photochemical processing of isoprene in the atmosphere. *J. Atmos. Chem.* [J], 44: 39-55.
- Rudolph J, Czuba E, Norman A L, Huang L, Ernst D 2002. Stable carbon isotope composition of nonmethane hydrocarbons in emissions from transportation related sources and atmospheric observations in an urban atmosphere. *Atmospheric Environment* [J], 36: 1173-1181.
- Smallwood B J, Paul Philp R, Allen J D 2002. Stable carbon isotopic composition of gasolines determined by isotope ratio monitoring gas chromatography mass spectrometry. *Organic Geochemistry* [J], 33: 149-159.
- Smith B N, Epstein S 1971. 2 categories of C-13/C-12 ratios for higher plants. *Plant Physiol.* [J], 47: 380-&.
- Turner N, Jones M, Grice K, Dawson D, Ioppolo-Armanios M, Fisher S J 2006. $\delta^{13}\text{C}$ of volatile organic compounds (VOCS) in airborne samples by thermal desorption-gas chromatography-isotope ratio-mass spectrometry (TD-GC-IR-MS). *Atmospheric Environment* [J], 40: 3381-3388.
- Vitzthum von Eckstaedt C, Grice K, Ioppolo-Armanios M, Jones M 2011. $\delta^{13}\text{C}$ and δD of volatile organic compounds in an alumina industry stack emission. *Atmospheric Environment* [J], 45: 5477-5483.
- Vitzthum von Eckstaedt C D, Grice K, Ioppolo-Armanios M, Kelly D, Gibberd M 2012. Compound specific carbon and hydrogen stable isotope analyses of volatile organic compounds in various emissions of combustion processes. *Chemosphere* [J], 89: 1407-1413.

In Figure 5, during the continental outflow with important influence of aqSOA formation, the $\delta^{13}\text{C}$ of oxalic acid ranges from -31‰ to -21‰, with some datapoints fall in the range of $\delta^{13}\text{C}$ of anthropogenic VOCs and biogenic VOCs. Following the discussion in Line 204-206, “As aqSOA formed from VOCs or SVOCs typically leads to lower $\delta^{13}\text{C}$ values relative to the precursors since lighter-isotope containing precursors are preferentially oxidized to form reaction products”, more depleted (lower) $\delta^{13}\text{C}$ values than $\delta^{13}\text{C}$ of anthropogenic VOCs and biogenic VOCs (around -29‰ to -26‰ in Figure 5) are expected, however, Figure 5 shows the opposite, with the most of continental outflow samples falls in the range -29‰ to -21‰.

Response: Thanks for your comments. The ^{13}C -enriched oxalic acid than its precursors (AVOCs and BVOCs) during the continental outflow does not preclude aqSOA formation, which lead to a ^{13}C -depletion. In fact, the observed $\delta^{13}\text{C}$ value is a balance of isotopic fractionations both by aqSOA formation and by photochemical aging. In another word, even though ^{13}C -depletion is expected for aqSOA formation, subsequent ^{13}C enrichment may also occur through photochemical

aging of recently formed particles. Oxalic acid can be removed from the atmosphere both by oxidation with OH and NO₃ radical attack in aqueous phase, and by forming a complex with iron in aqueous phase and then undergo decarboxylation reaction yielding CO₂ in the presence of sunlight (Zuo and Hoigne, 1994). A rapidly ¹³C-enrichment in the residual oxalic acid has been demonstrated in both of the aging mechanisms (Pavuluri and Kawamura, 2011, 2016). Therefore, subsequent photochemical aging ($\delta^{13}\text{C}$ -enrichment) would overwhelm the $\delta^{13}\text{C}$ -depletion trend. However, thanks to the unique geographic setting of the Heshan receptor site, we observed $\delta^{13}\text{C}$ -depletion signal for individual diacids during continental outflow compared to coastal background, which is the evidence standing for the proposed atmospheric aqueous-phase formation of oxalic acid.

References:

- Pavuluri C M, Kawamura K 2012. Evidence for ¹³ - carbon enrichment in oxalic acid via iron catalyzed photolysis in aqueous phase. *Geophysical Research Letters* [J], 39: 3802.
- Pavuluri C M, Kawamura K 2016. Enrichment of ¹³C in diacids and related compounds during photochemical processing of aqueous aerosols: New proxy for organic aerosols aging. *Scientific Reports* [J], 6.
- Zuo Y, Hoigné J 1994. Photochemical decomposition of oxalic, glyoxalic and pyruvic acid catalysed by iron in atmospheric waters. *Atmospheric Environment* [J], 28: 1231-1239.

2. The correlation between ffossil-C2 and C2 concentrations for continental outflow is weak. And there is very weak correlation for coastal background (Supplementary Figure 6). The concentrations of C2 oxalic acid and ffossil-C2 for continental outflow are larger than those for coastal background; however, it is not reasonable to pool the data of continental outflow and costal background together, and conclude that “The concentrations of C2 oxalic acid increased with the increase of ffossil-C2”. In the Supplementary Figure 6, the increased concentrations and ffossil-C2 are found in the continental outflow, but the datapoints for continental outflow are scattered with no obvious correlations. For coastal background, the oxalic acid concentrations were low, and the non-fossil sources contributed more strongly to oxalic acid (i.e., ffossil-C2 <50%). Therefore, the argument “The concentrations of C2 oxalic acid increased with the increment of ffossil-C2($r^2= 0.52$, $P < 0.001$; Supplementary Fig. 5), suggesting that the precursors from fossil-fuel emission was the dominant factor in controlling the ambient abundance of oxalic acid” (Line 247-250) is an overinterpret of the data.

Response: Thanks for your comments. In the previous round of revision, per your good comments, we have revised this sentence to better express this evidence. The sentence you pointed out was not our revised sentence. Actually, the Line 247-250 is “*The concentrations of C₂ oxalic acid increased with the increase of f_{fossil}-C₂ (r² = 0.52, P < 0.001; Supplementary Fig. 6), suggesting that the precursors from fossil-fuel emission was the major factor responsible for the enhanced ambient abundance of oxalic acid” . We guess there was a confusion between our revised manuscript and the originally submitted one.*

We again agree with your point. The scattered datapoints appeared in the plot was difficult to draw convincing conclusion. Therefore, we have deleted the Supplementary Fig. 6 as well as the corresponding discussion (please refer to the paragraph from Line 235-251).

3. The argument “oxalic acid was more enriched in 13C and 14C than WSOC in summer” (Line 380-382) is not true for 14C in Beijing (BJ) and Chengdu (CD), challenging again the validation to extend to wider spatial coverage of China. The spatial extent of the findings at Heshan site is further challenged by Line 378-380 “The average $\delta^{13}\text{C}$ and 14C-derived f_{bio/bb} values of WSOC show no difference in winter and summer ($\delta^{13}\text{C}$: $-24.2 \pm 0.4\%$ vs. $-24.9 \pm 0.7\%$; f_{bio/bb}: $63 \pm 11\%$ vs. $60 \pm 9\%$)”, is not true for each city, e.g., in Chengdu, $\delta^{13}\text{C}$: -23.8% vs. -25.9% , f_{bio/bb}: 78% vs. 62% ; in Beijing, f_{bio/bb}: 60% vs. 45% ; Shanghai: f_{bio/bb}: 49% vs. 62% , and the seasonal variation of f_{bio/bb} of WSOC is different in different cities, in Wuhan and Shanghai, f_{bio/bb} of WSOC was higher in summer, and in Beijing and Chengdu, it shows the opposite.

Response: Thanks for your comment. Sorry for the confusion we made. First, as reminded by you, we found it was not necessary to use the average values in the discussion. Second, we would like to clarify that, what we really wanted to show is that the differences of carbon isotopes between oxalic acid and bulk WSOC in winter and in summer, respectively; whilst we did not intend to describe the seasonal variations of carbon isotopes of oxalic acid and WSOC, which is out of the scope of this manuscript. Therefore, we have rewritten this section, and have redrawn Fig.6. In the revised text and figure, we have emphasized that it is to compare the carbon isotopes of oxalic acid and WSOC, and have made more cautious interpretation of the dataset. As per your comments, the changes made are as follows:

– “Oxalic acid was more enriched in ¹³C (by 2.2‰) and ¹⁴C (f_{bio/bb} increased by 5.0%) than

WSOC in summer”: we think it is not necessary to include average values here, and Beijing and Chengdu should be addressed. The description has been thus rewritten to be “*On the contrary, in summer, oxalic acid was more enriched in ¹³C and ¹⁴C than the WSOC pool in Guangzhou, Wuhan and Shanghai. As shown in Fig. 6, a general trend of blue-above-red is displayed, despite the very close values in Beijing and Chengdu which are within the analytical uncertainties.*”

- “The average $\delta^{13}\text{C}$ and ^{14}C -derived $f_{\text{bio/bb}}$ values of WSOC show no difference in winter and summer ($\delta^{13}\text{C}$: $-24.2 \pm 0.4\%$ vs. $-24.9 \pm 0.7\%$; $f_{\text{bio/bb}}$: $63 \pm 11\%$ vs. $60 \pm 9\%$)”: we found that this determination based on average values was misleading and not within our discussion scope. We have thus deleted this sentence.
- Please refer to the revised manuscript, in which changes are highlighted in yellow.

The revised text:

To test the spatial extent of the findings, we retrieved aerosol samples from five emission hotspot megacities of China (Fig. 6a and Supplementary Table 9). The difference of $\delta^{13}\text{C}$ and ^{14}C -derived $f_{\text{bio/bb}}$ values between oxalic acid and bulk WSOC were compared in winter and in summer, respectively (Fig. 6b). Although oxalic acid accounts for a large fraction of WSOC (~5.2%), probably the single most abundant compound, we observed significant but opposite difference in the $\delta^{13}\text{C}$ and $\Delta^{14}\text{C}$ compositions between oxalic acid and WSOC during the two seasons. In winter, oxalic acid was more depleted in ^{13}C and ^{14}C than the WSOC pool in each of the cities, red-above-blue in Fig. 6b (refer to Supplementary Table 9). This agrees with the dominant role of aqueous-phase formation processes triggered by the increased ALW in winter³². As mentioned above, overwhelming fossil contributions to aqSOA compounds were observed, that were consistent with or even higher than the fossil content in water-insoluble OA reported in Chinese urban areas during winter haze^{52, 56}. Substantial fossil-derived precursors are likely oxidized to WSOC aerosol through secondary aqueous processing in winter, resulting in products (such as oxalic acid) with more negative $\delta^{13}\text{C}$ values and more fossil contributions than the bulk WSOC pool.

On the contrary, in summer, oxalic acid was more enriched in ^{13}C and ^{14}C than the WSOC pool in Guangzhou, Wuhan and Shanghai. As shown in Fig. 6, a general trend of blue-above-red is displayed, despite the very close values in Beijing and Chengdu which are within the analytical uncertainties. A probable interpretation is that biomass/biogenic component from the WSOC pool

is preferentially subjected to oxidative aging to small molecules^{12, 57}, which results in oxalic acid enriched both in $\delta^{13}\text{C}$ and $\Delta^{14}\text{C}$. The differing isotopic signatures of different emission sources would also contribute to the enrichment of carbon isotopes of oxalic acid. Note here that biogenic contributions to oxalic acid can be expected to increase during summer due to it being the growing season. (Line 367-392)

Figure 6. The $\delta^{13}\text{C}$ composition and ^{14}C -based source apportionment of WSOC and oxalic acid in $\text{PM}_{2.5}$ collected from five emission hotspot megacities of China. (a) Locations of the five megacities (Beijing, Guangzhou, Wuhan, Chengdu and Shanghai) and aerosol optical depth (AOD) at 550 nm during the sampling period. Abbreviations on the map: North China Plain (NCP), Yangtze River Delta (YRD), Pearl River Delta (PRD), the middle reach of the Yangtze River (MYR), and Sichuan Basin (SC). (b) The difference of $\delta^{13}\text{C}$ and ^{14}C -based fraction of non-fossil sources between oxalic acid and WSOC in winter (January 2018) and in summer (July 2018), respectively.

4. Line 350-353: “ ^{14}C analyses of aerosol OC and its sub-fractions have been interpreted such that the majority of fossil-derived OA was water-insoluble 53,54”. However, in fact, ^{14}C analysis interprets the majority of primary fossil-derived OA was water-insoluble (see reference 54). Therefore, the conclusion “Hence, these earlier field and laboratory studies show that water-soluble OA is only to very limited extent stemmed from fossil sources” is not always true. And it is contrary to what the authors state in the Introduction: “Radiocarbon-based estimations attributed ~50% of water-soluble OA to fossil sources in the East Asia outflow” (Line 61-62). Previous studies of direct ^{14}C measurement on water-soluble OC found that large fossil contributions to water-soluble OC, e.g., ~50% in winter in Beijing (reference 54), ~30% in autumn in Xi’an (Pavuluri et al., 2013),

~50% in summer in Guangzhou (Liu et al., 2018), ~50% of Chinese outflow in spring 2011 in Jeju Island (Fang et al., 2017).

Pavuluri CM, Kawamura K, Uchida M, Kondo M, Fu P. Enhanced modern carbon and biogenic organic tracers in Northeast Asian aerosols during spring/summer. *Journal of Geophysical Research: Atmospheres* 2013, 118(5): 2362-2371.

Liu J, Ding P, Zong Z, Li J, Tian C, Chen W, et al. Evidence of Rural and Suburban Sources of Urban Haze Formation in China: A Case Study From the Pearl River Delta Region. *Journal of Geophysical Research: Atmospheres* 2018, 123(9): 4712-4726.

Fang W, Andersson A, Zheng M, Lee M, Holmstrand H, Kim S-W, et al. Divergent Evolution of Carbonaceous Aerosols during Dispersal of East Asian Haze. *Scientific Reports* 2017, 7(1): 10422

Response: First, our citation is identical to the original text in the reference 54 (Zhang et al., 2018; 10.5194/acp-18-4005-2018), in which the authors did not differentiate the primary OA and secondary OA. We copy the original text here: *“On average, the majority (60–70%) of the fossil OC was water insoluble at these three sites, indicating that fossil-derived OA mostly consisted of hydrophobic components and thus is less water soluble than OA from non-fossil sources. This result is consistent with findings reported elsewhere such as at an urban or rural site in Switzerland (Zhang et al., 2013), a remote site on Hainan Island, southern China (Zhang et al., 2014a) and at two rural sites on the east coast of the United States (Wozniak et al., 2012)”*. Similarly, in reference 53, Wozniak et al., 2012 (JGR; 10.1029/2011jd017153) suggested that *“Aerosol WSOC consistently showed low fossil content (<8%) relative to the TOC (5-50%) indicating that the majority of fossil OC in aerosol particles is insoluble.”*

Second, we cite the above-mentioned point for the purpose of bringing forward its contradiction to the fact that *“Radiocarbon-based estimations attributed ~50% of water-soluble OA to fossil sources in the East Asia outflow”* (in the introduction section Line 59-61), which is indeed what our paper looks at. Therefore, our manuscript was written to give compound-specific radiocarbon evidence that fossil-derived secondary OC can be water-soluble, which could be different from its primary counterpart. We tend to believe that you may also hold this point, noting that you inserted “primary” in the comment text.

Responses to Reviewer #2:

With their revision of "Large contribution of fossil-derived components to aqueous secondary organic aerosols in China," Xu et al. have answered the majority of my concerns with the initial submission. The data appear to show evidence for the aqueous phase production of secondary organic aerosols (SOA) from fossil-derived carbon which, to my knowledge, is a novel finding within the aerosol literature. I suggest that the paper be accepted for publication. Below, I note a few outstanding comments for the authors to consider prior to publication.

Response: Thank for your nice summary of our paper and positive assessment of the importance of this work. We have carefully revised the manuscript following your comments. Our responses to your comments are given below, and the revised parts are shown after the responses (in blue font). Please refer to the revised manuscript, in which changes are highlighted in yellow.

- Ultrasonication - the experimental data presented to demonstrate that ultrasonication does not effect the concentrations and isotopic signatures of WSOC and oxalic acid was convincing. The prior argument that it has been used extensively in the literature as evidence for the validity of the extraction technique was not at all convincing, however, and I caution the authors against using this type of argument. Rather, the experimental data are important for validating the lack of effects in these previous studies.

Response: We are grateful to your kind advice. We appreciate your suggestion in the last round of reviewing, which reminded us to validate the extraction technique experimentally in the lab. This will be a memorable experience/story in our group, especially for the students!

- lines 375-393 - The authors argue that seasonal differences in different compound classes (longer chain diacids) are evidence for aqueous processing rather than for different trajectories, but I'm missing something in their argument. It seems to me that there remains the possibility that the differences in the isotopic signatures could result from differing isotopic signatures of the precursor compounds coming from the two different source regions. Throughout the paper, the authors argue convincingly that the WSOC isotopic signature is not necessarily diagnostic for the aqSOA isotopic signature because the WSOC isotopic signature is a mixture of many different components with

different isotopic signatures. With this argument in lines 375-393 though, they seem to suggest that the components do all start with the same signature before becoming altered via aqSOA, and I'm having a hard time accepting that. I think it's fine for the authors to suggest that the differences are due to aqSOA, but I would like to see a better argument against the oxalic acid and Pyr precursors coming from the different trajectories/source regions simply having different isotopic signatures (or an acknowledgment that this is unresolved). The data presented in Fig. 3 and lines 198-213 share this trajectory/source region isotopic signature vs. aq processing issue. I'm very comfortable with the authors arguing for aqueous processing, but there remains some ambiguity at least in my mind that ought to be acknowledged or addressed.

Response: Thanks for your comments. We agree with the reviewer that the differences in the isotopic signatures of emission sources may also contribute to the isotopic variations of oxalic acid. In this section, we proposed that the differences in the isotopic signatures between oxalic acid and WSOC were due to atmospheric processing at the Heshan site. Per your comments, we aware that the discussions on other possibility (isotopic difference of different emission sources in different seasons) in the five megacities are lacked. For example, as you mentioned, natural biomass inputs would increase during summer due to it being the growing season. In this scenario, oxalic acid with more contemporary carbon compared to WSOC can also be expected. However, existing data does not sufficient to support to quantitatively look at the contribution from the differences in emission sources. The extent to which the atmospheric processing is representative for the differences in isotopic signatures remains to be investigated in more details in different geographic regions.

Therefore, we have revised the text (Line 367-392) and figure (Line 796-805) corresponding to the carbon isotopes of WSOC and oxalic acid in five Chinese megacities. First, we have made more cautious interpretations of the summer-dataset. Second, we acknowledged other factors which may contribute to the carbon-isotope differences in the last of this section (**Effects of aqueous-phase processing on organic aerosols**). We have also further addressed the limitations in the last section (**Atmospheric implication**) of the manuscript.

The revised text-1 and Figure 6:

On the contrary, in summer, oxalic acid was more enriched in ^{13}C and ^{14}C than the WSOC pool in Guangzhou, Wuhan and Shanghai. As shown in Fig. 6, a general trend of blue-above-red is displayed, despite the very close values in Beijing and Chengdu which are within the analytical

uncertainties. A probably interpretation is that biomass/biogenic component from the WSOC pool is preferentially subjected to oxidative aging to small molecule^{12,57}, which results in oxalic acid enriched both in $\delta^{13}\text{C}$ and $\Delta^{14}\text{C}$. The differing isotopic signatures of different emission sources would also result in the enriched carbon isotope of oxalic acid. Note here that biogenic contributions to oxalic acid can be expected to increase during summer due to it being the growing season. (Line 383-392)

Figure 6. The $\delta^{13}\text{C}$ composition and ^{14}C -based source apportionment of WSOC and oxalic acid in $\text{PM}_{2.5}$ collected from five emission hotspot megacities of China. (a) Locations of the five megacities (Beijing, Guangzhou, Wuhan, Chengdu and Shanghai) and aerosol optical depth (AOD) at 550 nm during the sampling period. Abbreviations on the map: North China Plain (NCP), Yangtze River Delta (YRD), Pearl River Delta (PRD), the middle reach of the Yangtze River (MYR), and Sichuan Basin (SC). (b) The difference of $\delta^{13}\text{C}$ and ^{14}C -based fraction of non-fossil sources between oxalic acid and WSOC in winter (January 2018) and in summer (July 2018), respectively.

The revised text-2:

In this study, the $\delta^{13}\text{C}$ differences of oxalic acid was dominantly attributed to kinetic isotope effects in atmospheric processing. It should be noted that the differences in the isotopic signatures of emission sources may also contribute to the isotopic variations of oxalic acid. However, these factors are not quantitatively constrained. The extent to which the atmospheric processing is representative for the differences in isotopic signatures remains to be investigated in more details. We also stress that apart from aqueous-phase processes, both biomass burning and gas-phase photochemical aging could also be sources for oxalic acid. Therefore, cautious interpretations of

oxalic acid isotopic signals are needed where there are intensive biomass burning, extensive atmospheric aging, and seasonality of C3/C4 vegetation changes. Future dual-carbon isotope studies on other aqSOA constituents (e.g., glyoxal) are strongly warranted to better constrain the chemical mechanism responsible for fossil-derived aqSOA formation. (Line 447-459)

- I am confused why the authors present ALW data in Figure 3b and supplementary Figure 5, but in comment 8 in their rebuttal, they say that they cannot calculate ALW due to not having RH% or temperature data. What am I missing for those two things to be true?

Response: Thanks for your comments. Our manuscript contained two sampling campaigns. The one is year-round aerosol sampling in a regional receptor site (Heshan Atmospheric Environmental Monitoring Superstation) (refer to Line 470-480). The atmospheric environmental monitoring superstation provided hourly meteorological data, including RH% and temperature data. Therefore, the ALW in aerosols collected at the Heshan site can be calculated, as presented in Figure 3b and supplementary Figure 5.

The other sampling campaign is winter–summer aerosol snapshots in the five major emission megacities of China (refer to Line 481-486). The meteorological data in the specific sampling sites is not available. In the responses to comment 8, we approximately used the average RH% and temperature in each city from the China National Environmental Monitoring center. However, as our responses in comment 8: *“On one hand, the average RH% and temperature of the city may not well represent the real meteorological parameters at the sampling sites. On the other hand, as the PM_{2.5} samples in one week were pooled for the ¹⁴C analysis, we can only roughly calculate the weekly averaged ALW content, which is weak to support the aqueous-phase processes.”* Therefore, we did not include the ALW data of the five Chinese cities in the manuscript.

REVIEWERS' COMMENTS

Reviewer #1 (Remarks to the Author):

The authors have addressed some of my comments, some remained. Especially, the use of oxalic acid as an aqueous-phase SOA tracer, without clearly differentiating from other formation pathways, to explain the results from field measurements is still very challenging...The authors should at least specifically discuss this issue with caution....

Reviewer #2 (Remarks to the Author):

The authors have addressed my comments, and I have no further comments.

Author responses to reviews of *Nature Communications* manuscript “Large contribution of fossil-derived components to aqueous secondary organic aerosols in China” (Manuscript ID: NCOMMS-21-48555C)”

Responses to Reviewer #1:

General comments:

The authors have addressed some of my comments, some remained. Especially, the use of oxalic acid as an aqueous-phase SOA tracer, without clearly differentiating from other formation pathways, to explain the results from field measurements is still very challenging...The authors should at least specifically discuss this issue with caution....

Response: Thanks for your comments. The case made in the manuscript for aqueous processing was confined to wintertime when the air mass arrived at Heshan from upstream cities. We agree with the reviewer that the diversity of formation pathways of oxalic acid would present in the atmosphere. In field measurements, these pathways would be difficult to differentiate. In the last section of the manuscript, we have specifically discussed the limitation for the use of oxalic acid as an aqueous-phase SOA tracer (please refer to the paragraph from Line 447-459).

The revised manuscript:

In this study, the $\delta^{13}\text{C}$ differences of oxalic acid was dominantly attributed to kinetic isotope effects in atmospheric processing. It should be noted that the differences in the isotopic signatures of emission sources may also contribute to the isotopic variations of oxalic acid. However, these factors are not quantitatively constrained. We also stress that apart from aqueous-phase processes, both biomass burning and gas-phase photochemical aging could also be sources for oxalic acid. To differentiating these pathways in field measurements is still challenging. Therefore, cautious interpretations of oxalic acid isotopic signals are needed where there are intensive biomass burning, extensive atmospheric aging, and seasonality of C3/C4 vegetation changes. Future dual-carbon isotope studies on other aqSOA constituents (e.g., glyoxal) are strongly warranted to better constrain the chemical mechanism responsible for fossil-derived aqSOA formation. (Line 447-459)